**Technical Report**

# Single-cell Micro-C profiles 3D genome structures at high resolution and characterizes multi-enhancer hubs

Honggui Wu ®[1,2,4,5], Jiankun Zhang ®[1,2,5], Longzhi Tan ®[3,5] &
Xiaoliang Sunney Xie[1,2] ✉

In animal genomes, regulatory DNA elements called enhancers govern precise spatiotemporal gene expression patterns in specific cell types. However, the spatial organization of enhancers within the nucleus to regulate target genes remains poorly understood. Here we report single-cell Micro-C (scMicro-C), a micrococcal nuclease-based three-dimensional (3D) genome mapping technique with an improved spatial resolution of 5 kb, and identified a specialized 3D enhancer structure termed 'promoter–enhancer stripes (PESs)', connecting a gene's promoter to multiple enhancers. PES are formed by cohesin-mediated loop extrusion, which potentially brings multiple enhancers to the promoter. Further, we observed the prevalence of multi-enhancer hubs on genes with PES within single-cell 3D genome structures, wherein multiple enhancers form a spatial cluster in association with the gene promoter. Through its improved resolution, scMicro-C elucidates how enhancers are spatially coordinated to control genes.

The human genome is hierarchically folded into a myriad of three-dimensional (3D) structures[1–4], composed of DNA loops at different genomic scales. This intricate organization facilitates interactions between regulatory elements, thereby exerting a profound influence on cellular identity and function. Enhancers are *cis*-regulatory elements (CREs) bound by a specific combination of transcription factors (TFs), which activate the expression of target genes by TF-recruited co-activators[5]. Notably, enhancers can be positioned at considerable genomic distances—several hundreds of kilobases to even megabases—away from their target genes[6]. Evidence indicates that many enhancers function through the formation of chromatin loops, bringing the enhancer and promoter into spatial proximity[7,8]. However, the investigation of the structure of enhancer–promoter (E–P) loops at the single-cell level has been impeded by limitations in spatial resolution.

There are more than 1 million CREs in the mammalian genome, among which tens of thousands are active within specific cell types. This diversity adds complexity to the regulatory relationship between enhancers and their target genes: multiple enhancers may collectively regulate a single gene, whereas a single enhancer may regulate one gene or influence the expression of multiple genes. Recent studies have reported the existence of multi-enhancer hubs, where multiple enhancers form a spatial cluster to coordinate gene regulation[9–15], including, in some cases, enhancers from different chromosomes[16]. Despite the observation of multi-way interactions among enhancers using long-read 3C derivatives[14,15] and SPRITE[17] techniques, the prevalence and 3D organization of multi-enhancer hub structures within individual cells remain poorly understood.

Loop extrusion has a central role in shaping 3D genome organization. During this process, the structural maintenance of chromosome (SMC) complexes, condensin and cohesin, bind to chromatin and reel flanking DNA into growing loops until the complexes run into roadblocks such as convergently oriented CCCTC-binding factor (CTCF)[18–24]. Loop extrusion is also suggested to have a role in facilitating the establishment of E–P interactions[25–27]. Loop extrusion may offer an

[1]Biomedical Pioneering Innovation Center (BIOPIC) and School of Life Sciences, Peking University, Beijing, China. [2]Changping Laboratory, Beijing, China. [3]Department of Neurobiology, Stanford University, Stanford, CA, USA. [4]Present address: Broad Institute of MIT and Harvard, Cambridge, MA, USA. [5]These authors contributed equally: Honggui Wu, Jiankun Zhang, Longzhi Tan. ✉e-mail: sunneyxie@biopic.pku.edu.cn

efficient mechanism for promoters or CREs to scan adjacent chromatin in search of their targets[12,28–31], although this role for extrusion has not been extensively characterized.

Chromosome conformation capture (3C or Hi-C) assays have advanced the understanding of 3D genome structures by generating genome-wide contact maps. These maps are created using restriction enzymes to cut specific sequences, allowing spatially proximal DNA fragments to ligate before conducting whole-genome sequencing[32,33]. Bulk Hi-C has relatively low resolution due to restriction enzymes having limited cutting sites in the genome. Recently, Micro-C has been developed using micrococcal nuclease (MNase), which cuts genomic DNA between two nucleosomes regardless of specific sequence, creating nucleosome-sized fragments for proximity ligation[34–36], thereby achieving much higher resolution. However, bulk Hi-C and Micro-C measurements can only measure population-averaged contacts and are unable to distinguish the variability of chromosome folding among individual cells. To solve this problem, we previously developed Dip-C to determine the 3D genome structure of a single human cell at 20-kb resolution by distinguishing the paternal and maternal alleles based on their SNPs[37]. In this work, we further improved the precision of single-cell 3D genome determination by developing single-cell Micro-C (scMicro-C).

## Results

### Development of scMicro-C

The previously reported bulk Micro-C chemistry, although offering nucleosome-level contact maps, cannot be applied to single cells because of substantial loss of DNA caused by overdigestion and low ligation efficiency[34,35]. To achieve scMicro-C, we implemented three key improvements (Fig. 1a; Methods). First, we employed an ionic detergent (sodium dodecyl sulfate (SDS)) to solubilize chromatin and improve ligation efficiency. Second, we systematically titrated the degree of MNase digestion to assess the effect on chromatin structure detection. Third, we omitted biotin enrichment and adopted a transposon-based whole-genome amplification (WGA) technique called multiplex end-tagging amplification (META)[37] to enhance the detection of chromatin contacts.

Inspired by the Hi-C procedure[38], we introduced SDS treatment after MNase fragmentation in the Micro-C procedure. This step improved the chromatin accessibility for end-repair enzymes, thereby boosting ligation efficiency compared to the original bulk Micro-C procedure[34,35], independent of chromatin digestion levels (Fig. 1b and Extended Data Fig. 1a). To test whether enhanced ligation efficiency improves the detection of chromatin contacts in individual cells, we isolated and processed single cells and performed WGA and sequencing. The data show that SDS treatment increases detected chromatin contacts by 8.1 times at similar sequencing depths (Fig. 1c, Extended Data Fig. 1b–d and Supplementary Table 1). This may be attributable to the larger size of end repair enzymes (T4 polynucleotide kinase = 132 kDa and Klenow fragment = 68.2 kDa) compared to MNase (16.9 kDa), which poses challenges in accessing chromatin without SDS.

After establishing a protocol with high ligation efficiency, we next aimed to investigate how MNase digestion levels impact chromatin structure detection in scMicro-C experiments. Using four distinct MNase concentrations (200U, 600U, 800U and 1,000U containing approximately 4 million nuclei in a 100-μl reaction volume), we tested varying digestion degrees from low to high (Fig. 1d). Notably, even the highest concentration (1,000U) did not reach the level of digestion of standard bulk Micro-C. We note that the high degree of digestion in bulk Micro-C statistically diminished ligation efficiency (Extended Data Fig. 1a), as previously observed in the Micro-Capture-C (MCC) procedure[39]. Subsequently, we employed fluorescence-activated nuclei sorting to sort individual nuclei into a 96-well plate for each MNase concentration. Sequencing analysis revealed median counts of chromatin contacts per cell−671.6k (200U), 817.1k (600U), 863.2k

(800U) and 874.4k (1,000U; Extended Data Fig. 2a). We observed slightly increased inter-chromosomal contacts at higher levels of MNase digestion (Extended Data Fig. 2b).

With increasing MNase concentration, the single-cell profiles exhibited a higher percentage of reads containing contacts (referred to as contact ratio) and a greater number of unique contacts at the same sequencing depth (Fig. 1e,f and Extended Data Fig. 2c). We note that the contact ratio is compromised when compared to bulk Micro-C employing biotin pulldown, but is notably higher than our previously developed scHi-C method, Dip-C[37], particularly at higher MNase concentrations (Fig. 1e,f).

Notably, we found that chromatin structures across various scales remain largely consistent among different MNase concentrations (Fig. 1g and Extended Data Fig. 2d–g). Like bulk Micro-C, scMicro-C preserves nucleosome occupancy and TF footprinting profiles[39], with nucleosome occupancy profiles becoming slightly sharper at higher MNase concentrations (Fig. 1h and Extended Data Fig. 3). These features are not offered by restriction enzyme-based scHi-C (Fig. 1h and Extended Data Fig. 3). Based on our observations, we determined that the optimal choice for scMicro-C experiments is a digestion level of 800U.

Following the optimal scMicro-C procedure, we applied scMicro-C to GM12878 cells. In total, we profiled over 800 GM12878 cells (Supplementary Table 1) and kept 724 high-quality cells after quality control (Methods), achieving a median of 835k unique chromatin contacts per cell (s.d. = 467k, min = 228k and max = 3.5 m). To assess the fidelity of scMicro-C in capturing multiscale 3D genome organization, we aggregated all single-cell profiles (referred to as ensemble scMicro-C), yielding a contact map containing 750 million contacts.

To validate our modified Micro-C methodology, we generated two bulk Micro-C datasets for GM12878 cells using distinct protocols, as detailed in the Methods (modified bulk Micro-C). One dataset adhered to our modified Micro-C protocol, denoted as dataset 1, while the other dataset was produced using the original Micro-C procedure[34], designated as dataset 2. The comparison between the two datasets confirms that the modified protocol does not compromise data quality (Supplementary Fig. 1). Together, the two datasets yielded a total of 4.4 billion valid contacts. Upon comparison with published GM12878 Hi-C data at the highest available sequencing depth (4.9 billion contacts)[40], our bulk Micro-C data exhibited a higher signal-to-noise ratio, as evidenced by a greater number of chromatin loops (HICCUPS−20882 versus 9738) and chromatin stripes (Stripenn−3414 versus 2722), and stronger chromatin loop and stripe signals (Supplementary Fig. 2).

Comparison between bulk Micro-C and ensemble scMicro-C reveals a high level of agreement across scales, encompassing A/B compartments, topologically associating domains and fine-scale chromatin loops (Fig. 1i,j). Notably, ensemble scMicro-C delineates chromatin features at resolutions as fine as 1 kb (Extended Data Fig. 4c). Quantitatively, our bulk Micro-C achieves 1-kb resolution, and ensemble scMicro-C 5-kb resolution (Supplementary Fig. 3; Methods), with resolution defined as previously described[40]. Additionally, akin to Micro-C, when compared to bulk Hi-C and scHi-C methodologies, scMicro-C exhibits a superior signal-to-noise ratio in the detection of chromatin loops and stripes (Extended Data Fig. 4).

### scMicro-C resolves 3D genomes at kilobase resolution

Contact map-based analysis is inherently limited to two dimensions and is unable to accurately determine the position of genomic loci in 3D space. To address this limitation, several algorithms have been devised for reconstructing the 3D genome structure of both haploid and diploid cells from single-cell contact maps[37,41,42], providing insights into the radial organization of genomic loci[43], chromosome compaction[37] and multiway interactions. Notably, our previous restriction enzyme-based Dip-C method enabled the reconstruction of 3D genome structures of diploid cells at 20-kb resolution[37]. A resolution of 20 kb

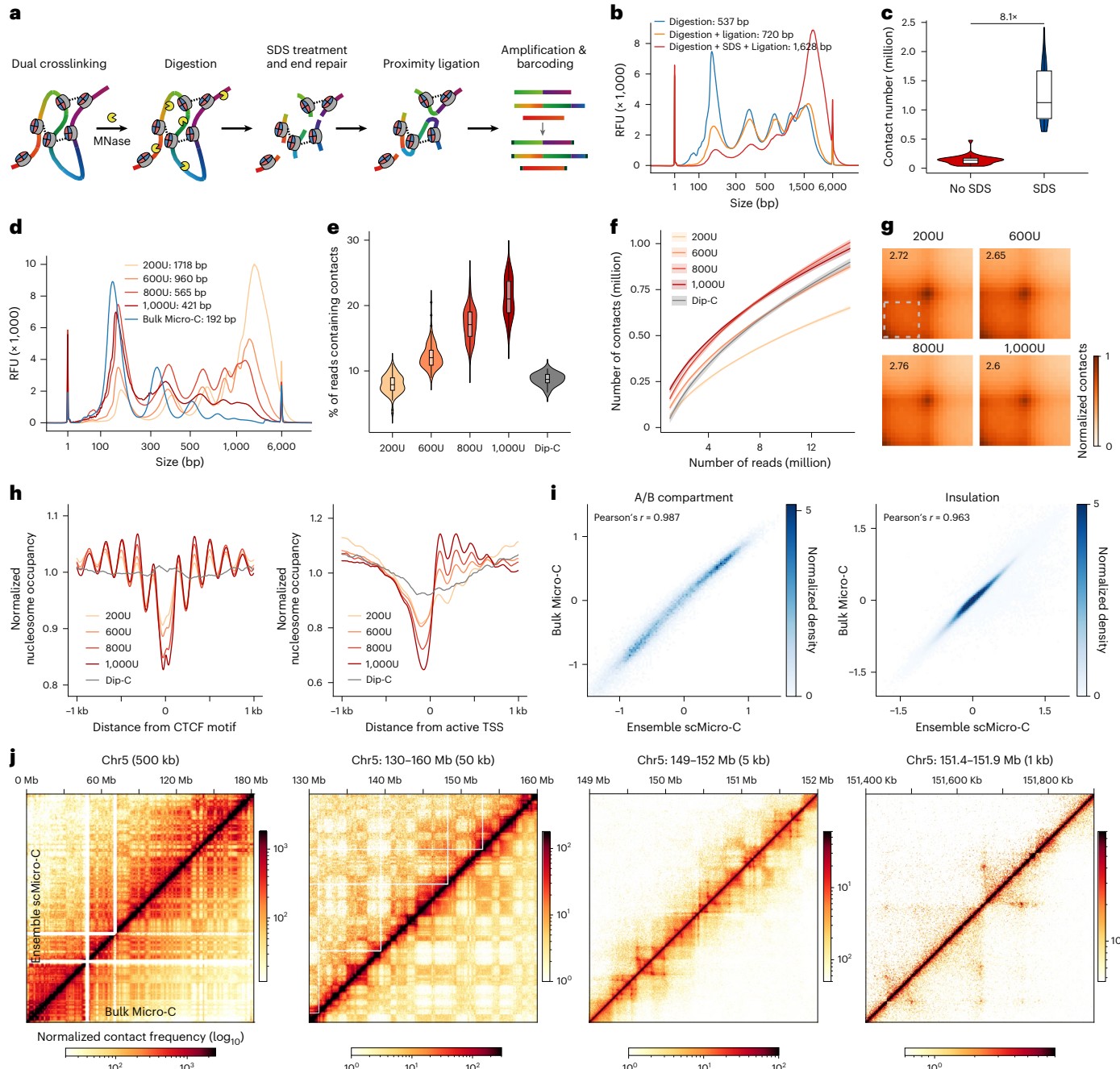

**Fig. 1 | Development of scMicro-C. a**, Schematic of the scMicro-C procedure. **b**, Chromatin length distribution to compare the ligation efficiency between SDS treatment and the original Micro-C protocol. Data generated by the capillary electrogram. **c**, Violin plot to compare the contact number between SDS treatment and original protocol (SDS, $n = 22$; original, $n = 33$). The box middle line represents the median value, the box limits the 25% and 75% quantiles, and the whiskers show the minimum and maximum. **d**, The chromatin length distribution depicts the degree of digestion resulting from MNase titration. **e,f**, Comparison among scMicro-C with different MNase titration and Dip-C, showing the percentage of reads containing contacts (**e**); the horizontal line and the box represent the median and quartiles, respectively (bottom). The whiskers indicate minima and maxima (**e**) and a downsample plot depicting the relationship between the number of reads and the number of unique contacts (**f**). The line indicates the median value, and the shadow indicates the 95% confidence interval. (**f**). 200U, $n = 96$; 600U, $n = 96$; 800U, $n = 96$; 1,000U, $n = 96$ and Dip-C, $n = 17$. **g**, Pile-up results of chromatin loops (bulk Micro-C detected loop set, $n = 45{,}174$) for four MNase titration groups. **h**, Normalized nucleosome occupancy around CTCF binding sites (left) and active TSS (right) of five scMicro-C datasets. **i**, Two-dimensional histograms to show the correlation of A/B compartment values at 100-kb resolution (left) and insulation scores at 10-kb resolution (right) between ensemble scMicro-C and bulk Micro-C. **j**, Contact maps of ensemble scMicro-C (top left) and bulk Micro-C (bottom right) at 500 kb, 50 kb, 5 kb and 1 kb from left to right to show the compartments, TADs and chromatin loops, respectively. TADs, topologically associating domains.

signifies that each particle of the 3D genome structure reconstruction represents a genomic locus of 20 kb. The definition of resolution is data-driven, reflecting minimal variability across multiple independent reconstructions (Methods). This concept aligns with DNA FISH-based

microscopy methods. Despite this advancement, the 20-kb resolution of the current scHi-C technique remains insufficient for investigating the intricate folding patterns of finer-scale structures like E–P loops and chromatin stripes.

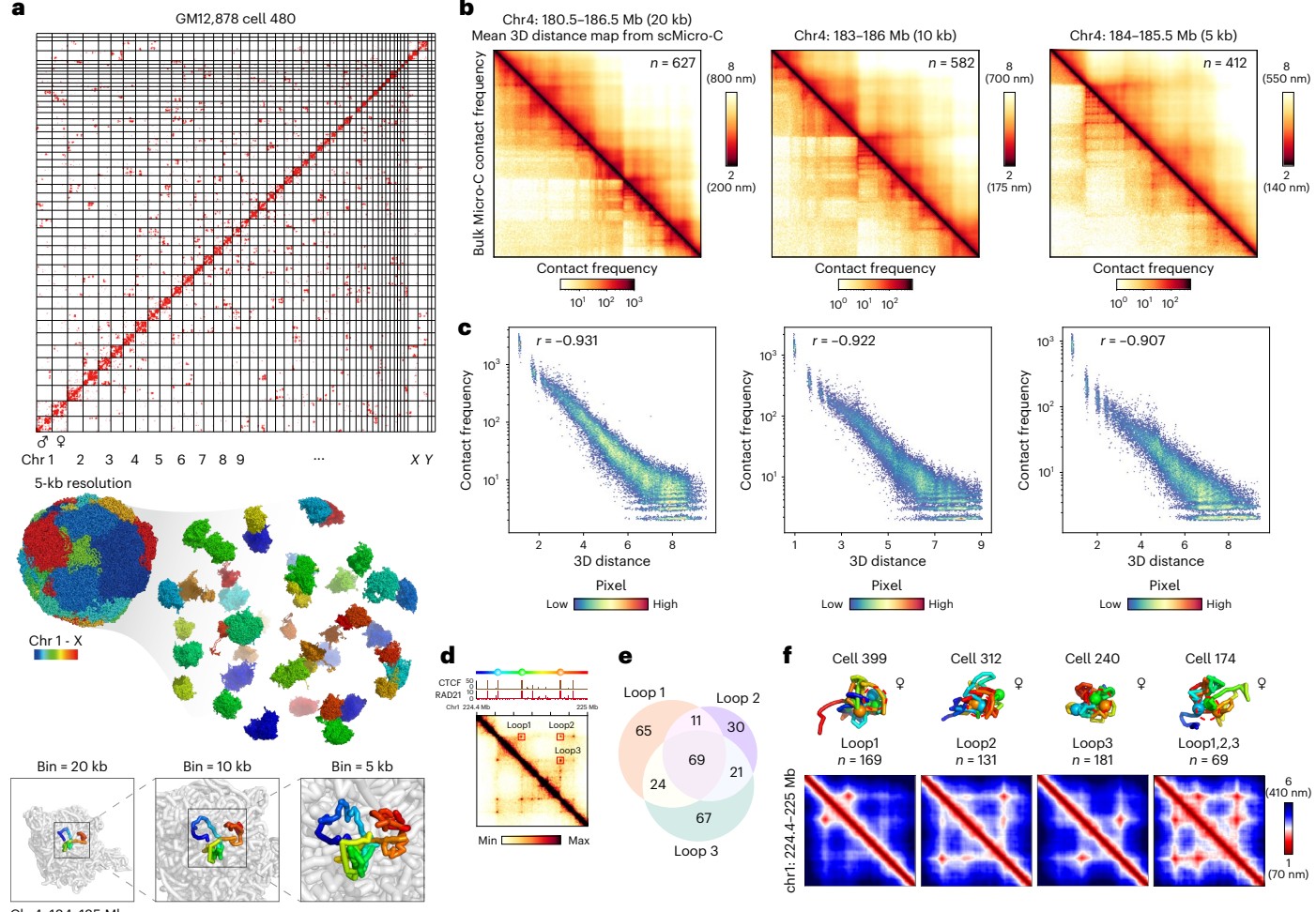

**Fig. 2 | Kilobase 3D genome structures of scMicro-C. a,** A haplotype-imputed contact map of a representative GM12878 cell (top), the corresponding reconstructed 3D genome structure at 5-kb resolution (middle) and a selected region (chr4: 184–185 Mb) are shown at 20-kb, 10-kb and 5-kb resolution (bottom). **b,** Heatmaps show the mean 3D distance matrices measured from single-cell 3D genome structures (top right) and the bulk contact frequency maps (bottom left) for the corresponding regions at 20-kb, 10-kb and 5-kb resolution, respectively. **c,** Scatter plots show the correlation between mean 3D distance measured from single-cell 3D genome structures and contact probability measured from bulk Micro-C at 20-kb, 10-kb and 5-kb resolution, respectively, of the same region at **b. d,** Contact map of a nested chromatin loop region (chr1: 224.4–225 Mb), CTCF and RAD21 ChIP–seq tracks, and cartoon schematics of three loop anchors were shown at the top. **e,** Venn diagram to show the overlap of the occurrence of three nested chromatin loops. A chromatin loop was defined as being formed when the 3D distance between two loop anchors was less than 3.5 particle radii (~240 nm). For this region, 428 single-cell 3D structures were available. The co-occurrence between loop1 and loop2, $P = 2.3 \times 10^{-9}$; co-occurrence between loop1 and loop3, $P = 2.43 \times 10^{-5}$; co-occurrence between loop1 and loop3, $P = 1.02 \times 10^{-12}$. A hypergeometric test (one-sided) was used. **f,** Representative single-cell 3D chromatin structures (top) and corresponding mean 3D distance matrices (bottom) for forming loop1, loop2, loop3 and co-occurrence of three loops.

Applying our previously developed Dip-C algorithm to scMicro-C data, we reconstructed 3D genome structures of individual GM12878 cells (Methods). Among the 724 cells analyzed, 358 cells (49.4%), 335 cells (46.1%) and 281 cells (38.8%) resolve whole-cell 3D genome structures of 20-kb, 10-kb and 5-kb resolution, respectively. Only cells demonstrating low uncertainty at each specific resolution were considered for downstream structure-based analysis (Methods). Quality control confirmed that cells displayed high consistency across different replicates and resolutions (Extended Data Fig. 5a,b). The attainment of 3D genome structures at a 5-kb 'particle size' enables the exploration of the finer-scale folding intricacies within chromatin structures (Fig. 2a).

To assess the fidelity of the reconstructed single-cell 3D genome structures, we measured the pairwise 3D distance matrices, which exhibit a high degree of anticorrelation with bulk Micro-C contact frequency across various resolutions (Pearson's r values = −0.931 at 20 kb, −0.922 at 10 kb and −0.907 at 5 kb; Fig. 2b,c). Consistent results were also observed in other genomic regions (Extended Data Fig. 5c). These

findings confirmed that the single-cell kilobase-resolution 3D structures resolved by scMicro-C are both reliable and rich in information.

Reanalysis of deeply sequenced Dip-C data revealed the presence of limited 5-kb resolution cells. Despite the limited number of Dip-C cells, we compared the reconstructed 3D genome structures with those obtained from scMicro-C and observed a high level of consistency between the two methods (Extended Data Fig. 5d,e). However, the low number of 5-kb resolution Dip-C cells precludes quantitative benchmarking. Nonetheless, based on the visual comparison of contact maps, we speculate that scMicro-C-derived 3D genome structures offer superior analysis of fine-scale chromatin structure compared to restriction enzyme-based Dip-C.

Using hundreds of 5-kb single-cell 3D genome structures, we investigated fine-scale chromatin folding, particularly multiway interactions. Although the majority of contacts identified through scMicro-C were pairwise, multiple pairwise interactions within a 5-kb bin implied simultaneous multi-loci proximity in 3D genome structures. Focusing

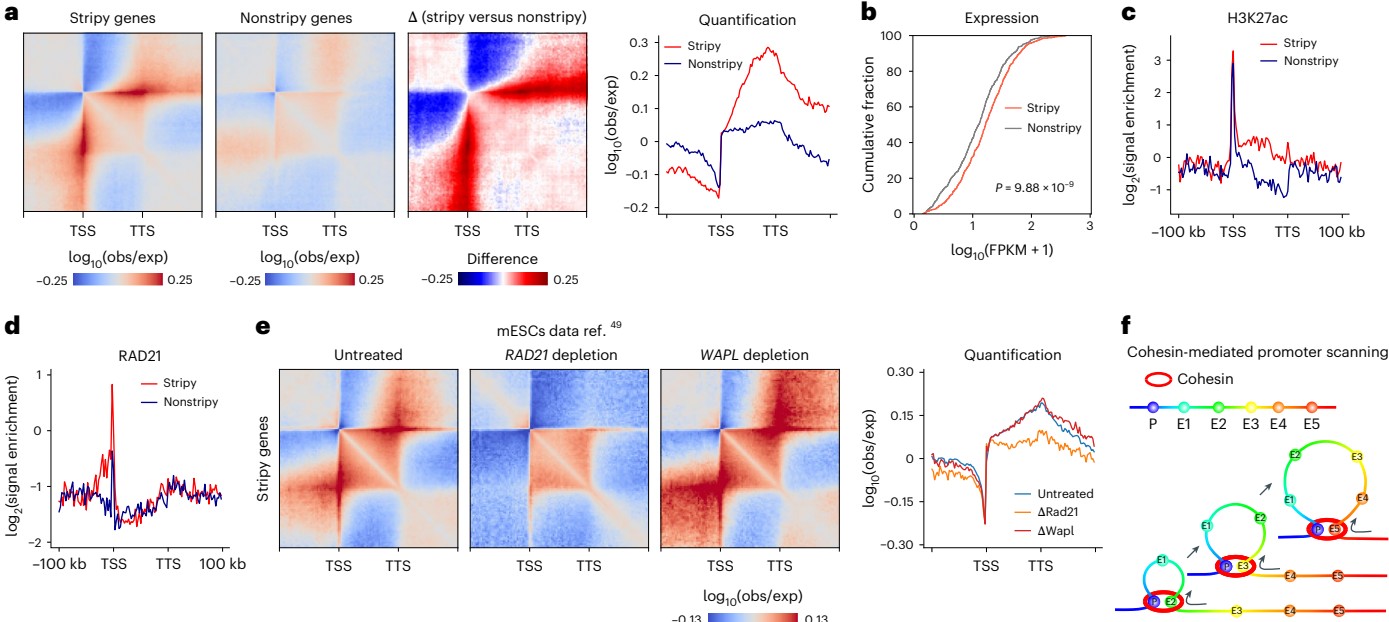

**Fig. 3 | Characterization of PESs. a**, Pile-up results of GM12878 PES genes (left, $n$ = 819), length-distribution-matched non-PES genes (left center, $n$ = 819) and the difference between stripy and nonstripy genes at 1-kb resolution (right center). Right, pile-up results, rescaled between the TSS and the TTS. **b**, Cumulative curves of expression level for PES and non-PES genes of GM12878. A two-sided Mann–Whitney $U$ test was used. **c**, Metagene plots of mean H3K27ac signal of stripy and nonstripy genes of GM12878. **d**, Metagene plots of mean RAD21 signal of PES and non-PES genes of GM12878. **e**, Pile-up results of PES genes ($n$ = 853) for untreated, *RAD21*-depleted and *WAPL*-depleted Micro-C data from mESC at 1-kb resolution. Signal value along the TSS is shown on the far right. Data are from ref. 49. **f**, Cartoon model depicting cohesin-mediated loop extrusion facilitating the scanning of downstream enhancers by the promoter.

on the nested CTCF-anchored chromatin loop (Fig. 2d), characterized by the formation of pairwise chromatin loops between loop anchors, we assessed whether these formed independently or concurrently (Supplementary Fig. 4a). Analysis of the overlaps in single-cell 3D structures that form these loops revealed a statistically significant tendency for nested chromatin loops to form simultaneously (Fig. 2e). Our scMicro-C 3D structures directly visualized the dynamic loop formation events (Fig. 2f and Supplementary Fig. 4b), with similar results observed in other regions featuring nested chromatin loops (Supplementary Fig. 4c–h). Nested chromatin loop formation was also observed in the SPRITE and MC-4C techniques[12,17], which detect multi-way high-order interactions. We speculate that the concurrence of nested CTCF loops is driven by the synergistic stabilization effect of spatially clustered CTCFs, which protects cohesin against WAPL-mediated release[44].

### Characterization of promoter–enhancer stripe (PES)

We next focused on analyzing E–P interactions. We observed that a large number of genes displayed a distinct PES chromatin structure, characterized by a line-like structure on the chromatin contact map. These stripes are anchored at the transcription start site (TSS) and extend in the direction of transcription, indicating frequent and sustained promoter–gene body interactions (Extended Data Fig. 6a).

A similar stripe structure has been reported in previous studies, including at the *SOX9* gene (referred to as a promoter-associated stripe), in the Hi-C pile-up analysis of *Stag2*-knockout-downregulated genes (referred to as promoter-anchored stripes) and in the pile-up results at the active promoter region of bulk Micro-C data (referred to as gene stripes)[34,45,46], albeit with a limited number of genes and without extensive characterization. Here we named genes exhibiting this PES configuration as 'PES genes', which included key TFs essential for B-cell function, such as early B cell factor 1 (EBF1) and interferon regulatory factor 2 (IRF2). In contrast to PES, previous studies focused primarily on 'architectural stripes', which are commonly located at topologically

associating domain boundaries and formed between two strong convergently oriented CTCF binding sites[47]. These architectural stripes are typically longer and stronger than PES (Supplementary Fig. 1f) and are not always associated with genes. However, it remains possible that certain PES overlap with or could themselves be considered architectural stripes.

Upon examination, we observed the presence of multiple H3K27ac-marked enhancers distributed along the PES (Extended Data Fig. 6a). Noticeably, these stripes frequently exhibited focal interactions at the enhancer sites (Extended Data Fig. 6a). Motivated by this observation, we performed de novo detection of PES genes in GM12878 using our bulk Micro-C dataset. We identified 819 genes with PES of 4,077 active long genes (≥50 kb) in GM12878, and pile-up results confirmed the existence of the PES (Fig. 3a and Extended Data Fig. 6b,c). Notably, genes with PES exhibit significantly higher expression levels than genes without ($P$ = 9.88 × 10⁻⁹; Fig. 3b).

Metagene analysis revealed stronger enhancer signals (as indicated by the H3K27ac signal) in stripy and nonstripy gene bodies (Fig. 3c). These findings were reinforced by activity-by-contact model predictions[48], which identified multiple stripy gene-associated enhancers distributed along PES (Extended Data Fig. 7a–c,i). Promoter–enhancer interaction pile-ups revealed localized intensity peaks at enhancer loci (Extended Data Fig. 6d), and robust interactions among multiple enhancers regulating the same stripy gene suggested the formation of enhancer hubs (Extended Data Fig. 6e). Collectively, these findings suggest that the potential function of the PES is to facilitate interactions between the promoter and multiple downstream enhancers.

We then focused on elucidating the mechanism of PES formation. We noted statistically stronger cohesin binding at stripy gene promoters compared to nonstripy genes, as indicated by cohesin subunits RAD21, SMC1A and SMC3 (Fig. 3d and Extended Data Fig. 7f,i), indicating cohesin's involvement in PES establishment. To test this hypothesis, we examined the effects of cohesin depletion using a Micro-C dataset in mouse embryonic stem cells (mESC)[49] and observed statistically

reduced PES formation after cohesin depletion. While cohesin unloading factor−WAPL depletion extended PES downstream (Fig. 3e and Extended Data Fig. 7g,h). These experiments demonstrate that PES formation is dependent on cohesin-mediated loop extrusion.

We next investigated whether E−P and enhancer−enhancer (E−E) interactions in genes with PES were influenced by the depletion of cohesin. Our re-analysis of this cohesin depletion Micro-C data[49] in the context of PES confirms that both E−P and E−E interactions were statistically attenuated following cohesin depletion, with a more pronounced reduction in stripy compared to nonstripy genes (Extended Data Fig. 7j,k). Specifically, 21% of E−P interactions in genes with PES were identified as E−P loops, compared to 10.5% in genes without PES. Among these E−P loops, 48.5% in PES genes were statistically weakened, whereas only 35.5% of E−P loops in genes without PES exhibited such weakening (Extended Data Fig. 7l).

Given the data, we hypothesize that cohesin-mediated loop extrusion facilitates promoter scanning of downstream genomic loci, thereby fostering interactions with multiple enhancers (Fig. 3f). However, further mechanistic studies are required to test this hypothesis. Intriguingly, the presence of nested tiny stripes derived from enhancers in certain stripy genes indicates that the scanning may also be initiated from enhancers (Extended Data Fig. 6a), a possibility that is not represented in the current model schematic.

### scMicro-C visualizes dynamic E−P interactions in PES genes

Upon gaining an understanding of the function and formation mechanism of PES, we aimed to characterize the E−P interactions of genes with PES. To do this, we used scMicro-C data to visualize the 3D structure of a specific PES gene called *EBF1*. This gene is located on chromosome 5, spans 404 kb, and has an expression level of 7.59 FPKM. EBF1 is a key regulator of B cell lineage development[50].

The *EBF1* locus exhibits a distinct PES on the bulk Micro-C and ensemble scMicro-C contact map, connecting its promoter to seven activity-by-contact model-predicted *EBF1* enhancers (named E1−E7; Fig. 4a). Notably, multiple nested weak stripes within the *EBF1* PES connecting multiple enhancers were clearly visible (Fig. 4a).

To explore the dynamics of PES formation, we sorted single-cell contact maps based on interactions with the TSS region, observing a gradual transition of contacts from the TSS to downstream regions (Fig. 4b). This series of snapshots captures the dynamic contact between the TSS and downstream loci, highlighting the dynamic nature of the observed 'stripes' across individual cells. In addition to the main transition line, sporadic contacts were also noted, indicating multiple pairwise interactions between downstream loci and the TSS in individual cells (Fig. 4b).

Three-dimensional distance analysis of six enhancer loci (E1−E7, E3/E4 merged at 5-kb resolution; Fig. 4c) showed looping events between *EBF1* promoter and downstream enhancers (E2−E7) occurring in approximately 30% of single-cell structures (Fig. 4d), with a looping event

defined as a 3D distance between the TSS (P) and another locus (E1−E7) less than 3.5 particle radii (~240 nm, see Methods). This frequency contrasts with nonstripy controls, where looping frequency declined with genomic distance (Fig. 4d and Supplementary Fig. 5), indicating that genes exhibiting PES have a higher frequency of E−P interactions.

We visualized the dynamic looping process of PES using single-cell 3D genome structures, directly capturing the dynamic E−P looping events between the promoter and multiple downstream enhancers (Fig. 4e). Our observations at the *EBF1* locus were corroborated by similar findings in the analysis of the *IRF2* gene (chr4: 88.9 kb, 47.7 FPKM), which, like *EBF1* exhibited a prominent PES structure and contained multiple enhancers within the stripe region (Extended Data Fig. 8a−d). These findings collectively suggest that E−P interactions are highly dynamic and heterogeneous among individual cells. It is important to note that while our observations provide insights into the dynamics of looping events based on snapshots of single-cell 3D genome structures, live-imaging techniques are needed for direct visualization of these dynamic chromatin looping events[51,52].

### Multi-enhancer hubs observed in single-cell 3D genomes

How multiple enhancers are spatially organized to regulate gene expression is not extensively studied. Previous studies using long-read 3C derivatives revealed cooperative 3D interactions between regulatory elements[14,15], such as simultaneous interactions between multiple enhancers and promoters at the α/β-globin gene locus[12,53] and interchromosomal multi-enhancer hubs regulating olfactory receptors (ORs)[54].

We aimed to understand how the multiple enhancers of stripy genes are arranged within individual 3D structures. Upon analyzing single-cell 3D structures at the *EBF1* locus, we observed a statistically significant number of structures in which more than one enhancer simultaneously interacts with the *EBF1* promoter (Fig. 4f). We refer to this structure as a 'multi-enhancer hub'. Three-dimensional visualization of these multi-enhancer hubs in individual cells revealed a spatial cluster of enhancers associating with the *EBF1* promoter (Fig. 4g). The simultaneous interaction between the *EBF1* promoter and multiple enhancers was also supported by findings from the scNanoHi-C study, which uses third-generation long-read sequencing to capture high-order multiway interactions[14].

Further examination of the 3D distance matrix of single-cell structures harboring between one and six enhancers revealed that a higher number of enhancers in the hub led to a more extensive interconnected enhancer network (Fig. 4h). Additionally, we observed that with an increase in the number of enhancers, chromatin structure became more compacted, as indicated by a decrease in the radius of gyration (Fig. 4i). These results highlight the cooperative interactions between multiple enhancers in association with the promoter, suggesting a distinctive mechanism of enhancer regulation in stripy genes.

To ascertain whether the presence of multi-enhancer hubs is a common feature among stripy genes or specific to the *EBF1* gene, we

**Fig. 4 | Multi-enhancer hubs captured by single-cell high-resolution 3D genome structures. a**, Contact map of ensemble scMicro-C (top left) and bulk Micro-C (bottom right) at the *EBF1* locus, accompanied by ChIP-seq signals of CTCF, H3K27ac, POLR2A and RAD21, and ABC model-predicted enhancers. **b**, Sorted scMicro-C contact profiles (*n* = 723) of the *EBF1* gene with PES; the 5-kb bins covering the *EBF1* TSS were used for this analysis. **c**, Schematic representation of the promoter and enhancers of the *EBF1* gene selected for single-cell 3D genome structure analysis. **d**, Line plot showing the percentage of single-cell 3D chromatin structures with the proximity of corresponding E−P pairs. Six regions without stripes were used as controls. The dots indicate the median value, and the whiskers of control regions indicate the standard deviation. **e**, Top row, schematic representation of the E−P loops in each cell (columns). Middle row, representative single-cell 3D structure of the E−P loops. Bottom row, mean 3D distance matrices of single-cell structures, with the depicted loop indicated by a yellow circle. **f**, Bar plot of the percentage of cells (*y* axis) that have the indicated

number of enhancer interactions with the promoter (*x* axis) at the *EBF1* (blue) or control locus (red). For the *EBF1* locus, the number of cells is indicated at the top of each bar. For control regions, the bar indicates the median value, and the error bar indicates the s.d. **g**, Representative single-cell 3D genome structures forming a multi-enhancer hub (red dashed circles) at *EBF1*. **h**, Mean 3D distance matrices measured from single-cell structures with the indicated number of enhancer loci interacting with *EBF1* promoter from left to right, respectively. **i**, Boxplot of the radius of gyration (*y*-axis) of single-cell 3D genome structures with the indicated number of enhancer loci (*x* axis−*n* = 114, *n* = 66, *n* = 51, *n* = 44, *n* = 28, *n* = 8 for each group) interacting with the *EBF1* promoter. Pairwise statistical tests showed statistical significance (*P* < 0.05, two-sided Wilcoxon rank sum exact test) for all pairs, except for 1–2 and 3–4. The six-enhancer group was excluded due to an insufficient number of single-cell structures. The centerline indicates the median, the box the 25% and 75% quantiles, and the whiskers the maximum and minimum values. ABC, activity-by-contact.

investigated several other stripy genes, including *IRF2* and *PIEZO2*. Analysis revealed the existence of multi-enhancer hub structures in these genes as well (Extended Data Fig. 8f and Supplementary Fig. 6a–e). This prevalence of multi-enhancer hubs was signified by the extensive E–E interactions observed in stripy genes (Extended Data Fig. 6c), confirming

the assembly of multi-enhancer hubs. Additionally, the conclusions regarding interenhancer connectivity and chromatin compaction were confirmed within the *IRF2* gene locus (Extended Data Fig. 8g,h).

In conclusion, single-cell kilobase-resolution 3D genome structures generated through scMicro-C technology allow for the

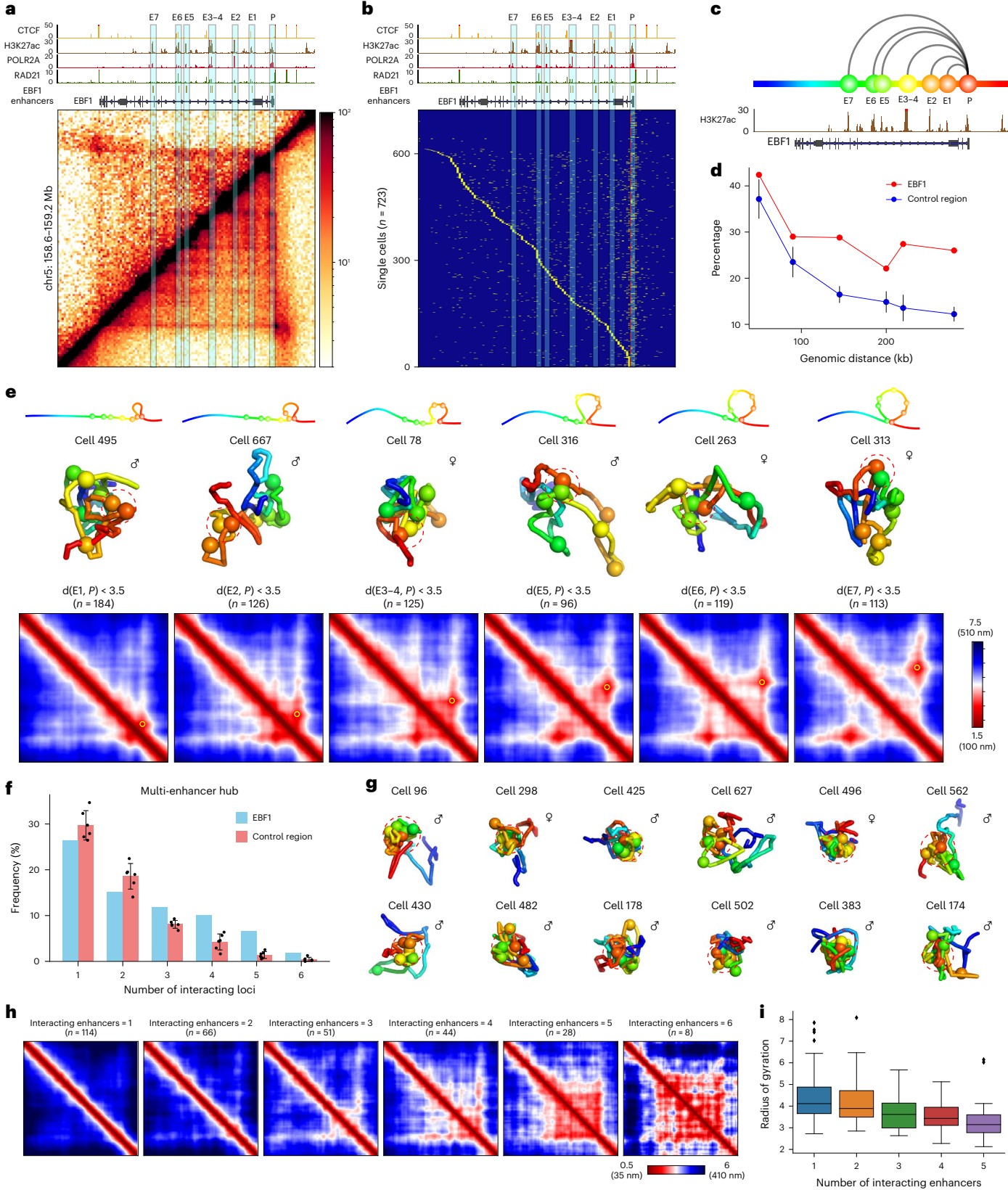

investigation of E–P interactions at the single-cell level. This approach has unveiled the dynamic and heterogeneous nature of E–P interactions within individual cells, leading to the identification of multi-enhancer hub structures in stripy genes. These findings offer valuable insights into the distinct regulatory mechanisms governing stripy genes through the coordination of multiple enhancers.

## Polymer simulations of PES genes

To dissect the mechanisms driving multi-enhancer interaction networks in genes with PES, we conducted polymer simulations using the open2c 'polychrom' framework. We simulated the *EBF1* locus with the following three key tunable parameters: cohesin-mediated loop extrusion (cohesin density, extrusion speed), cohesin capture probability of enhancers, and molecular affinities between promoters and enhancers (Extended Data Fig. 9a; Methods).

In our simulation settings, we positioned two strong boundary elements at the TSS and TTS of the *EBF1* gene, reflecting CTCF binding at these sites (Extended Data Fig. 9a,b). Consistent with previous studies, cohesin blocking was sufficient to establish the stripe pattern[46,47]. However, in the absence of E–P/E–E molecular affinities and enhancer-mediated cohesin capture, neither apparent E–P/E–E loops nor enhancer stripes formed (Extended Data Fig. 9c, left). Introducing strong molecular affinities among enhancers and the promoter generates E–P/E–E interactions but fails to replicate stripes unless weak cohesin capture by enhancers was included (Extended Data Fig. 9c–e). These patterns completely disappeared upon cohesin depletion. Strong E–P and E–E affinities combined with cohesin capture by enhancers best match experimental observations (Extended Data Fig. 9d,e, right).

Because the above observations were derived from population-ensemble contact maps, we next focused on single chromatin fiber traces to verify the existence of multi-enhancer hub structures in these simulations. We found that only simulations with strong affinities among promoters and enhancers and cohesin capture by enhancers produced a prominent number of multi-enhancer hub structures (Extended Data Fig. 10a,b). The multi-enhancer hub structures were almost completely depleted in the absence of cohesin (Extended Data Fig. 10a,b). Additionally, the distance matrices from simulated structures closely resemble observations from scMicro-C 3D genome structures (Extended Data Fig. 10c,d).

In summary, computational modeling of 3D chromatin folding provides a complementary approach for exploring possible regulatory mechanisms underlying stripe gene folding, offering insights that are challenging to obtain experimentally. The simulations also confirm the physical feasibility of interconnected multi-enhancer networks at the single-chromatin fiber level.

## Discussion

In this study, we improved the spatial resolution of single-cell 3D genome mapping by developing scMicro-C, allowing us to study the intricate folding of fine-scale chromatin structures such as chromatin loops and stripes. We argue that the multiple pairwise interactions overlapping a given genomic segment represent high-order multi-way interactions in 3D genome structures. This proposition is supported by the identification of cooperative interactions among nested chromatin loop anchors and multiple enhancers associated with stripy genes. Our dataset advances our knowledge of genome structure and function, serving as a valuable resource for examining the folding characteristics of uncharacterized chromatin structures, including chromatin jets[55] and fountains[56].

Recent studies reported that TFs, coactivators (like Mediator and BRD4) and RNA Pol II assemble into condensates at active DNA regulatory elements and mediate E–P interactions[57–60]. These transcriptional factors contain intrinsically disordered regions, which are believed to mediate their interactions through multivalent binding.

The potential mechanism underlying the formation of multi-enhancer hubs observed in our scMicro-C structures may involve condensates mediated by enhancers and promoter-bound TFs and coactivators. Furthermore, our findings demonstrate that PES, E–P and E–E interactions of PES genes are dependent on cohesin. Thus, cohesin-mediated loop extrusion likely aids the scanning of promoters through multiple downstream enhancers, promoting the establishment of molecular affinities between regulatory elements, leading to the formation of multi-enhancer hubs (Extended Data Fig. 10e). The collaboration between cohesin and molecular affinities among transcriptional proteins in establishing E–P interactions has been discussed previously[4] but requires further experimental validation.

Multi-enhancer hub structures have been identified in olfactory sensory neurons to ensure singular OR expression[54,61]. Knockout experiments have proven that the inter-chromosomal multi-enhancer hub is mediated by adapter protein LIM domain binding 1 (LDB1)[54]. LDB1 interacts with OR enhancer-bound TF LHX2 (a LIM domain protein) and possesses a self-association domain that facilitates long-range chromatin interactions through oligomerization[62]. Additionally, LDB1 has been reported to have an important role in mediating long-range chromatin interactions, such as the interactions between the locus control region and the β-globin locus[63]. Given that there are hundreds of TFs containing LIM domains in the human genome[64], LDB1 may act as a universal mediator of E–P interactions. Recent degron experiments have demonstrated the role of LDB1 in mediating interactions between CREs[65]. Therefore, the multi-enhancer hub structure in stripy genes may be mediated by LDB1 (Extended Data Fig. 10f).

Recently, a live-imaging study showed that a dynamic assembly of condensates involving super-enhancers and the promoter controls *Sox2* transcriptional bursting[66]. Given the dynamic nature of multi-enhancer hub assembly across individual cells, we speculate that the function of multi-enhancer hubs is to modulate transcriptional bursting of stripy genes. Verification of the existence of these multi-enhancer hubs will require orthogonal techniques. In the future, using the multiplexed DNA fluorescence in situ hybridization (FISH) technique to simultaneously label multiple enhancers and promoter loci may allow direct observation of cooperative interactions. Additionally, incorporating nascent RNA FISH and immunofluorescence targeting transcription-related factors will provide further insights into the functional roles and formation mechanisms of multi-enhancer hubs.

## Online content

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

## Methods

### Cell culture

GM12878 lymphoblastoid cells, karyotype normal diploid cells, were grown in RPMI1640 medium (Gibco; Thermo Fisher Scientific, 11875093) supplemented with 15% FBS (Gibco, 10091148) and 1% Pen/Strep (Gibco, 15140122), maintained at 37 °C with 5% $CO_2$ at recommended density. Upon collection, cells were spun down and washed once with ice-cold PBS.

### Modified bulk Micro-C

The Bulk Micro-C protocol was adopted from the published Micro-C protocol for mammalian cells[34,35] with several modifications. Briefly, collected cells were crosslinked with freshly made 1% PFA (EMS, 15714) followed by 3 mM DSG (Thermo Fisher Scientific, 20593), then cells were lysed and titrated to test the appropriate MNase concentration (Extended Data Fig. 1a). Before end repair, we added 50 µl 0.3% SDS (Sigma-Aldrich, 71736-100M) and incubated at 62 °C for 10 min, then quenched with 50 µl 3% Triton X-100 (Sigma-Aldrich, 93443) and incubated at 37 °C for 15 min. Then nuclei were end-repaired and biotin labeled, followed by in situ ligation with T4 ligase (NEB, B0202L). Before DNA extraction, nuclei were incubated with exonuclease III (NEB, M0206) to remove unligated biotinylated DNA ends. Extracted DNA, spanning all lengths, was sonicated to an average size of 350 bp for library preparation. Then, performed biotin pull-down and adapter ligation for sequencing.

We generated two datasets. The first followed a modified bulk Micro-C procedure, using all ligated fragments and sonication to ~350 bp for library preparation, termed dataset 1. Another exactly followed original bulk Micro-C protocol[34] without SDS treatment and only selected the ligated di-nucleosome fragments for library preparation, termed dataset 2.

### scMicro-C protocol

Our protocol was modified from mammalian cell Micro-C protocols with three key modifications. First, we optimized MNase digestion levels to produce longer DNA pieces and reduce DNA loss to ensure 40–50% mononucleosomes and 20–25% di-nucleosomes (average 500 bp) instead of 80–90% mononucleosomes and 10–20% dinucleosomes (average 200 bp) as originally Micro-C suggested, because over-digestion produces too short fragments, which hampers transposon-based WGA procedures. Second, we added an ionic detergent (SDS) to solubilize chromatin between the MNase digestion and end repair step, which both preserves Micro-C characteristic nucleosome-resolution chromatin interactions (as suggested by our bulk data) and dramatically increases ligation efficiency (Extended Data Fig. 1a), further increasing the length of the final product from ~450 bp (without SDS) to ~1,300 bp, which is critical for single-cell WGA. Third, we used our state-of-the-art WGA method, META, to further increase contact detection efficiency in single cells. We omitted all biotin-related steps to maximize the number of contacts detected per cell, as described in the Dip-C procedure[37]. For a comprehensive step-by-step guide to the scMicro-C protocol, please refer to protocols. io (https://doi.org/10.17504/protocols.io.kqdg39wbzg25/v1).

### Cell crosslinking

Cells were fixed with 1% PFA (EMS, 15714) at room temperature for 10 min with rotation. PFA was quenched by the addition of 2 M Tris–HCl, pH 7.5, to a final concentration of 0.75 M and incubated at room temperature for 5 min. Then wash twice with ice-cold 1× PBS supplemented with 1× BSA (centrifugation 3,000g for 5 min). After crosslinking, some pellets appear and then disappear after the sample is washed twice. Then cells were further fixed in 3 mM DSG in PBS and incubated at room temperature for 45 min. DSG was quenched by the addition of 2 M Tris–HCl, pH 7.5, to a final concentration of 0.75 M and incubated at room temperature for 5 min. After fixation, cells were washed twice with ice-cold 1× PBS supplemented with 1× BSA (centrifugation 3,000g for 5 min). The pellets were stored at −80 °C.

### MNase digestion

In total, 1 million cells were permeabilized with 100 µl Micro-C buffer1 (50 mM NaCl, 10 mM Tris–HCl pH 7.5, 5 mM $MgCl_2$, 1 mM $CaCl_2$, 0.2% IGEPAL CA630, 1× protease inhibitor cocktail) and incubated on ice for 20 min. The cell pellet was resuspended in 100 µl Micro-C buffer 1, and titrated to find the appropriate amounts of MNase (NEB, M0247S) to digest chromatin to 40–50% mononucleosomes and 20–25% dinucleosomes (Supplementary Fig. 2b), incubated at 37 °C for 10 min. Furthermore, 0.5 M ethylene glycol-bis(β-aminoethyl ether)-N,N,N′,N′-tetraacetic acid was added to a final concentration of 4 mM to stop the reaction, omitting the heat inactivation step.

*MNase titration.* To test the effect of MNase digestion efficiency on chromatin structure, we performed a series of MNase titrations (200U, 600U, 800U and 1,000U) to produce digestion degrees from low to high.

### End repair

To compare the ligation efficiency between SDS treatment and the original protocol, for the same MNase-digested sample, we split the sample into two aliquots, one treated with SDS and the other directly subjected to end repair.

### SDS treatment procedure

Cell pellets were resuspended in 50 µl 0.5% SDS and incubated at 62 °C for 10 min. A total of 170 µl of 1.5% Triton X-100 was added, followed by incubation at 37 °C for 15 min. The pellet was washed once with 100 µl Micro-C buffer 2 (50 mM NaCl, 10 mM Tris–HCl pH 7.5, 10 mM $MgCl_2$). End digestion and repair was executed with the following two steps: first, resuspend the cell pellet in 45 µl end repair buffer 1 (1× NEBuffer 2.1, 2 mM adenosine triphosphate, 5 mM dithiothreitol, 2.5 µl 10 U µl⁻¹ T4 polynucleotide kinase) and incubate at 37 °C for 15 min with gentle vortex to add 5′ phosphate and remove 3′ phosphoryl groups; next, add 5 µl 5 U µl⁻¹ Klenow fragment (NEB, M0210S) and incubate at 37 °C for 15 min with gentle vortex to remove 3′ overhangs. Blunt end repair was performed by the addition of 25 µl of end repair buffer 2 (200 µM dNTPs each, 1× T4 ligase buffer, 100 µg ml⁻¹ BSA), incubated at room temperature for 45 min. Wash once with 1 ml Micro-C buffer 3 (50 mM Tris–HCl pH 7.5, 10 mM $MgCl_2$).

### Proximity ligation

Ligation was performed by the addition of 250 µl ligation mix (1× T4 ligase buffer, 100 µg ml⁻¹ BSA, 20 U µl⁻¹ T4 ligase buffer) and rotating at room temperature for 2.5 h. Then the cell pellet was stained with DAPI/7-AAD and only G1-phase cells were sorted for downstream amplification.

### Plate-based single-cell amplification

Nuclei were sorted with BD Aria SORP under single-cell mode (FACSDiva v9.0 software) to 96-well PCR plates containing 2 µl lysis buffer (10 mM Tris pH 8.0, 20 mM NaCl, 1 mM ethylenediaminetetraacetic acid, 0.1% Triton X-100, 500 nM Carrier ssDNA, 1.5 mg ml⁻¹ QIAGEN protease), then nuclei were lysed by incubating at 50 °C for 1 h, 65 °C for 1 h and 70 °C for 15 min. After lysis, nuclei could be stored at −80 °C for several months. Lysed nuclei were first transposed by the addition of a 6 µl transposition mix (leading to a final concentration of 10 mM TAPS pH 8.5, 5 mM $MgCl_2$, 8% polyethylene glycol 8000 and 0.3 nM META transposome dimer) and incubated at 55 °C for 10 min. META transposome was assembled as previously described[37]. Transposition was stopped by the addition of a 2 µl stop mix (250 mM NaCl, 37.5 mM ethylenediaminetetraacetic acid, 2 mg ml⁻¹ QIAGEN protease) and incubation at 50 °C for 30 min and 70 °C for 15 min. The barcoding strategy is the same as previously described[67]. Then the transposed DNA was amplified by the addition of 15 µl preamplification mix (12.5 µl 2× Q5 master mix, 0.8 µl 50 µM META16 primer mix, 0.5 µl 100 mM $MgCl_2$, 1.2 µl $H_2O$) and incubated at 72 °C for 5 min, 98 °C for 30 s, 12 cycles of (98 °C for 10 s, 62 °C for 30 s, 72 °C for 2 min) and 65 °C for 5 min. Next, add 0.8 µl of 50 µM indexed META16-ADP1 primer and 0.8 µl of 50 µM META16-ADP2 primer to generate a 12 × 8 cell barcode combination for each 96-well plate, and

incubate at 98 °C for 30 s, three cycles of '98 °C for 10 s, 62 °C for 30 s, 72 °C for 2 min' at 65 °C for 5 min. After cell barcoding, a whole plate was pooled together for purification with ZYMO DCC5.

**Library preparation.** A total of 120 ng (10 μl) of the purified amplicon was used for each plate for library preparation. Add 40 μl PCR mix (25 μl 2× Q5 Master Mix, 5 μl NEBNext index primer i5 (E7600S), 5 μl NEBNext index primer i7 (E7600S) and 0.05 μl 100 mM MgCl₂) and incubate at 98 °C for 30 s, two cycles of '98 °C for 10 s, 68 °C for 30 s and 72 °C for 2 min' at 72 °C for 5 min. Then purified with 0.8× solid-phase reversible immobilization beads to remove <300 bp fragments.

### Analysis of bulk Micro-C data

**Generation of contact maps.** Bulk Micro-C datasets were processed using the distiller pipeline (https://github.com/open2c/distiller-nf, v0.3.3). In brief, raw FASTQ files were mapped to the human reference genome assembly GRCh38 using the BWA-MEM (v0.7.17). Pairs were extracted from the mapped reads using the pairtools package (https://github.com/open2c/pairtools, v0.3.0). PCR duplicates were then filtered out, and only pairs with MAPQ > 20 were kept. Contact matrices in the .mcool and .hic format were generated and balanced using the cooler (https://github.com/open2c/cooler, v.0.8.11) and Juicer package (https://github.com/aidenlab/juicer/, v1.19.02).

**Evaluation of data resolution.** Notably, the 'resolution' has distinct definitions in different contexts throughout the manuscript. The 'resolution' commonly denotes the bin size of a Hi-C or Micro-C contact matrix. For bulk or pseudobulk datasets, we adopted the standard criterion[40], which defines the resolution as the bin size at which ≥80% of loci have at least 1,000 contacts, consistent with other bulk studies. Owing to the limited DNA content in individual cells, this criterion is too stringent and is not applicable to single-cell studies. As such, for single-cell data, we defined the resolution as the bin size at which ≥80% of loci have at least one contact. This definition has been shown to be reliable and robust in a previous single-cell Hi-C study[68].

Moreover, the single-cell contact matrix allows for the reconstruction of 3D genome structures using a simulated annealing protocol. Contact noise and sparsity contribute to deviations between 3D genome structures generated from the same contact matrix across multiple rounds. Here, the resolution refers to the locus size used to reconstruct single-cell 3D genome structures and is determined through structure comparisons, which is fundamentally distinct from the definitions applied to contact matrices as discussed before.

**Contact scaling curves.** We used the cooltools expected-*cis* and logbin-expected functions (https://github.com/open2c/cooltools, v0.5.2) to calculate the normalized contact probability as a function of genomic separation within chromosome arms and the scaling derivatives of the curve on contact matrices at 1-kb resolution. The contact scaling curves using unbinned short-range contacts (contact distance < 10 kb) were calculated separately for each contact orientation (IN-OUT, IN-IN and OUT-OUT) using the cooltools compute_scaling function.

**A/B compartments.** We used the cooltools eigs-cis function to calculate A/B compartments at 500-kb and 100-kb resolution.

**Insulation scores.** The cooltools insulation function was used to calculate the insulation scores at 10-kb resolution (with a window of 100 kb). Genomic loci with boundary strength > 0.2 were considered insulation boundaries and used for downstream analyses. We used the bedtools intersect function to compare two lists of insulation boundaries, and a 10-kb offset on each side was tolerated when performing this intersection. Average profiles of insulation scores around boundaries were calculated using the deepTools computeMatrix function (https://github.com/deeptools/deepTools, v3.5.2).

**Chromatin loops.** The chromatin loops were identified using the Juicer HiCCUPS algorithm with default parameters and the chromosight (https://github.com/koszullab/chromosight, v1.6.2) package at 10-kb and 5-kb resolution. Chromatin loops between techniques and conditions were overlapped using the bedtools pairtopair function with '-slop 20000'. Pileup analyses of chromatin loops were performed using the coolpuppy package (https://github.com/open2c/coolpuppy, v1.0.0).

**Chromatin stripes.** We called chromatin stripes using two different algorithms, Stripenn (https://github.com/ysora/stripenn, v1.1.65.7) and StripeCaller (https://github.com/XiaoTaoWang/StripeCaller, v0.1.0), at 5-kb resolution. For StripeCaller, the chromatin stripes were extended to the main diagonal of the contact matrix, and redundant calls anchored at the same locus were merged into a single stripe with maximum length. The comparisons of chromatin stripes between different techniques and conditions were then performed based on the stripe anchor and orientation.

**PES.** After obtaining chromatin stripes, we used the Stripenn algorithm to identify promoter stripes at 10-kb and 5-kb resolution. At each resolution, we only focused on genes that are longer than ten bins. First, we performed intersections between promoters and chromatin stripe anchors and considered PES as those that anchored at the promoter. This was done by using the bedtools intersect function. Because many PES exhibited only moderate signals and Stripenn used a stringent filtering criterion, some clearly visible PES were missed. Next, we identified PES directly from the contact matrix using the Stripenn score function, which quantifies stripe signal strength, or stripiness, of any stripe-like region. For each gene, we used the promoter as the anchor point and the entire gene body as the stripe interval to calculate the stripiness. We note that the uncertainty in anchor localization has a large impact on the stripiness calculation and must be taken into account. To do so, we also took into account the two bins adjacent to the bin containing the promoter. The three bins provided a total of six possible anchors, among which three anchors were one bin wide, two anchors were two bins wide and one anchor was three bins wide, resulting in six possible regions. As recommended by the Stripenn tutorial, we considered genes with a positive stripiness to have a PES. If the gene had a positive stripiness in any of the six regions, we retained it. Here *P*-value filtering was not applied. Ultimately, we combined the PES obtained by intersections and quantifications and further combined PESs at different resolutions to generate a set of PES.

Pileup analyses for PES were performed using coolpuppy with '--flip-negative-strand –rescale –local'.

**Nucleosome occupancy.** The nucleosome occupancy signals were extracted from the mapped reads. Briefly, PCR duplicates were filtered out from raw SAM files using the samblaster package with '-ignoreUnmated -r'. The DANPOS2 package (https://sites.google.com/site/danposdoc/, v2.2.2) was then used to calculate the nucleosome occupancy signals with 'dpos-a 5 --count 1000000'. The average signals of nucleosome occupancy around genomic elements of interest were then calculated with computeMatrix and normalized by setting their mean to 1.

### Analysis of scMicro-C data

**Generation of contact maps.** Single-cell contact maps were generated from raw sequencing data as we previously described[37], using the hickit (https://github.com/lh3/hickit) and dip-c packages (https://github.com/tanlongzhi/dip-c). Briefly, raw sequencing data was mapped with 'bwa mem' and then proceeded with hickit to generate a contact pairs file with default parameters. Then the contact pairs were transformed into dip-c format for further analysis with dip-c.

**Cell exclusion.** Cells with less than 200k contacts or with >45% inter-chromosomal contacts were excluded.

**Generation of 3D genomes.** The generated contact pairs were imputed to generate a haplotype-resolved contact map using hickit. Genomic regions with chromosome abnormalities were excluded from the contact map and removed from the 3D genomes afterwards. Single-cell 3D structures were generated as we previously described[37], with the hickit package (with parameters '-M' and 'Sr1m -c1 -r10m -c2 -b4m -b1m -b200k -D5 -b50k -D5 -b20k -D5 -b10k -D5 -b5k'). We generated five replicate structures for each cell, using different random seeds (1–5). Repetitive regions were also removed from the 3D structure with 'dip-c clean3'. Similar to our previous studies, each 20 kb particle represents a radius of ~100 nm (~85 nm for each 10 kb particle and ~68 nm for each 5 kb particle).

**Three-dimensional genome structure alignment and quality control.** For each cell, the median and root mean square (r.m.s.) and r.m.s. deviation (r.m.s.d.; across all particles) were calculated with 'dip-c align' at 20 kb, 10 kb and 5 kb resolution. Structures with median r.m.s.d. ≤ 2 particle radii were considered as low uncertainty and kept for downstream single-cell 3D genome structure analysis.

**Three-dimensional genome structure visualization.** For whole cell structure visualization, the dip-c cleaned 3dg file was converted to an mmCIF file with 'dip-c color -n hg38.chr.txt' to color by chromosome, 'dip-c color -d3' to visualize chromosomal intermingling, 'dip-c color -c hg38.cpg. ${resolution/1000}k.txt' to visualize A/B compartment (CpG frequency).

To visualize a region of interest, the region was extracted from the whole cell by 'dip-c reg3' to generate 3dg file containing that region, then transformed to an mmCIF file with 'dip-c color -l hg38.chr.len'. The loop anchor file was generated by 'dip-c pos -l 'leg file'' to extract loop anchor 3dg positions from the whole cell structure, then transformed to mmCIF files by 'paste 'anchor name file' 'anchor color file' 'anchor 3dg file' | python /dip-c/scripts/name_color_x_y_z_to_cif.py'.

The mmCIF file was visualized in PyMol (https://pymol.org/2/).

**Distance matrix analysis.** For a region of interest, we performed two filtering criteria to select available single-cell 3D structures. First, we require the whole-cell 3D genome structure with median r.m.s.d. ≤ 2. Second, we require the 3D genome structure of that region with r.m.s.d. ≤ 1.5.

For single-cell distance matrix analysis, we extracted the region of interest from the whole cell 3D genome file with 'dip-c reg3' and then measured the distance matrix with 'dip-c rg -d'.

**Polymer simulations of loop extrusion.** We performed polymer simulations of loop extrusion with the 'open2c/polychrome' package[69] and the adapted version developed by the Boettiger laboratory[46,70]. The chromatin polymer chain consists of 1,200 monomers, with each monomer representing 500 bp. To match the chromatin structure of the *EBF1* locus, we positioned the promoter and TTS at monomer 1,000 and 190, respectively, and positioned 7 enhancers on the gene body at monomer 440, 560, 600, 710, 720, 820 and 900. Loop extrusion factors (LEFs, cohesin) were primarily loaded at the promoter with a probability of 90% (with parameters 'LFE lifetime = 200, LEF separation = 200, trajectory length = 5,000'). For simulations without loop extrusion, the number of LEFs was set to 0 as previously described[71]. We positioned a bidirectional extrusion-blocking element adjacent to the promoter at monomer 1,001 with a capture probability of 99%, which produces the prominent stripe pattern. To investigate the effect of loop extrusion on E–E and E–P interactions, we allowed enhancers to weakly block extrusion from the left with a capture probability of 30% or 60%. The stripe emanating from the TTS was simulated by adding the extrusion-blocking property to the TTS

monomer with the same capture probability as enhancers. In all simulations, a release rate of 0.1% was used for all extrusion-blocking elements. To study the effect of E–P adhesion, we added a self-affinity of $0.1 KT$ to all monomers and set the affinity between enhancers and promoters as 0.5 or 0.8 $KT$, respectively, where $K$ is the Boltzmann constant and $T$ is the temperature. The energy functions and the related parameters were unchanged from the previous work mentioned above. Ensemble contact maps were then generated from polymer traces using the polychrom 'monomerResolutionContactMapSubchains' function with a threshold of 5 monomer radii. For the analysis of individual polymer traces, we extracted coordinates using the polychrom 'polymerutils.fetch_block' function and then calculated distance matrices. The PDB file for each trace was generated using the polychrom 'polymerutils.save' function and visualized in PyMol.

## Statistics and reproducibility

A total of 810 cells from five batches were generated in this study, and 52 cells were excluded due to poor quality ('Cell exclusion'). For bulk Micro-C, two replicates were generated. Randomization and blinding were not applicable to this study. Specific statistics used for each analysis are referred to in the Methods. All aspects of benchmarking with Dip-C are described in detail in the Methods.

### Reporting summary

Further information on research design is available in the Nature Portfolio Reporting Summary linked to this article.

## Data availability

Raw sequencing data generated during this study have been deposited at the Sequence Read Archive with accession number PRJNA788160. The processed data generated in this study have been uploaded to the Gene Expression Omnibus under accession number Superseries GSE281150. The subseries includes GSE225201 for bulk Micro-C, GSE278191 for scMicro-C 1000U, GSE279479 for scMicro-C 600U, GSE279583 for scMicro-C 800U, GSE279732 for scMicro-C 200U, GSE280482 for SDS test, GSE192759 for scMicro-C pool. Source data are provided with this paper.

## Code availability

The code used in this study is publicly available at GitHub (https://github.com/tanlongzhi/dip-c, https://github.com/lh3/hickit and https://github.com/zhang-jiankun/Single-cell-Micro-C)[72–74].

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

## Acknowledgements

We thank the staff at the Peking University High-throughput Sequencing Center (HTSC) for help with next-generation sequencing and flow cytometry; Beijing Berry Genomics for help with next-generation sequencing; and we thank X. Ji (School of Life Sciences) and X. Zhang (Biomedical Pioneering Innovation Center) at Peking University for interactions. This work was financially supported by Changping Laboratory and the Noncommunicable Chronic Diseases-National Science and Technology Major Project (2023ZD0520000).

## Author contributions

H.W., L.T. and X.S.X. designed the experiments. H.W. performed the experiments. J.Z. and H.W. analyzed the data. H.W., J.Z., L.T. and X.S.X. wrote the manuscript.

## Competing interests

The authors declare no competing interests.

## Additional information

**Extended data** is available for this paper at https://doi.org/10.1038/s41588-025-02247-6.

**Correspondence and requests for materials** should be addressed to Xiaoliang Sunney Xie.

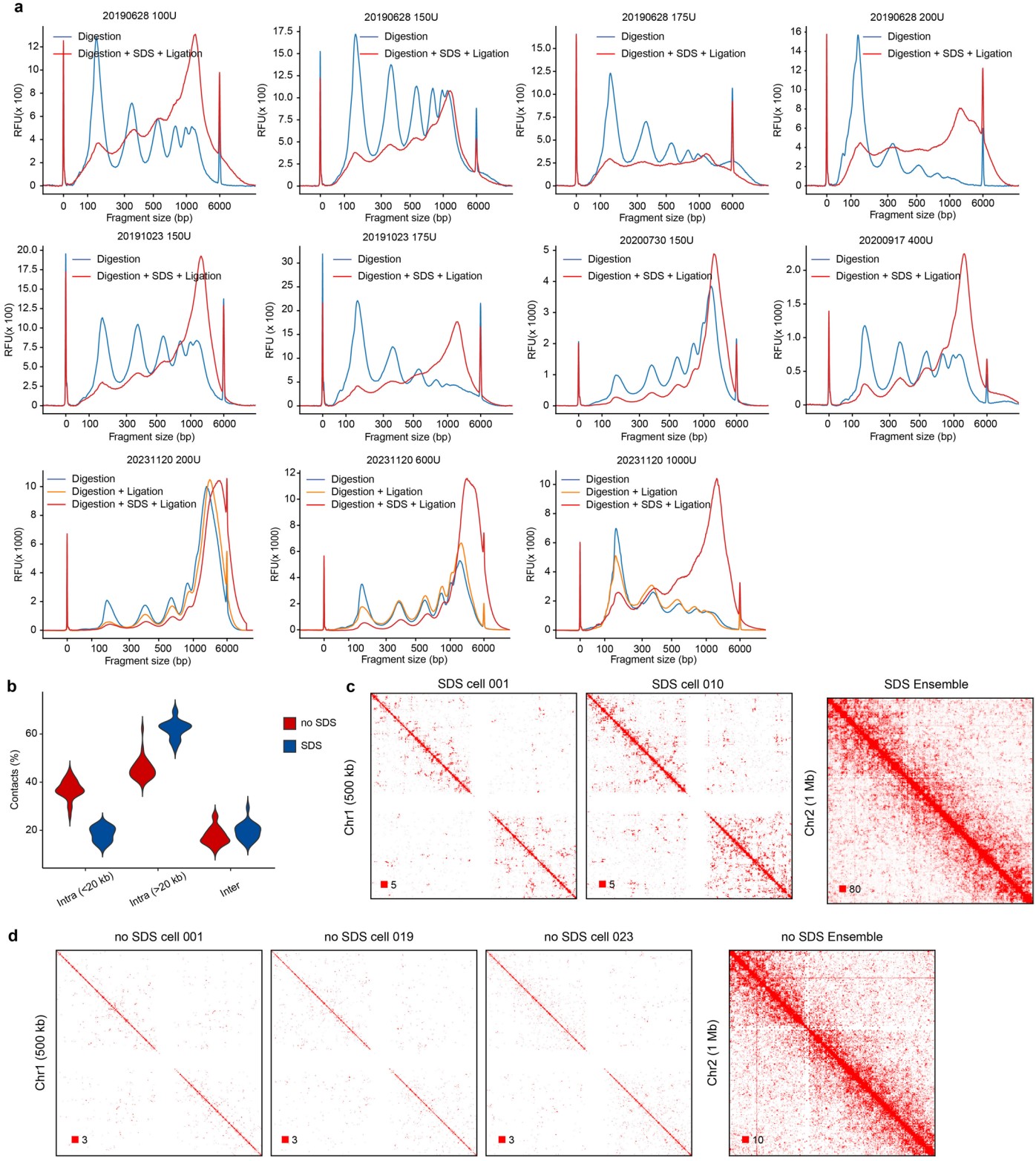

**Extended Data Fig. 1 | SDS treatment improves Micro-C ligation efficiency.**
**a**, Chromatin length distribution for different batches of MNase digestion. Results were obtained from the Fragment Analyzer. **b**, Violin plot compares the composition of contacts between SDS treatment and no SDS group. **c**, Contact maps of SDS treatment of single cells and ensemble. **d**, The same as **c** for no SDS group.

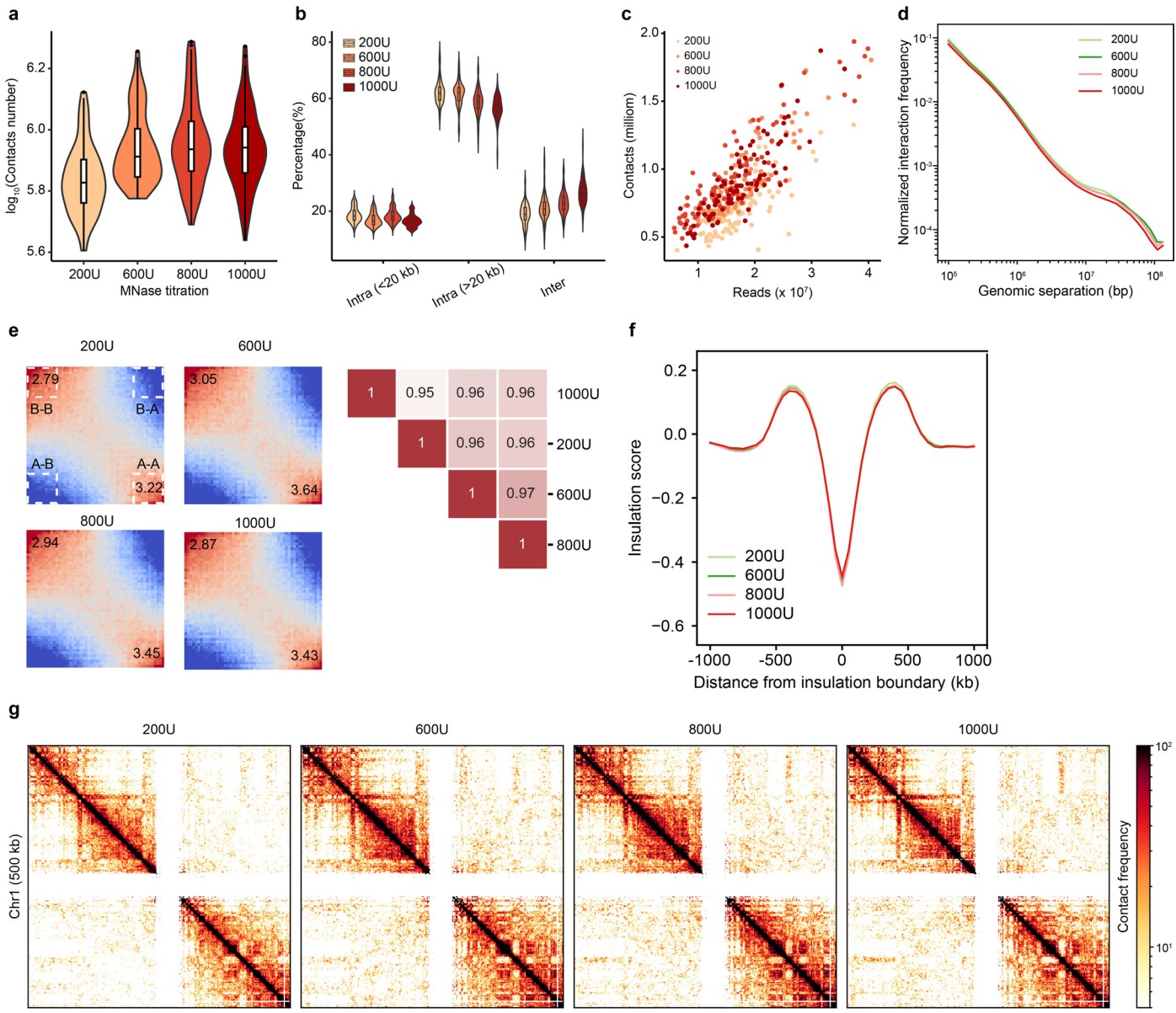

**Extended Data Fig. 2 | Comparison of MNase titration. a,** Violin plot compares the number of contacts for MNase titration groups (200U: n = 90, 600U: n = 92, 800U: n = 92, 1000U: n = 91). The box middle line represents the median value, box limits the 25% and 75% quantiles, and the whiskers show the minimum and maximum. **b,** Violin plot compares the composition of contacts between MNase titration group. The box middle line represents the median value, box limits the 25% and 75% quantiles, and the whiskers show the minimum and maximum.

**c,** Scatter plot depicts the relationship between number of reads and number of unique contacts. **d,** Contact probability on genomic separation. **e,** Left: saddle plot shows the compartment strength for four MNase titration groups. Right: Pearson correlation of the first eigenvalues at 100 kb resolution. **f,** Insulation scores at insulation boundary. **g,** Ensemble contact maps of four MNase titration groups at 500 kb resolution for chromosome 1.

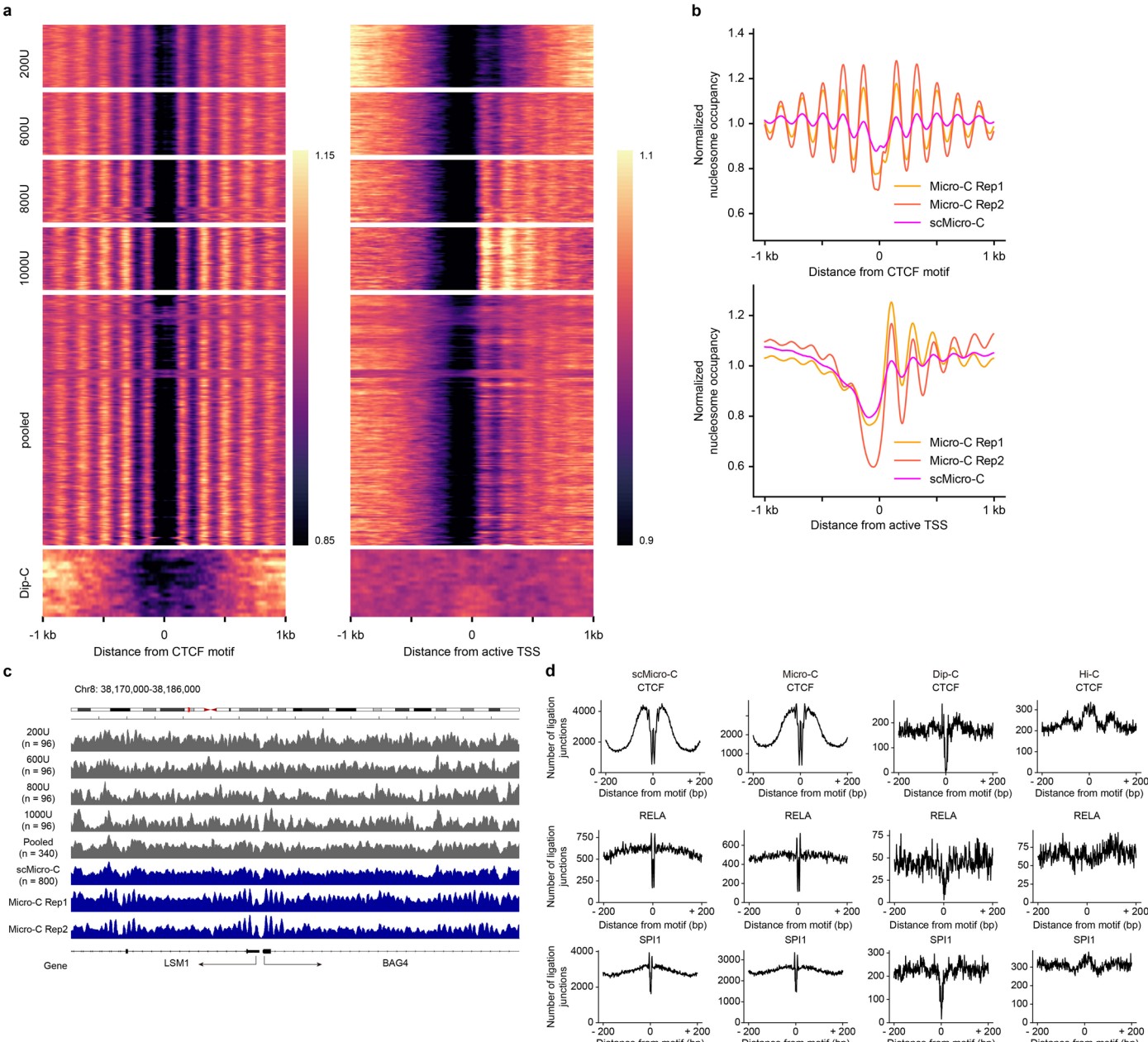

**Extended Data Fig. 3 | scMicro-C retains protein occupancy information.**
**a**, Heatmap shows the normalized nucleosome occupancy around CTCF
motif (left) and active TSS (right). Each row represents a single cell, scMicro-C
and Dip-C data were shown. **b**, Line plot shows the normalized nucleosome
occupancy around CTCF (top) and active TSS (bottom), two bulk Micro-C
replicates and ensemble scMicro-C are shown. **c**, Genomic track shows the
nucleosome occupancy around LSM1 gene locus. **d**, Line plot shows transcription
factor footprinting, calculated by detected ligation junction around
corresponding TF motif.

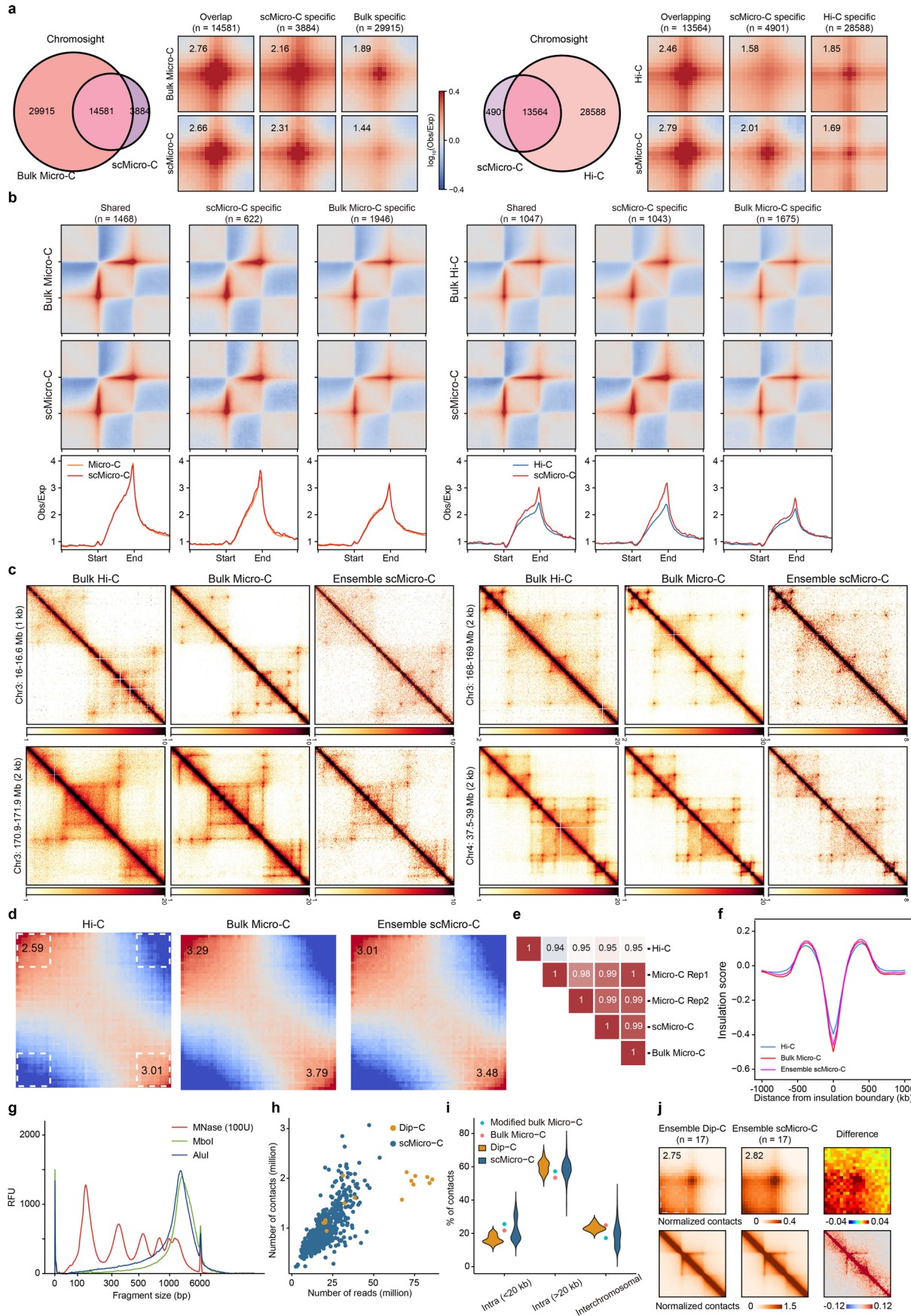

**Extended Data Fig. 4 | See next page for caption.**

**Extended Data Fig. 4 | Ensemble scMicro-C faithfully recapitulates multi-scale 3D genome organization. a**, Left: comparison of chromatin loops between ensemble scMicro-C and bulk Micro-C. Venn diagram depicting overlap of loops between bulk Micro-C and ensemble scMicro-C using chromosight and chromatin loop APA showing bulk Micro-C (top) and ensemble scMicro-C (bottom) around shared and unique loops. The same as left, compare ensemble scMicro-C and bulk in situ Hi-C. **b**, Comparison of detected chromatin stripes between scMicro-C and bulk Micro-C or bulk Hi-C. **c**, Contact maps of bulk Hi-C, bulk Micro-C and ensemble scMicro-C at selected genomic regions of 1 kb or 2 kb resolution. **d**, Saddle plots show the A/B compartmentalization strength.

**e**, Heatmap shows the Pearson correlation of A/B compartment between bulk and scMicro-C datasets. **f**, Insulation scores of bulk Hi-C, bulk Micro-C and scMicro-C. **g**, Chromatin length distribution depicts the difference between MNase digestion and restriction enzymes. **h**, Scatter plot demonstrates the relationship between scMicro-C (n = 744) and Dip-C (n = 17). **i**, Violin plot compares the composition of chromatin contacts between scMicro-C (n = 744) and Dip-C (n = 17), accompanied by original bulk Micro-C and modified bulk Micro-C, only ≥ 1 kb contacts were counted. **j**, Chromatin loops pile-up plot for Dip-C (left), scMicro-C (middle) and their difference.

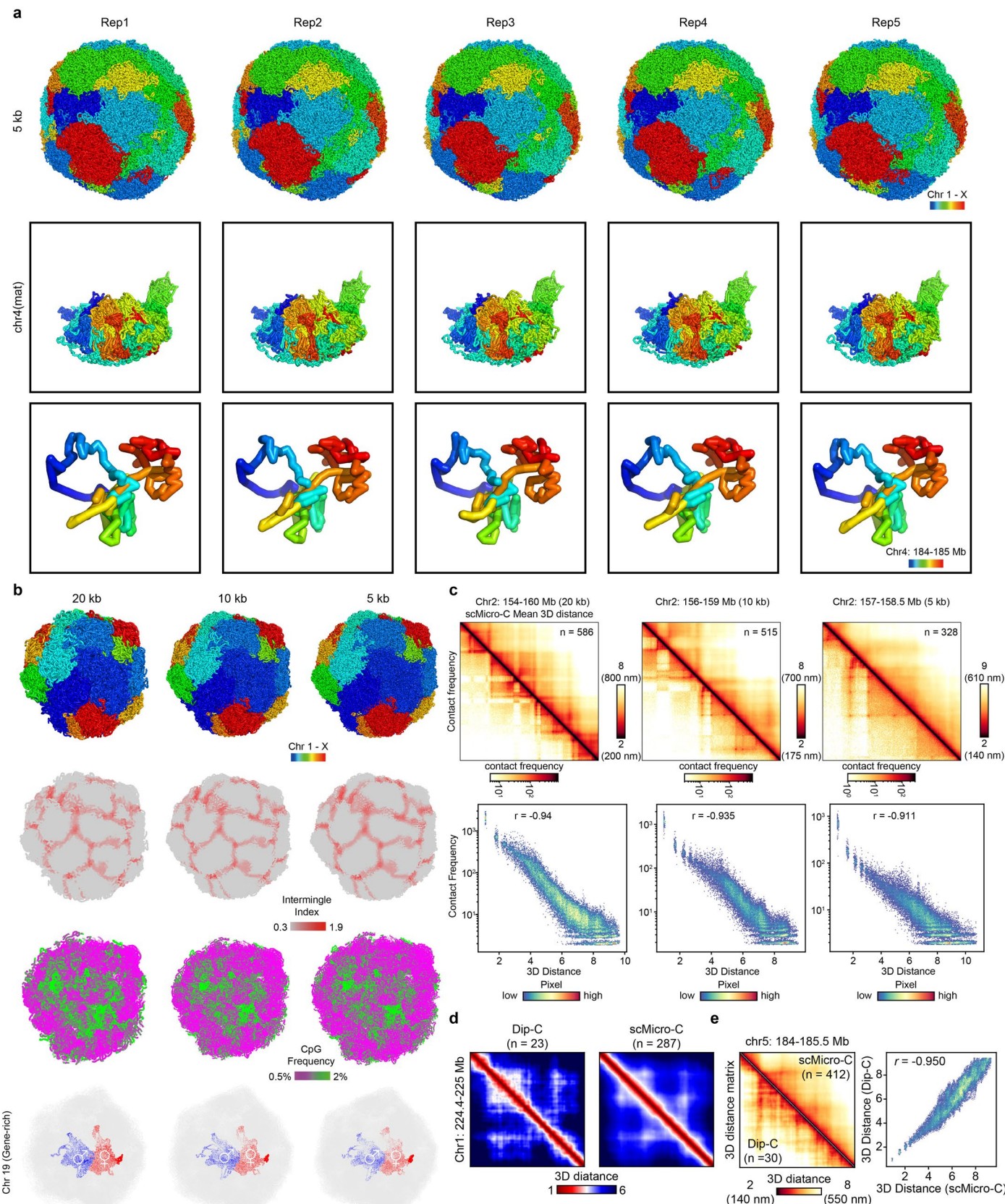

**Extended Data Fig. 5 | See next page for caption.**

**Extended Data Fig. 5 | Validation of kilobase single-cell 3D genome structure.**
**a**, 3D genome structures of a representative GM12878 cell (cell 25), 5 independent reconstructed structures are shown. **b**, 3D genome structures at different resolutions of a representative GM12878 cell (cell 25), chromosome territory, chromosome intermingling, A/B compartments and selected chromosomes are shown. **c**, Top: mean scMicro-C 3D distance matrices (top right) and bulk Micro-C contact matrices (bottom left) of selected genomic regions at 20 kb (chr2: 154–160 Mb), 10 kb (chr2: 156–159 Mb) and 5 kb (chr2: 157–158.5 Mb). Bottom: 2D histograms show the correlation between contact frequency and mean 3D distance of the same regions and resolutions. **d**, Mean 3D distance matrix of Dip-C (bottom left, n = 23) and scMicro-C (top right, n = 287) at 5 kb resolution at nested CTCF loops region shown in Fig. 2d (chr1: 224.4–225 Mb). **e**, Left: mean 3D distance of Dip-C (bottom left, n = 30) and scMicro-C (top right, n = 412) at 5 kb resolution at chromatin region shown in Fig. 2b (Chr5: 184-185.5 Mb). Right: Scatter plot showing the correlation of 3D distance matrix between Dip-C and scMicro-C at region shown in left.

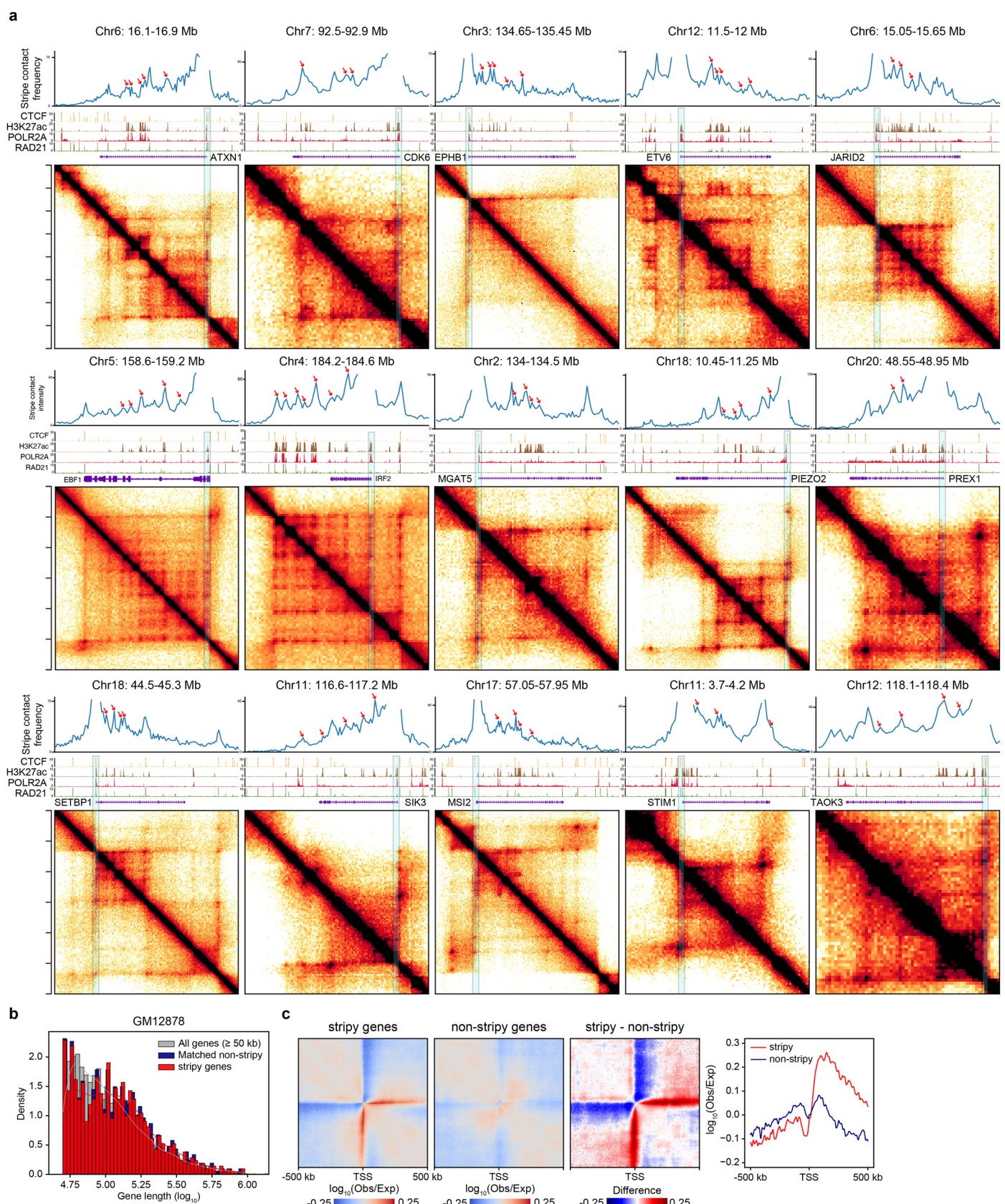

**Extended Data Fig. 6 | PES genes show focal interactions at multiple downstream enhancer loci. a**, Representative contact maps depicting 'PES genes' along with corresponding tracks for CTCF, RAD21, H3K27ac, and PLOR2A ChIP–seq data displayed above the contact map. The quantification of stripe interaction intensity was aligned at the top, with the red arrow indicating the peak of focal interaction at the enhancer sites. **b**, Gene length distribution histogram comparing all expressed genes ≥50 kb, PES genes, and length-matched non-PES genes in GM12878 cells. **c**, Left: pile-up analysis comparing PES genes and non-PES genes centered at the TSS without rescaling of GM12878; right: the difference between stripy and non-stripy and quantification at the TSS.

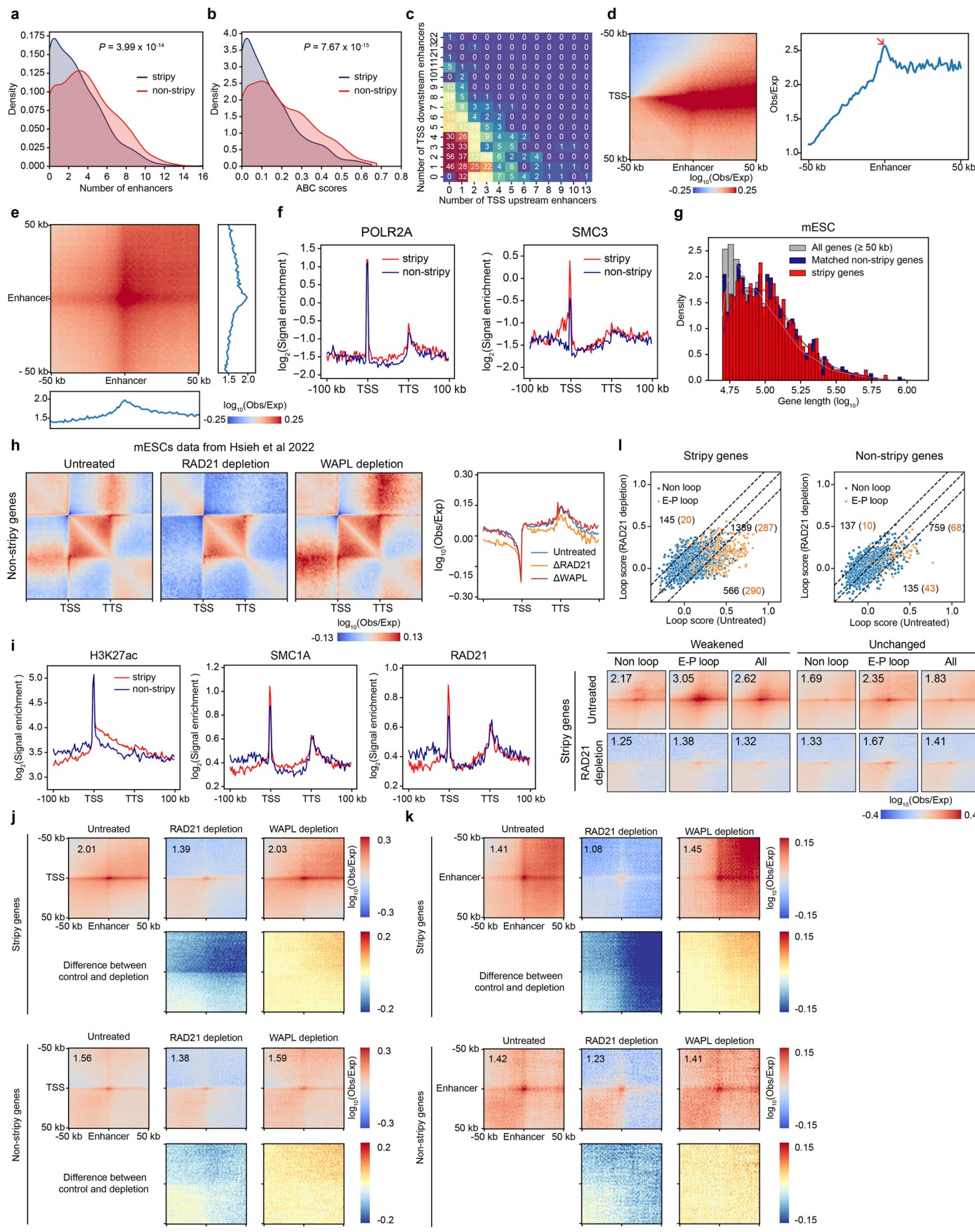

**Extended Data Fig. 7 | See next page for caption.**

**Extended Data Fig. 7 | 'Promoter–enhancer stripe' links the promoter of PES genes to multiple downstream enhancers. a**, Density plots illustrating the number of enhancers predicted by the ABC model for PES genes versus non-PES genes of GM12878, two-sided Mann–Whitney U test was used. **b**, Density plots displaying the ABC scores of predicted enhancers for PES genes compared to non-PES genes of GM12878, with statistical analysis conducted using the two-sided Mann–Whitney U test. **c**, Heatmap visualizing the distribution of ABC model-predicted enhancers associated with PES genes of GM12878. Among PES genes, 66.2% have more downstream enhancers than upstream, 27.1% have more upstream enhancers than downstream, and 6.7% have equal downstream and upstream enhancers. **d**, Left: APA results between the transcription start sites (TSS) and downstream enhancer loci for PES genes (n = 3,116 E–P pairs); Right: quantification of the pile-up intensity, with the red arrow highlighting the peak at the enhancer site. **e**, APA result between pairwise enhancers of the same stripy gene (n = 16,614 E–E pairs). All enhancers (marked by H3K27ac) were used. **f**, Metagene plots showing the signal of PLOR2A and the cohesin subunit SMC3 at PES genes versus non-PES genes in GM12878 cells. **g**, Similar analyses as in **a**

performed for mESC. **h**, Pile-up results comparing non-PES genes (n = 600) in untreated, RAD21-depleted, and WAPL-depleted bulk Micro-C data from mESC. **i**, Metagene plots illustrating the distribution of H3K27ac, POLR2A, SMC1A, and RAD21 at 'PES genes' versus non-PES genes in mESC. **j**, APAs between the TSS and multiple gene body enhancers of PES genes (top, n = 2,260) and non-PES genes (bottom, n = 1,109), plotted for untreated, RAD21-depleted and WAPL-depleted mESC Micro-C. **k**, APAs between pairwise gene body enhancers of PES genes (top, n = 9,510) and non-PES genes (bottom, n = 3063), plotted for untreated, RAD21-depleted and WAPL-depleted mESC Micro-C. **l**, Top: scatter plots of loop scores plotted for all E–P interactions between gene body enhancers and promoter of stripy (left) and non-PES genes (right) in the untreated and Rad21-depleted cells. Detected E–P loops were labeled as orange cross, non-loop E-P interactions were labeled as blue dot. The E–P interactions were grouped as strengthened, unchanged and weakened. Bottom: pile-up of loop and non-loop E–P interactions of PES genes for weakened (left) and unchanged (right) interaction upon cohesin depletion.

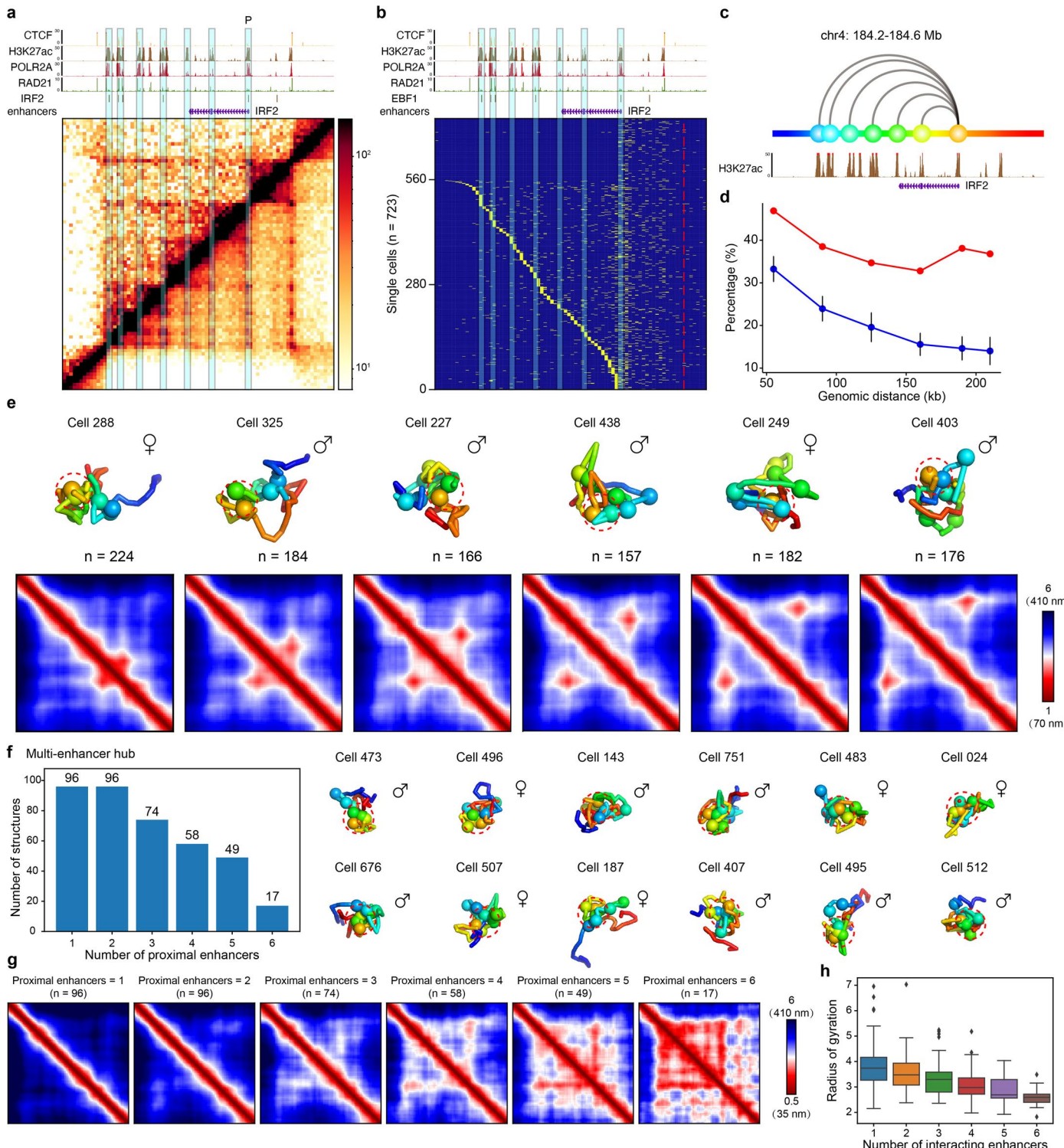

**Extended Data Fig. 8 | Single-cell 3D genome structure analysis of *IRF2* gene locus. a**, Contact map (top left is ensemble scMicro-C, bottom right is bulk Micro-C) at *IRF2* gene locus along with CTCF, H3K27ac, POLR2A and RAD21 ChIP–seq tracks shown above, the ABC model-predicted IRF2 enhancers were indicated. **b**, Sorted single-cell contact profiles of *IRF2* gene stripe. **c**, Cartoon depicting the genomic loci selected for single-cell 3D genome analysis. **d**, Line plot illustrating the percentage of single-cell 3D chromatin structures with the proximity of corresponding E–P pairs. Six non-stripe regions were taken as control. The whiskers of control regions represent standard deviation. **e**, Top: representative structures forming corresponding E–P loops. Bottom: mean 3D distance matrices of single-cell 3D genome structures forming corresponding E–P loops. **f**, Left: histogram summarizing single-cell 3D genome structures

forming multi-enhancer hubs; Right: representative single-cell 3D structures forming multi-enhancer hubs, the hub was indicated with red circle. **g**, Mean 3D distance matrices of single-cell 3D genome structures forming multi-enhancer hubs with indicated number of enhancers. **h**, Boxplot illustrating the radius of gyration of individual single-cell 3D structures forming multi-enhancer hubs containing 1 to 6 enhancers. Pairwise statistical tests were conducted, revealing statistical significance ($P < 0.05$) in all comparisons except for the group4-group5 comparison. The analysis used a two-sided Wilcoxon rank sum exact test. Within each boxplot, the centerline represents the median value, while the box limits indicate the 25th and 75th percentiles. The whiskers extend to the maximum and minimum values observed in the data distribution.

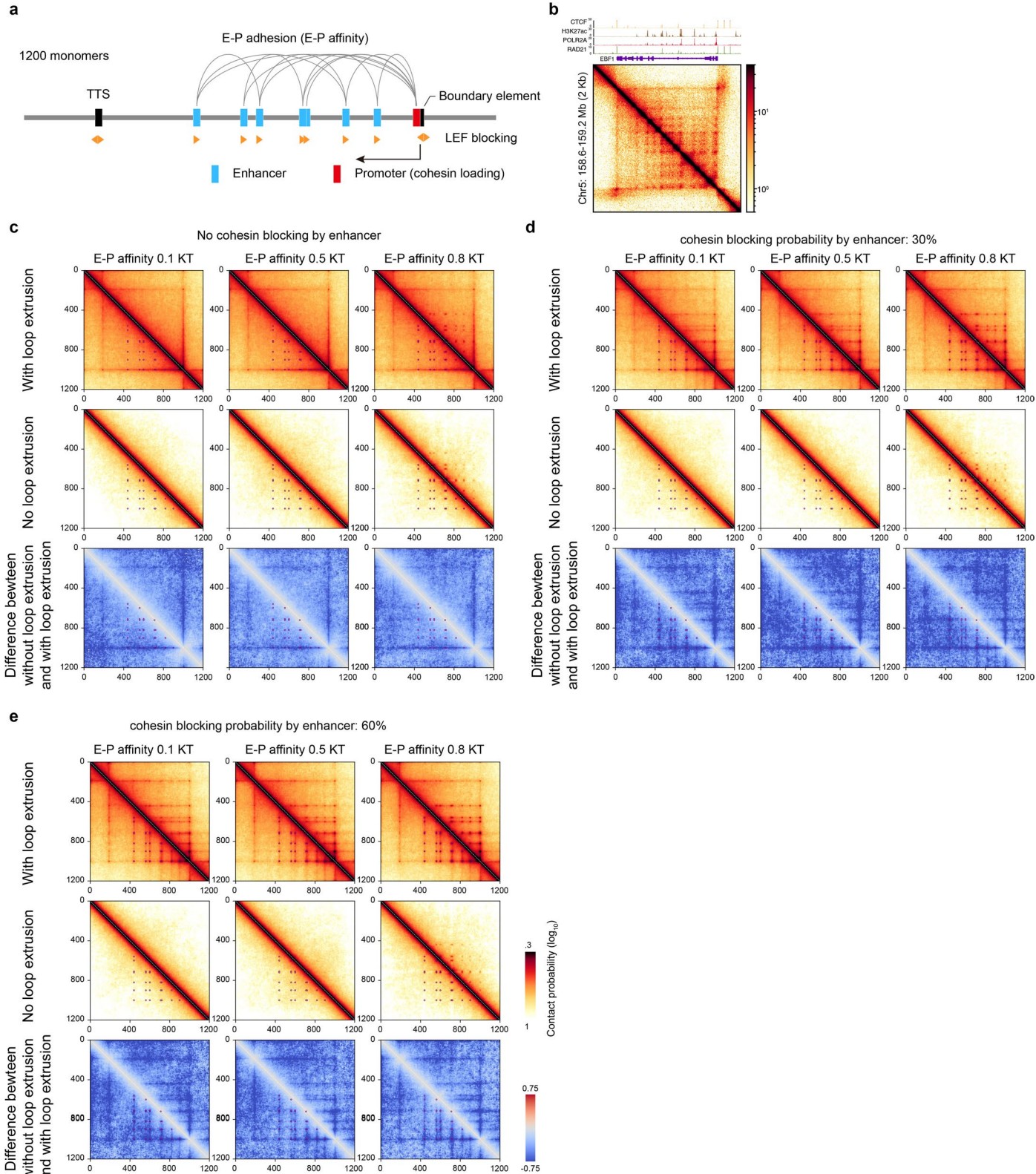

**Extended Data Fig. 9 | Polymer simulations of the gene stripe and enhancer interaction network at EBF1 locus. a**, Schematic representation of the EBF1 locus used in polymer simulations. The TSS and TSS region are modeled as bi-directional cohesin barriers, while enhancers are modeled as uni-directional cohesin barriers with tunable strength. Arc shows the E–P and E–E interactions with adjustable affinities. Detailed simulation settings are provided in the Methods section. **b**, Chromatin contact map of the EBF1 locus from GM12878 bulk Micro-C at 2 kb resolution. **c–e**, Simulated contact matrices under different enhancer blocking strength: no blocking (**c**), 30% blocking probability (**d**) and 60% blocking probability (**e**). For each blocking condition, E–P and E–E affinities are varied: 0.1 KT (left), 0.5 KT (middle), and 0.8 KT (right). Each panel shows results with cohesin extrusion (top), without cohesin extrusion (middle), and the difference between the two conditions (bottom).

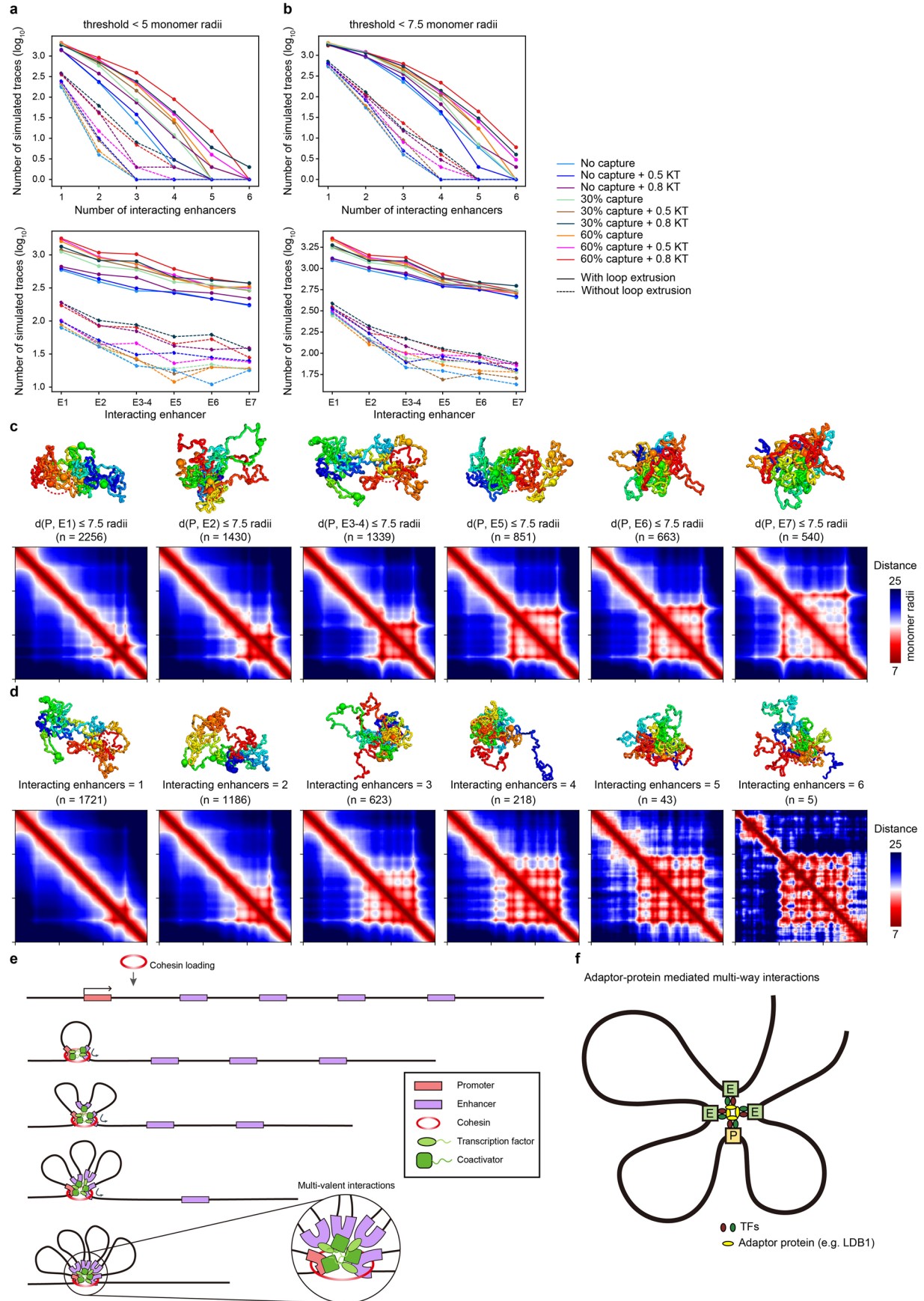

**Extended Data Fig. 10 | See next page for caption.**

**Extended Data Fig. 10 | Polymer simulations recapitulate multi-enhancer hub structures. a**, Top: line plots summarize the number of simulated structures forming multi-enhancer hub under a stringent threshold of 5 monomer radii with different simulation parameters. Botton: line plots illustrate the number of simulated structures forming E–P loops between EBF1 promoter and enhancers E1–E7, respectively. **b**, The same analysis as in **a**, but with a less stringent threshold of 7.5 monomer radii. **c**, Top: representative simulated 3D structures. Bottom: mean distance matrix of simulated structures forming E–P loops with enhancers E1 to E7, arranged from left to right. **d**, Top: representative simulated 3D structures. Bottom: mean distance matrix of simulated structures with 1 to 6 interacting enhancers. Results of **c** and **d** are based on parameter: cohesin blocking probability of enhancer is 60%, E–P and E–E affinities set to 0.8 KT. **e**, Cohesin-medaited loops extrusion brings multiple downstream enhancers and promoter into contact and is further stabilized by molecular affinities among transcriptional proteins bound at these elements. **f**, Adapter protein LDB1-mediated multi-enhancer hub interactions. LDB1 binds to LIM domain-containing TFs, then undergoes oligomerization through self-association domain.

# Reporting Summary

## Statistics

For all statistical analyses, confirm that the following items are present in the figure legend, table legend, main text, or Methods section.

| n/a | Confirmed | |
|---|---|---|
| ☐ | ☒ | The exact sample size (*n*) for each experimental group/condition, given as a discrete number and unit of measurement |
| ☐ | ☒ | A statement on whether measurements were taken from distinct samples or whether the same sample was measured repeatedly |
| ☐ | ☒ | The statistical test(s) used AND whether they are one- or two-sided *Only common tests should be described solely by name; describe more complex techniques in the Methods section.* |
| ☒ | ☐ | A description of all covariates tested |
| ☒ | ☐ | A description of any assumptions or corrections, such as tests of normality and adjustment for multiple comparisons |
| ☐ | ☒ | A full description of the statistical parameters including central tendency (e.g. means) or other basic estimates (e.g. regression coefficient) AND variation (e.g. standard deviation) or associated estimates of uncertainty (e.g. confidence intervals) |
| ☐ | ☒ | For null hypothesis testing, the test statistic (e.g. *F*, *t*, *r*) with confidence intervals, effect sizes, degrees of freedom and *P* value noted *Give P values as exact values whenever suitable.* |
| ☒ | ☐ | For Bayesian analysis, information on the choice of priors and Markov chain Monte Carlo settings |
| ☒ | ☐ | For hierarchical and complex designs, identification of the appropriate level for tests and full reporting of outcomes |
| ☒ | ☐ | Estimates of effect sizes (e.g. Cohen's *d*, Pearson's *r*), indicating how they were calculated |

*Our web collection on statistics for biologists contains articles on many of the points above.*

## Software and code

Policy information about availability of computer code

| Data collection | FACSDiva. |
|---|---|
| Data analysis | The code used in this study is publicly available at GitHub (https://github.com/tanlongzhi/dip-c and https://github.com/lh3/ hickit). BWA-MEM (v0.7.17), cooler (v.0.8.11), Juicer (v1.19.02), cooltools (v0.5.2), deepTools (v3.5.2), chromosight (v1.6.2), coolpuppy (v1.0.0), Stripenn (v1.1.65.7), StripeCaller (v0.1.0), DANPOS2 (v2.2.2) |

For manuscripts utilizing custom algorithms or software that are central to the research but not yet described in published literature, software must be made available to editors and reviewers. We strongly encourage code deposition in a community repository (e.g. GitHub). See the Nature Portfolio guidelines for submitting code & software for further information.

## Data

Policy information about availability of data

All manuscripts must include a data availability statement. This statement should provide the following information, where applicable:

- Accession codes, unique identifiers, or web links for publicly available datasets
- A description of any restrictions on data availability
- For clinical datasets or third party data, please ensure that the statement adheres to our policy

Raw next generation sequencing data were deposited on Sequence Read Archive (SRA; https://www.ncbi.nlm.nih.gov/sra) with accession number PRJNA78. All processed data were deposited on the Gene Expression Omnibus (GEO; http://www.ncbi.nlm.nih.gov/

geo/) with accession number GSE281150 (Superseries). The subseries includes GSE225201 for bulk Micro-C, GSE278191 for scMicro-C 1000U, GSE279479 for scMicro-C 600U, GSE279583 for scMicro-C 800U, GSE279732 for scMicro-C 200U, GSE280482 for SDS test, GSE192759 for scMicro-C pool. Data were mapped to genome assembly GRCh38.

# Research involving human participants, their data, or biological material

Policy information about studies with human participants or human data. See also policy information about sex, gender (identity/presentation), and sexual orientation and race, ethnicity and racism.

| | |
|---|---|
| Reporting on sex and gender | Not applicable. There is no human participants involved in this study. |
| Reporting on race, ethnicity, or other socially relevant groupings | Not applicable. |
| Population characteristics | Not applicable. |
| Recruitment | Not applicable. |
| Ethics oversight | Not applicable. |

Note that full information on the approval of the study protocol must also be provided in the manuscript.

# Field-specific reporting

Please select the one below that is the best fit for your research. If you are not sure, read the appropriate sections before making your selection.

☒ Life sciences        ☐ Behavioural & social sciences        ☐ Ecological, evolutionary & environmental sciences

For a reference copy of the document with all sections, see nature.com/documents/nr-reporting-summary-flat.pdf

# Life sciences study design

All studies must disclose on these points even when the disclosure is negative.

| | |
|---|---|
| Sample size | No statistical method was used to predetermine sample size. For single-cell Micro-C data, there were 810 cells sequenced. For bulk Micro-C, there were at least two replicates were sequenced for each condition. No sample size estimation is needed. We |
| Data exclusions | 52 scMicro-C cells were excluded due to low quality. |
| Replication | We have extensively demonstrated the robustness and reproducibility of our methodology. There were at least two independent biological replicates were performed for all bulk sequencing samples (bulk Micro-C). For single-cell data, five batches generated. All replicates were successfully amplified and sequenced. |
| Randomization | Not applicable. There is no treatment/perturbation groups in this study. |
| Blinding | Not applicable. Blinding was not required since our sample is taken from normal cultured cell line, there is no treatment or perturbation. |

# Reporting for specific materials, systems and methods

We require information from authors about some types of materials, experimental systems and methods used in many studies. Here, indicate whether each material, system or method listed is relevant to your study. If you are not sure if a list item applies to your research, read the appropriate section before selecting a response.

## Materials & experimental systems

| n/a | Involved in the study |
|---|---|
| ☒ | ☐ Antibodies |
| ☐ | ☒ Eukaryotic cell lines |
| ☒ | ☐ Palaeontology and archaeology |
| ☒ | ☐ Animals and other organisms |
| ☒ | ☐ Clinical data |
| ☒ | ☐ Dual use research of concern |
| ☒ | ☐ Plants |

## Methods

| n/a | Involved in the study |
|---|---|
| ☒ | ☐ ChIP-seq |
| ☐ | ☒ Flow cytometry |
| ☒ | ☐ MRI-based neuroimaging |

# Eukaryotic cell lines

Policy information about cell lines and Sex and Gender in Research

| | |
|---|---|
| Cell line source(s) | GM12878 (Coriell Institute) is a EBV-transformed B lymphocyte from a female. |
| Authentication | All cell lines were validated with morphology and gene expression and other epigenetic states with published datasets. |
| Mycoplasma contamination | Mycoplasma contamination test is negative. |
| Commonly misidentified lines (See ICLAC register) | No commonly misidentified cell lines were used in the study. |

# Plants

| | |
|---|---|
| Seed stocks | No plant material was used in this study. |
| Novel plant genotypes | Not applicable. |
| Authentication | Not applicable. |

# Flow Cytometry

## Plots

Confirm that:

☒ The axis labels state the marker and fluorochrome used (e.g. CD4-FITC).

☒ The axis scales are clearly visible. Include numbers along axes only for bottom left plot of group (a 'group' is an analysis of identical markers).

☒ All plots are contour plots with outliers or pseudocolor plots.

☒ A numerical value for number of cells or percentage (with statistics) is provided.

## Methodology

| | |
|---|---|
| Sample preparation | GM12878 cells followed Micro-C procedure were filtered with 30 um cell strainer, then stained with 7-AAD to distinguish nuclei from debris. |
| Instrument | BD, FACS Aria SORP |
| Software | BD FACSDiva v9.0 Software |
| Cell population abundance | All nuclei were sorted without biased, the 7-AAD-negative nuclei was selected. |
| Gating strategy | Nuclei were distinguished from debris based on FSC-A and SSC-A, then the multiplets were removed by two step gating of FSC-W and FSC-H, SSC-W and SSC-H. Then nuclei were selected based on PerCP-cy5-5-A. |

☒ Tick this box to confirm that a figure exemplifying the gating strategy is provided in the Supplementary Information.

