## [Peer Review File · Nature Genetics]

Single-cell Micro-C profiles 3D genome structures at high resolution and characterizes multi-enhancer hubs

Corresponding Author: Professor Xiaoliang Xie

Version 0:

Decision Letter:

31st Oct 2023

Dear Prof. Xie,

Your Technical Report, entitled "Extruding transcription elongation loops observed in high-resolution single-cell 3D genomes", has now been seen by 4 referees. You will see from their comments below that while they find your work of interest, some important points are raised. We are interested in the possibility of publishing your study in Nature Genetics, but would like to consider your response to these concerns in the form of a revised manuscript before we make a final decision on publication.

Reviewer #1 thinks that overall this is technically sound work, and that the methodology is impressive. However, they have serious concerns about the TOPO inhibitors. Unless you can show that there's no DNA damage, it might be better simply to remove these data.

Reviewer #2 says that the method and the data are impressive. However, they strongly disagree with certain claims, e.g. about bursting, which should either be removed or substantially toned down.

Reviewer #3 finds scMicro-C exciting as a new method but feels that you should focus more on the novel technical aspects rather than on speculative biological findings or those that have been reported before.

Reviewer #4, like the other reviewers, sees scMicro-C as an important contribution to the field, but questions some of the conclusions and data interpretation.

Since this manuscript is a Technical Report, instead of an Article, we suggest you focus on technical aspects instead of developing full mechanistic stories, and also carry out comprehensive benchmarking and add more detailed methodological descriptions.

We therefore invite you to revise your manuscript taking into account all reviewer and editor comments. Please highlight all changes in the manuscript text file. At this stage we will need you to upload a copy of the manuscript in MS Word .docx or similar editable format.

We are committed to providing a fair and constructive peer-review process. Do not hesitate to contact me if there are specific requests from the reviewers that you believe are technically impossible or unlikely to yield a meaningful outcome. I'd be happy to discuss the reviewers' comments in detail with you.

*2) If you have not done so already please begin to revise your manuscript so that it conforms to our Technical Report format instructions, available

[here](http://www.nature.com/ng/authors/article_types/index.html).

*3) Include a revised version of any required Reporting Summary: <https://www.nature.com/documents/nr-reporting-summary.pdf>

It will be available to referees (and, potentially, statisticians) to aid in their evaluation if the manuscript goes back for peer

review.

Link Redacted

We hope to receive your revised manuscript within 3 months. If you cannot send it within this time, please let us know.

Sincerely,

Tiago

Tiago Faial, PhD
Chief Editor
Nature Genetics
<https://orcid.org/0000-0003-0864-1200>

Reviewers' Comments:

Reviewer #1:

Remarks to the Author:

In this manuscript, Wu and colleagues present single-cell Micro-C data to investigate high resolution 3D genome structures and how they form. The scMicro-C data themselves are of high quality (please see my comments on usage below), and the analysis is in most parts technically thorough. The key question addressed here, as formulated by the authors is: "we set out to investigate whether cohesin-mediated loop extrusion facilitates transcription" and this is also pointed out in the concluding sentence of their abstract. However, I would argue that topoisomerases (with all the caveats discussed below) and RNAPs seem to have an equal, if not a more important, role here. Despite its technological advance, which not at all small, the manuscript does suffer in some aspects, and I provide below my thoughts on these that I hope the authors will find useful.

KEY REMARKS:

- As the authors themselves state, they noted that early investigations uncovered related gene-associated structures, including gene-body-associated domains and gene domains, which manifest as squares on chromatin contact maps at actively transcribed genes. However, no distinct "line" structure was discernible at the TSS, probably due to the limited resolution provided by Hi-C." This is indeed true, and although the higher resolution does help make these structures crisper, I am still a bit skeptical as to whether high-resolution TELs are now attributed to different mechanistic origins than those inferred previously (please also see my comments below). Moreover, although the bulk of "stripes" are not TELs, and when averaged out "stripes" do not collectively behave as TELs, in multiple instances in this work (see the examples in Extended Data Fig. 6, where the two clearly mix in multiple loci) and in previous work by others having used Micro-C, TELs were clearly a subset of stripes. And I feel that this needs to be pointed out in the main text.

- The discussion of the data on TELs following RNAPII depletion is very confusing to me. The authors show an (expected in my opinion) dependency of TELs on RNAPII, which, must be noted, appears less strong than that of Rad21 depletion. This is left without any discussion, as is the fact that RNAPII depletion leads to a reduction of cohesin occupancy (shown recently by Zhang et al., Sci Adv 2021; Zhang et al., Nat Genet 2023). It would be good to see how this relationship between the cohesin effects, the RNAPII effects (and later also the TOPO effects) are reconciled by the authors mechanistically, and how

their relative contributions or hierarchies are dissected. Right now, this remains unaddressed.

- Then, the authors move to R-loops, to build an assumption that is not supported by any data (that I can find at least). they do not map R-loops to the TSSs of TELs in their lines, they do not perform experiments with RNase H overexpression to assess potential changes in TEL structure and emergence, nor do they correlate high RNAPII occupancy with R-loops (which is a general trend found elsewhere). Also, I do not see how this at all relates with the reduced cohesin occupancy caused by RNAPII depletion, how this is relevant to TELs themselves, etc. I feel that this part of the manuscript is largely irrelevant or that its needs significantly more work to merit inclusion (mostly as a cartoon) and a place in a main figure.

- I think it is important to note here that TELs seems to be equally affected by FLV and RNAPII depletion, which is on the one hand expected (one would intuitively think that TELs rely on elongating RNAPs), but on the other hand surprising, as complete removal of RNAPII should fully abolish TELs. Indeed, chromatin-bound PolII cannot be so efficiently degraded in these degree systems, and the authors don't seem to degrade more than 70% or so of Ser2/5-PolII (which in our hands is not enough to see very strong 3D-genome changes and comparable to a prolonged TRP treatment). This is due to the 4-hour time point used, and it would be nice to employ a longer time point and get depletion to >85% to provide a more precise picture to readers.

- The next part on topoisomerase and TELs, is potentially very interesting, but there are a number of concerns associated with it. First, although the premise of the hypothesis is really interesting, the means by which the authors have chosen to interfere with TOP1 and TOP2 activity can prove very problematic. Both TPT (a drug developed on the basis of camptothecin) and Etoposide are both strong inducers of DNA damage, they do not really alleviate TOPO activity but rather lock complexes in place (see Gothe et al., Mol Cell 2019 for an exemplar of this) and certainly cause significant DNA damage (this is why they are used as anticancer drugs) even at short time frames and low doses. The authors have not controlled for this, but looking at the cell cycle analysis, there is a pronounced S-phase signal accumulation signaling cell cycle arrest (as would be expected by these drugs and by damage). To my surprise though, the numbers in the inset of the graphs in Extended Data Fig. 10a do not at all reflect the curves in the plot -- S-phase percentages seem to only increase mildly, and G1 ones not to really drop, but the structure of the data curves clearly shows something else. Moreover, the figure legend (as most figure legends in the manuscript actually) offer very little information to understand the plot. For example, the the coloured contours seems to be fitted to the squiggly line that represents the raw data. Is this true? Because the fit is not good at all. What is quantified here? Simple comparison of the overarching squiggly line disagrees with the numbers presented. I am also pretty sure that the genes unregulated by these drugs would show clear signs of cell cycle arrest and DNA damage response induction, but they do not seem to be analysed in this direction. Therefore, the proper way of addressing the question the authors are posing here is to use either siRNA-mediated depletion or, even better, TOP1/TOP2 degen lines. These pharmacological inhibitors are not fit for purpose.

- Second, even with the caveats of these inhibitors aside, the authors report effects on TELs following TOPO inhibition for 8 h. They report that "TEL signals were significantly weakened upon TOP2 inhibition, while [...] slightly strengthened upon TOP1 inhibition" and they then correlate that with "cohesin binding at gene body [being] significantly weakened upon TOP2 inhibition but not TOP1". Given the interdependency of RNAPII elongation and supercoiling resolution (e.g., see work by the Baranello lab), and the fact that TELs rely on elongating RNAPs, was this not to be expected? To advance our mechanistic understanding, one would need to map actual supercoils/topoisomerases and were these accumulate upon drug inhibition to really assess the phenomenon. Moreover, the drop in cohesin levels and "retention of cohesin at gene body" claimed is not that strong in the mean plot of Fig. 2g (and the word "significantly" implies some sort of statistical evaluation that I cannot locate in the text), but also does not explain the strengthening of upon TOP1 interference if cohesin remains unchanged. But, what happens to RNAPII in either case is not investigated and therefore the cartoon in panel 2h remains purely speculative.

- Now, concerning the single-cell conformation analyses, I remain a bit ambiguous. On the one hand, it is clear to this reviewer that scMicro-C surpasses Dip-C and is probably the best tool to date. At the same time though, there are three issues that need to be addressed in my opinion. First, it is the resolution issue. The central claim of "improving the spatial resolution of single-cell 3D genome mapping to 5 kb" (already highlighted in the manuscript's summary), can be misleading. How is resolution defined in such a sparse dataset? How is effective resolution defined? I assume one could bin Dip-C data to 5-kbp resolution and would still see sparse signal. How can this signal be interpreted on its own. It seems to me that the authors (a) rely on the bulk signal to identify looping interactions and TELs and then map these back onto single-cell structures, usually on the basis of single in 1 or 2 bins at best (see examples in Extended Data Fig. 4), and (b) rely on the distance modelling to infer conformations. None of these two are per se wrong, of course, but they do substantially moderate the claim of bona fide single cell-resolution observations. My reading of this work is that it relies on high-quality bulk data observations, and that the single-cell component is rather a derivation. Especially, the (otherwise very nice) structures deduced from a TEL cannot be claimed to represent the gradual elongation of the loop, as these snapshots can only be temporally ordered artificially and cannot be related with the respective positions of RNAP and/or cohesin in each cell. Therefore, I would urge the authors to tone-down these claims, and to also remove or rephrase the single-cell bit of their title accordingly.

- The changes in bursting inference is a welcome addition to the paper, but (a) simple modelling of scRNA-seq data does not suffice to substantiate the claim and orthogonal real-time data is needed, and (b) the dependence of bursting on cohesin has been thoroughly demonstrated before (e.g., Luppino et al. Nat Genet 2020, Robles-Rebollo et al., Nat Commun 2022) and does not seem to be confined to TELs, even in this work. So, I would suggest to turn this part of the manuscript into a Supplemental Figure.

SECONDARY REMARKS:

- I see that the authors explicitly avoid "overdigestion" with MNase and therefore have picture of mono-, di-, ori-nucleosomes,

etc. The "classical" Micro-C protocols urge users to indeed digest down to mono-nucleosomes. How do you think the high-order nucleosome stretches affect your data? Can you decipher and look at them separately? What can we learn from it?

- Ext. Data Fig. 1c: showing 500-kbp-resolution maps from whole chr6 does not allow assessment of the true scMicro-C coverage, which, although impressive compared to the single-cell 3D assays, is not in itself devoid of very high sparsity (see Fig. 3a). Therefore, please replace with zoom-ins of the level of information claimed by the manuscript's summary (i.e. 5 kbp) from different chromosomes. The same should be done at different instances in the main Figs.
- Ext. Data Fig. 1f,g should include data from the "standard" and from the authors' modified bulk Micro-C for comparison.
- Ext. Data Fig. 1e: the authors claim an average of 1M contacts per cell, but the histogram shows a red line at the 0.8M bin. Which is true?
- The GRO-seq axis labels in Fig. 1 seem to have moved out of place.
- The statement that "Lastly, in the absence of CTCF and cohesin unloading factor WAPL, cohesin accumulates at the 3' end of some active genes, implying that cohesin tends to unload near TTSs" is probably not very accurate, since this simply reflects where cohesin will end up given the forces exerted on it and around it on the finer (e.g., transcription, replication, diffusion). I would rephrase or the authors can test this by combining WAPL depletion with RNAPII depletion to assess each effect's weight on this effect.
- RNAPII is referred to as a "barrier to cohesin" in many places in the manuscript by citing the Bannigan et al. work, whereas direct evidence to this was recently provided by Zhang et al., Nat Genet 2023. I would probably cite both papers accordingly.
- I am sorry if I missed this, but are all cells to which scMicro-C was performed synchronised and in the G1-phase? If not, how can cell cycle effects be ruled out?
- The statement "Conversely, an analysis of a similarly sized genomic region without TELs or stripe structures (chr2: 114.25-115.15 Mb) reveals a decrease in the number of structures with a larger separation of genomic distance (Extended Data Fig. 13g,i)" is very unclear to me and it would be helpful if the authors explained what this region is, why it was selected, and how this conclusion was reached. However, I would have expected that the control in this case would be an equisized and equally well transcribed but TEL-free region of the genome.
- I feel that the text would benefit from more detailed explanations in most parts, especially in the single-cell section, and that figure legends suffer rather than benefit from their brevity.

A. Papantonis

Reviewer #2:

Remarks to the Author:

This paper reports the development of a single-cell version of Micro-C and transcriptional elongation loops (TELs). scMicro-C seems impressive from a technical perspective. Getting 1 million unique contacts per cell is impressive. By re-analyzing Micro-C from Hsieh 2022 they show that TELs are cohesin dependent and by performing RNA Pol II depletion micro-C they also show that TELs are RNA POL II dependent. Furthermore, experiments +/- TOP1 and TOP2 inhibition show the involvement of topoisomerases.

Although the authors only gain modest value from the scMicro-C data (in the sense that most of their conclusions could have been made with bulk Micro-C), overall I think the combined contributions of a highly efficient scMicro-C protocol and interesting insights into TELs together make for a nice contribution to the literature, that I think will be of significant interest to the broad readership of Nature Genetics if the authors can address the major and minor concerns listed below.

MAJOR COMMENTS

BURSTING: I know there are papers out there claiming to quantify transcriptional bursting from single-cell RNA-Seq but I believe those papers are wrong for the following reasons:

- 1) Cells contain multiple allele (especially after replication) – thus the statistics will need to reflect the number of gene copies which the model does not
- 2) The authors fail to distinguish intrinsic vs. extrinsic noise
- 3) scRNA-Seq has low detection efficiency, if you cannot measure single RNAs reliably, you cannot quantify the distribution
- 4) transcriptional bursting is a dynamic property about the lifetime of bursts – you can measure dynamic properties using snapshot methods.

I think the authors should remove all bursting claims. If they want to make claims about transcriptional bursting, they should demonstrate their claims using live cell imaging of nascent transcription such as MS2 and PP7 imaging demonstrated in e.g. Wan...Larson Cell 2021.

LOW-RESOLUTION MICRO-C: It appears from Fig. 1b that their Micro-C has lower resolution than Hi-C using 4-cutters because the ligation products seem to be around 1kb in size. The authors should comment on this limitation in the main text. The authors should also provide an scMicro-C protocol with their paper.

TEL: around lines 334-340 the authors claim that TELs are novel. This is not fair – this concept has been known and discussed for decades. This paper is still a nice contribution, but it is not fair to prior work to describe things the way they do here. They should cite earlier studies on TELs as well.

PROVE: at many points in the abstract and text that author use the words "proof" and "prove" – given the caveats associated with experiments, I think this is too much and not reasonable. I would like to ask the authors to use words like show or demonstrate instead.

MINOR COMMENTS

The authors use the word Resolution several time – can they precisely define what they mean by this term?

Line 121 refers to an old NIPBL-ChIP-Seq study, but the recent paper from Banigan showed that many NIPBL ChIP-Seq papers suffered from bad antibodies. Please double-check this reference.

The authors refer to dynamic loops in lines 92 and 246 and other places, but they have not measured dynamics. Therefore they should cite recent papers that measured cohesin loop dynamics such as Gabriele 2022 and Mach 2022.

The authors mentioned that they did not have sufficient resolution to detect TELs at short genes. A recent very high resolution Micro-C study by Goel et al. 2023 includes cohesin depletion and transcriptional inhibition – can the authors use these data to test if TELs exist for short genes? This would help establish the generality of these TEL structures.

Reviewer #3:

Remarks to the Author:

Review of “Extruding transcription elongation loops observed in high-resolution single-cell 3D genomes”.

Transcription is a fundamental biological process, which involves a profound remodeling of chromatin/DNA structure. In this Technical Report, the authors use a new scMicro-C method to characterize Transcription Elongation Loops (TELs) in highly expressed long genes. These observations are combined with the (re)analysis of large amounts of bulk-cell studies (Micro-C, Hi-C, RNA-seq) and single-cell RNA-seq upon the perturbation of various factors involved in transcription (RNA-PolII, TOP2, Cohesin). Based on these studies, the authors conclude that TELs are the result of the interplay between the transcription, loop extrusion and topoisomerase machineries, and that TELs may be important to increase transcriptional bursting by paused RNA PolII release.

Using scMicro-C, the authors describe an intriguing pattern of chromatin interactions at transcribed long genes, which provide a promising insight into the link between chromatin/ DNA structure and transcription. A very large part of the study (Figs. 2 & 4) is dedicated to results from other types of experiments (bulk-cell and single-cell RNA-seq), with outcomes that in many cases remain correlative at best (and for some results even highly speculative).

Besides the fact that this part of the study deviates from the goals of a Technical Report, I am not convinced that it is appropriate to interpret the TEL kinetics, obtained from scMicro-C studies in normal cells, in the context of the correlative mechanistic insights obtained from the bulk-cell and single-cell RNA-seq studies. In my opinion, to make these results a better fit for a Technical Report in Nature Genetics, the authors should reduce their attention to the bulk-cell and single-cell RNA-seq data, and instead expand their analysis of the scMicro-C results, focusing on the different contact patterns at various genic and non-genic regions in the genome.

For ease of discussion, I have grouped my remarks for each figure together. Some general comments are provided at the bottom.

Figure 1.

R1.1: The development of scMicro-C is an impressive achievement, with a number of important improvements over their previously developed Dip-C method. The way that these improvements (increased resolution, increased contacts/cell) are justified is different from how the advantages of Micro-C are generally presented. As shown in Hsieh et al., Mol Cell 2020 and Krietenstein et al. Mol Cell 2020, the main advantage of Micro-C is not so much the increased resolution (commercial Hi-C kits using 4 restriction enzymes should theoretically generate much higher resolution data), but rather that the cuts are more regularly spaced. This generates data with a much-improved signal over noise, allowing reliable measurement of chromatin organization at shorter ranges: compare Fig. S2b to similar plots in Hsieh et al. and Krietenstein et al.

- The authors should modify the sections from line 43 and line 71 to better reflect this advantage of Micro-C (see Fig. S1g).

- The authors compare their modified bulk Micro-C protocol from GM12878 cells to matching bulk Hi-C data (from line 64).

Confirming previous comparisons, their modified Micro-C indeed reports more loops and stripes as compared to Hi-C. A better validation that the modifications do not compromise data quality would be to include data from conventional Micro-C as well. Although such data does not appear available for GM12878 cells, the inclusion of data from cell types that were sequenced at similar depths should be informative. Data from human cells is available in Krietenstein et al., or from the 4Dnucleome portal.

- Supplemental figures 1 and 2 are discussed out of order: Fig. S2 is discussed before Fig. S1e-g. I have not found any reference to Fig. S1c,d.

R1.2: Next, the authors discuss the visibility of TELs that cover long and active genes in bulk Micro-C data.

- Line 89: the definition of “long” is only provided in line 109. The authors should mention here what they consider long. For a better understanding, it will also help if the authors mention how many “long” genes exist, including the fraction that is active in GM12878 cells.

- An in-depth characterization of TELs and their link to long genes has not been reported. Yet, other studies beyond those mentioned from line 100 have observed similar loops: the first observation was made using 3C in yeast by the Proudfoot lab: Tan-Wong et al., Science 2012. Moreover, gene loops were highlighted by Hsieh et al., Mol Cell 2020 in their Micro-C results as well (see Fig. 4a).

- Line 97 and 112: I don't think that the references to Fig. S1f and Table S1 are correct.

Figure 2.

R2.1: After introducing scMicro-C and TELs in GM12878 cells, the manuscript switches gears to address the potential mechanisms of TEL formation. First the authors focus on the reanalysis of Rad21 and WAPL degron Micro-C data from mouse ES cells.

- Two recent studies in Nature Genetics have addressed the link between transcription, 3D genome organization and loop extrusion using Micro-C based studies (Zhang et al., 2023 and Barshad et al., 2023). Some of their insights are highly relevant for the interpretation of the results in Fig. 2 (e.g. from line 115 and line 147). The authors should critically revisit this part and, where necessary, incorporate findings from these two studies.

- I find the paragraph from line 132 difficult to follow. How does this increased signal at the TTS provide evidence about Cohesin unloading time? What is the barrier that is mentioned in line 136? Why does the increased signal upon WAPL depletion suggest nonetheless that this process is not WAPL dependent (line 144)? Instead, could the accumulation, and possibly loading of Cohesin at promoters carrying paused PolII play a role in this process (see Barshad et al., Nat Genet 2023)?

- Line 127: it should be mentioned that this result refers to Fig. 2a. Are these long highly transcribed genes, or all highly transcribed genes?

- Line 129: signals are reduced (to 1.13), but not completely eliminated. This should be rephrased.

- Line 130: these correlative results show that TEL formation requires active loop extrusion, but they do not directly confirm that TELs are formed by loop extrusion.

- For all Micro-C pile-ups (similar to Fig. 1e), it should be indicated that the tick marks represent TSS to TTS. The mention of 1 kb is not useful, because the span that is covered differs from gene to gene.

R2.2: Next, the authors generate Micro-C data in RBP1-degron cells, further complemented by reanalysis of Micro-C data upon transcriptional inhibition.

- Line 165: this result is very similar to the result reported in Zhang et al., Nat Genet 2023. Does this analysis include all active promoters, or only those of long genes?

- The section from line 167 about R-loops as barriers for loop extrusion is extremely speculative, with very little proof beyond a weak correlation between gene expression and R-loop signal (Fig. S9d,e). As it contributes little to the study, this should be removed. From line 172, it is discussed that RNAPII depletion has a less prominent impact than transcription inhibition. How is this determined? Visual comparison between Fig. 2c and S9b? If so, the authors should include a direct comparison here. Of note, the authors should keep in mind that the Auxin system is leaky, and that PolII levels in untreated RBP1-AID cells may already be considerably reduced. From line 173: the authors should abstain from making claims about degradation efficiency in different cell lines and different studies/laboratories, unless they have access to the material themselves.

R2.3: In this last part, the authors generate Micro-C data upon inhibition of Topoisomerase 1 and 2. Here they find that particularly inhibition of TOP2 reduces TELs.

- In line 218, it's mentioned that TELs are slightly strengthened upon TOP1 inhibition, yet no difference can be observed in Fig. 2f. In contrast, such a difference is observed in Fig. S11e. Is this because the DMSO signal in Fig. 2f is the average (or combination) of the result from the two rather different DMSO experiments in Fig. S11e and S11f? If yes, the authors should consider if it's correct to merge these control experiments. Conversely, this also shows that TEL signal in Micro-C experiments may be prone to experimental variation, with the Micro-C signal for downregulated genes upon TOP2 inhibition being mostly indiscernible from DMSO treatment in the TOP1 inhibition experiment. The authors should discuss this.

- Line 225: did the authors use a statistical test to determine the significance of this difference? If not, they should abstain from making this claim. Considering the rather weak reduction of Cohesin in the gene body upon TOP2 inhibition, I'm not convinced that this is sufficient to explain the rather drastic reduction of TELs at downregulated genes. The claim of cooperativity between cohesin, RNAPII and topoisomerase (line 229) remains therefore based in weak correlations. I consider it at least as likely that the effect of topoisomerase on cohesin is indirect, mediated solely by its effect on RNAPII activity at downregulated genes.

Figure 3.

In this section, the authors return to their scMicro-C data to characterize TEL kinetics in WT cells. This is potentially a very exciting application of the scMicro-C technology, but for now this has not been used up to its full potential.

- The sorted single-cell contact maps are not easy to grasp and lack annotation. I assume that these plots only indicate contacts that involve the 5kb bin that covers the TSS? This should be explicitly mentioned. Moreover, the position of the TSS and TTS should be indicated (particularly, no indication is provided in Fig. 3a). Instead, why do the authors not use a similar visualization as done for ChIA-Drop complexes (Zheng et al., Nature 2019) or Nano-C data (Chang et al., Nat Commun 2023)?

- The results are mostly discussed relative to the analysis of the MSI2 gene, where a nearly linear drop in the single-cell contact maps is observed. This drop is quite different from the EPHB1 gene, where a more gradual distance-dependent reduction is observed (Fig. S13a). Why did the authors decide to not include a single-cell contact map for the Piezo2 gene? The drop in contacts for the EPHB1 gene shares similarities with both the control locus (Fig. S13g) and resembles single-allele Nano-C data for non-genic regions as well (Chang et al., Nat Commun 2023). This raises the question if the organization at the MSI2 gene is truly as representative as it is presented, or if more diverse configurations may exist (linked to transcription level?). The characterization of TELs would be much improved if the authors would systematically analyze their scMicro-C data, for instance by providing pile-up plots of single-cell contact maps for different transcription levels, and for other non-genic regions (architectural stripes, "empty" regions). For an ideal understanding, these WT data on TEL dynamics would be compared to scMicro-C data from cells where the transcription machinery or loop extrusion is perturbed.

- The average distance matrixes resemble those that are used to visualize ORCA data (optical reconstruction of chromatin architecture; Mateo et al., Nature 2019). The authors should better explain, both in the Results and the Material & Methods sections, how these distances are determined and that this is inferred by modeling from the scMicro-C data.

- Line 241: Fig. 3c is discussed prior to Fig. 3b.

Figure 4.

In this last part, the authors continue their efforts to characterize TELs, using various scRNA-seq data to link their presence to transcriptional bursting and paused PolII.

- Line 263: the authors focus their analysis on long TOP2-downregulated genes, which display strong TEL signal (Fig. 4a). A very similar enrichment of TEL signal is observed in long TOP1-downregulated genes (Fig. S11a, S15b), yet the authors have previously discussed that these TELs are not affected upon TOP1 inhibition (Fig. 2e and S11e). The evidence that TOP2 exerts its function through TEL regulation will be much stronger if the authors can confirm that there are differences between the TOP2 and TOP1 downregulated genes.
- Line 265: the phrase “were significantly enriched in two categories: housekeeping genes and cell-type-specific genes” is confusing, because these two categories essentially encompass all genes. Instead, it appears as if the enriched GO categories are mostly non-related. Based on this result, I’m not convinced that the conclusion can be drawn that “genes with TEL structure are important for cell identity and function.” Instead, I’m wondering if these genes share the characteristic that they are long, but are otherwise not functionally enriched. To address this question, the authors should compare their results to similar GO analyses with all long genes (expressed or not) and with downregulated long genes upon TOP1 inhibition.
- Line 275: burst kinetics are not directly measured, but rather indirectly inferred. The authors should tone down this statement.
- The results in the section from line 282 raise the question how these differences relate to genes without TELs but that are nonetheless downregulated by TOP2 inhibition. In a similar vein, how are burst kinetics affected for genes with TELs upon TOP1 inhibition?
- Section from line 300: I have problems interpreting these results. The authors claim that upon Rad21 degradation, particularly long genes are enriched that are downregulated and display a reduction in burst frequency. Yet, according to Fig. S17h, around 40% of downregulated genes, independent on length, are associated with a reduced burst frequency (blue bars at left bottom). Rad21 depletion therefore may have a stronger effect on the activity of long genes that are sensitive to TOP2 inhibition, but this appears independent from burst frequency.
- Section from line 309: the conclusion that cohesin facilitates the release of RNAPII pausing, thereby ensuring high frequency bursting is based on a single correlative analysis and appears detached from the rest of the study. The authors should either remove it or analyse this aspect in more detail, for instance by reanalysis of Micro-C data upon perturbation of RNAPII pausing (Barshad et al., Nat Genet 2023).

General comments:

- The manuscript makes many switches between results from human GM12878 cells and mouse ES cells, including a number of back-and-forths between figures and supplemental figures. To facilitate the interpretation, the authors should clearly mention for each figure/figure panel if it concerns human GM12878 cells or mouse ES cells.
- Many sentences contain grammatical inconsistencies. The manuscript could benefit from improved proof-reading.

Reviewer #4:

Remarks to the Author:

In this manuscript, Wu and colleagues present a single-cell Micro-C (scMicro-C) approach. In addition, they use bulk Micro-C data to analyze patterns of genome folding at long, active genes. They find that such genes are characterized by “stripes” that span from the TSS to the TTS. They call these structures “Transcription Elongation Loops” (TELs). Using a combination of available and newly generated bulk Micro-C data, the authors show that these TELs are reduced upon depletion of Cohesin and RNA polymerase II and after treatment with inhibitors of transcription and topoisomerases. In addition, the authors use their scMicro-C data to show intermediate folding stages of TELs. Finally, the authors examine the relationship between TELs and transcription, incl. bursting and pausing.

This manuscript contains some innovative and interesting aspects. However, the rationale for and interpretation of the experiments is not always clear. In addition, some of the conclusions are overstated and not fully supported by the data.

Major comments:

1. The development of a scMicro-C protocol is of potential interest to the field. However, since the resolution is 5 kb, it is unclear whether the MNase digestion is really beneficial, as this could in principle be achieved by DpnII digestion as well, which would allow for much more efficient ligation (of cohesive ends). It would be helpful if the authors could clarify the benefits of a scMicro-C procedure over the existing scHi-C procedures. It is a bit unfortunate that the authors start the paper by introducing their scMicro-C procedure, but discover and characterize the TELs based on bulk Micro-C data. The analysis of the intermediate TEL stages in Fig. 3 does rely on single-cell data, but it seems that this could have been achieved with existing scHi-C approaches as well. The unique benefit of the presented sc-Micro-C approach is therefore not clear. The authors mention higher cost-effectiveness and signal-to-noise ratio of scMicro-C compared to scHi-C. This indeed seems to be the case and may be useful for the field, although these improvements do not seem striking to me. If this is the only benefit of scMicro-C, it would be good if the authors could clarify this.

2. The development of the scMicro-C procedure would benefit from more detailed description and benchmarking. In particular, the authors claim that the use of SDS significantly improves the ligation efficiency. However, they do not present sufficient data that allow for a direct comparison between Micro-C data generated with and without SDS. For the profiles shown in Fig. 1b, the digestion with and without SDS looks different, even though SDS is added after the digestion step. The ligation in the conditions with SDS indeed looks much better compared to the conditions without SDS, but since the

digestion profile without SDS looks over-digested compared to the digestion with SDS, this could have biased these results. In addition, the rationale for adding SDS and the potential mechanism by which the addition of a detergent would improve the ligation efficiency is not clear. It would be helpful if the authors could clarify this. Furthermore, it would be great if the authors could show a detailed direct comparison of Micro-C data with and without the addition of SDS to show that the proportion of useful ligation junctions is actually improved by SDS. The profiles shown in Fig. 1b are not that helpful, since it is not possible to discern if the longer ligation product may represent fragments that have not been digested in the first place. Finally, it seems that the authors do not biotinylate the ligation junctions. Since the authors present potentially important improvements to the current Micro-C procedure, it would be very helpful if more details could be provided. (I have understood from colleagues that they have tested addition of SDS after reading the pre-print of this paper but could not reproduce the results presented in Fig. 1b).

3. I find the conclusions in this paper generally a bit overstated. In the abstract, the authors state that: "We proved that TELs formation results from the joint interactions between cohesin-mediated loop extrusion, RNA polymerase II (RNAPII) and topoisomerases." I do not think this is actually shown. The authors show that the TELs are weakened in absence of Cohesin. However, the reduction in TELs following RNAPII depletion are not convincing. The authors explain this by incomplete depletion, which indeed seems to be the case. However, for that reason, they cannot draw this conclusion at the moment and would need additional data to support it (i.e. longer depletion times or a better degron line). Furthermore, the topoisomerase inhibitors only seem to remove the TELs in genes that are downregulated after treatment. This indicates that this effect could (likely) be secondary to reduced transcription, which is in line with the presented transcription inhibition experiments. Similarly, the effect of topoisomerase inhibitors on Cohesin distribution may be secondary to reduced transcription (which has been shown previously in the context of RNAPII and Mediator depletions, see other comment #9). It would be helpful if the authors could describe their results and discussion more precisely and take these limitations into consideration.

4. In line with major comment #3, the strong statement about the role of Cohesin in RNAPII pausing is not at all supported by data. The authors refer to Extended Data Fig. 17f, which shows a Volcano plot with a brief legend that does not mention how the pausing phenotype can be appreciated. The statements regarding to pausing either need a lot of additional experimental data to support them or should be removed entirely from the manuscript.

Other comments:

1. It would be helpful if the authors could specify what proportion of genes contain TELs, so the readers can appreciate how prevalent/important these structures are.

2. I find it a bit confusing that the authors decided to name the stripes that they observe at active, long genes "Transcription Elongation Loops", since these structures represent a stripe and not a loop.

3. Lines 58-60: "To achieve scMicro-C, we implemented three key improvements (Fig. 1a, Methods). Firstly, we optimized MNase titration to prevent over-digestion and reduce DNA loss (Extended Data Fig. 1a,b)." The authors present this as a new discovery, even though this has already been described in detail previously (<https://pubmed.ncbi.nlm.nih.gov/34108683/>).

4. Extended Data Fig. 2E: The Hi-C specific loops are equally strong in the Hi-C and Micro-C panels. The same is true for the scMicro-C specific loops in the scMicro-C and Micro-C panels in Extended Data Fig. 3C. It therefore seems that the analysis/detection of these "specific" loops is not very robust.

5. Lines 120-122: "Secondly, NIPBL, a cohesin loading factor, is enriched at promoters of active genes, which indicates that cohesin prefers to load at TSSs." It has been shown that NIPBL is not (only) a loader, but travels with Cohesin as it is extruding (<https://pubmed.ncbi.nlm.nih.gov/31753851/>). Furthermore, NIPBL is not enriched at promoters (<https://pubmed.ncbi.nlm.nih.gov/36897969/>; this paper is cited in the manuscript but not in this context). This statement and the implications of it throughout the manuscript should be modified accordingly.

6. Lines 143-146: "The augmentation of interaction between the TTS and TSS upstream region suggests that cohesin stalling near TTS is independent on WAPL. These findings further confirm that both characteristics surrounding highly expressed genes are dependent on cohesin and regulated by its unloading factor WAPL." I do not understand this statement: the interactions are independent of WAPL but also regulated by WAPL?

7. The loading controls of the western blots in Fig. 2b are highly variable. It would be good if the authors could repeat this blot with proper loading controls.

8. The authors do not refer to the cell cycle analysis presented in Extended Data Fig. 10 and it is not clear what the purpose of this analysis is.

9. Lines 164-167: "To further investigate how RNAPII affects TELs, we assessed cohesin occupancy by performing Rad21 CUT&Tag after RNAPII degradation. Our analysis revealed a noteworthy reduction, to approximately half the previous level, in cohesin binding at active promoter regions (Fig. 2d)." The authors present this as a new finding, but this has been described before in the context of perturbations that affect transcription perturbation, both after RNAPII and Mediator depletion (<https://pubmed.ncbi.nlm.nih.gov/37012454/> & <https://pubmed.ncbi.nlm.nih.gov/37430065/>).

10. Fig. 3 and associated Extended Data Figures: please show CTCF annotation.

11. Lines 262-268: "Using RNA-seq data generated from GM12878 and mESC upon topoisomerase inhibition (Extended Data Fig. 15a-d), we found that TOP2 inhibition-downregulated genes, exhibiting strong TEL signal (Fig. 4a and Extended Data Fig. 11b), were significantly enriched in two categories: housekeeping genes and cell-type-specific genes (immune-related and development-related for GM12878 and mESC, respectively) (Fig. 4b, Extended Data Fig. 10h and 15e)." This statement does not seem very meaningful to me, as the expressed genes in a particular cell type can generally be divided in housekeeping and cell-type-specific genes. If both these categories are enriched, that indicates that there is no particular signature.

12. Lines 283-285: "We found that genes with TELs that are downregulated by TOP2 inhibition have higher burst frequencies than other genes (Fig. 4d, left and Extended Data Fig. 16g-i)." This is potentially interesting. However, I wonder if this could also simply reflect that these TELs are generally very highly expressed. Can the authors distinguish if this is due to the presence of TELs or reflects the expression levels? If not, this statement should be modified accordingly.

13. Lines: 297-299 "Notably, we also observed an increase in burst size among genes with reduced burst frequency (Fig. 4e right panel), which could potentially account for the modest impact on total gene expression after acute cohesin depletion." Please clarify the colour scheme and legend of Fig. 4e to support this statement.

Version 1:

Decision Letter:

9th Jul 2024

Dear Sunney,

Your Technical Report, entitled "High-resolution 3D genome structures via single-cell Micro-C uncovers cohesin-mediated enhancer-promoter loops", has now been seen by the 4 original referees. You will see from their comments below that while they find your work improved, some important points are raised. We are interested in the possibility of publishing your study in Nature Genetics, but would like to consider your response to these concerns in the form of a revised manuscript before we make a final decision on publication.

Reviewer #1 no longer has any major technical criticisms but highlights the lack of biological novelty and thinks this manuscript is now more suitable for a methods journal.

Reviewer #2 potentially supports publication, provided you adequately cite/discuss key literature and tone down some claims.

Reviewer #3 thinks the paper is much better now. However, they feel that more could be extracted from the current datasets. They have some useful suggestions for improvement.

Reviewer #4 recognizes that the method is a valuable contribution to the field but finds the level of biological novelty underwhelming; overall, these are similar thoughts to reviewer #1's.

Please note that limited biological novelty is not a concern for a Technical Report, so you do not need to address these points. However, all other comments should be addressed textually and/or via additional analyses. No new datasets or experiments need to be generated or performed.

We invite you to revise your manuscript taking into account all reviewer and editor comments. Please highlight all changes in the manuscript text file. At this stage we will need you to upload a copy of the manuscript in MS Word .docx or similar editable format.

We are committed to providing a fair and constructive peer-review process. Do not hesitate to contact me if there are specific requests from the reviewers that you believe are technically impossible or unlikely to yield a meaningful outcome.

*1) Include a "Response to referees" document detailing, point-by-point, how you addressed each referee comment. If no action was taken to address a point, you must provide a compelling argument. This response may be sent back to the referees along with the revised manuscript.

*2) If you have not done so already please begin to revise your manuscript so that it conforms to our Technical Report format instructions, available

http://www.nature.com/ng/authors/article_types/index.html

*3) Include a revised version of any required Reporting Summary: <https://www.nature.com/documents/nr-reporting-summary.pdf>

Link Redacted

We hope to receive your revised manuscript within 8 weeks. If you cannot send it within this time, please let us know.

Sincerely,

Tiago

Tiago Faial, PhD
Chief Editor
Nature Genetics
<https://orcid.org/0000-0003-0864-1200>

Reviewers' Comments:

Reviewer #1:

Remarks to the Author:

First of all, I want to stress that the authors received and dealt with criticisms (from all reviewers) very seriously. At the same time, my appreciation of the quality of scMicro-C (esp. compared to Dip-C) still stands. However, this revised manuscript has essentially done away with all perturbations and mechanistic investigations they attempted in the original version. Instead, they now provide a more descriptive approach to the Micro-C stripes they observe linked to active genes. This limits the overall enthusiasm, as these observations are not entirely new and their characterization remains pretty much on the descriptive side. For example, multi-enhancer/-promoter hubs have been already proposed via other approaches (e.g., via Tri-C). Moreover, the manuscript now lacks any complementary approach to study these structures (the authors talk about the potential of imaging in their Discussion, but it might have been better if they added such work already), while, much like in the original manuscript, most of these observations rely on ensemble scMicro-C data rather than really single-cell-derived ones. As a result, the technical advance is appreciated, but the conceptual contribution is, in my opinion, now markedly more limited than what it was in the original publication (with all its caveats) and as such, I would see this manuscript as a better fit for a Methods-oriented journal.

Reviewer #2:

Remarks to the Author:

The authors have submitted a revised manuscript that is substantially different from the original manuscript. The revised manuscript now reads more like a methods paper. On the positive side, I think the revised manuscript makes much better use of the single-cell Micro-C data and I think some of the poorly substantiated claims related to transcriptional bursting has been removed. I also appreciate the submission of a protocol.

The authors have responded to my comments. Maybe because of a misunderstanding, they did not address my "define resolution" comment, so I have tried to explain what I meant more clearly.

My other main concern was that the original manuscript overstated novelty and undercited the prior literature. I think this has been improved, but I still think the authors undercite and overstate things and it would be nice if they could tone this down a bit.

Overall, in my view this is a very nice contribution and the number of single-cell contacts they achieve is impressive and I think this paper will be of wide interest to the field and receive significant attention and as such I fully support the eventual publication. In my view, leaving aside multi-omics, this paper represents the state-of-the-art of single-cell 3D genomics. I just wish the authors would tone down a bit with some of the overclaiming and present things in a more nuanced and balanced fashion.

USEFUL READS

Line 116, Fig 1e: why do they only get 10-25% of reads that contain contacts? The Micro-C protocol should give 100% contacts assuming you use biotinylated nucleotides and streptavidin pull-down.

Looking at the protocol (thanks to the authors by the way for providing the protocol, which is essential), it looks like they skip the biotin pull-down similar to the MCC/TMCC papers. This will result in a tiny fraction of reads being useful (see Fig. 1b in Goel 2023 for a comparison). If they skip this crucial step, the authors should include a brief discussion that compare to "regular Micro-C" (Hansen 2019, Hsieh 2020, Krietenstein 2020, Goel 2023) their protocol is substantially less efficient in terms of sequencing. Their protocol may have other advantages, but this skipping biotin-pull down makes sequencing quite wasteful and the authors should acknowledge this in the text to keep the comparison to Micro-C balanced.

Along the lines of Goel 2023 Fig 1b, can the authors please report fraction of "useful reads", (unique cis reads, >1kb distance), to help compare the protocols more easily (Extended Data Fig. 2 is a bit hard for me to decipher).

Since skipping the biotin-pull-down is a significant departure from the original bulk Micro-C method, I think more clearly explaining this early on would be helpful. E.g. in line 140-143 they compare the original and the new Micro-C protocols, but they do not explain what the key differences are.

TITLE: If the paper is largely focused on "promoter stripes" why does the title say enhancer-promoter stripes? Stripes are typically originating from a single CRE, whereas an E-P interaction would result in a dot, not a stripe.

Line 267-271, the authors analyze the data from Hsieh 2022, which found most E-P interactions to be robust to cohesin depletion, but it seems like the authors here report the opposite. Why is that? Is the strong effect of cohesin unique to stripy genes? If so, please state this more clearly so the reader can understand if the authors disagree with the general conclusions of Hsieh 2022, or if this is specific to the stripy genes.

DEFINE RESOLUTION: In my first review, I asked the authors to define resolution. In line 150 they define resolution as the size of the bins used for plotting. I apologize if I was unclear, but I do not think this is a useful definition.

We can think of resolution in 2 ways: 1) in terms of data or 2) in terms of plotting. #2 is not so helpful, since any dataset can be plotted at any resolution. Even a Hi-C dataset with only 1 read across the genome can be plotted at 1bp resolution.

Therefore, I would like to ask the authors to define "resolution" in terms of the data. Through the manuscript, they frequently discuss how their data reached this or that resolution. Therefore, if they want to discuss resolution in terms of their data, they should use a data-centric way of defining resolution.

E.g. line 365-366 says they reached 5kb resolution, but if resolution just refers to plotting, they could also plot their data at 1bp resolution and claim 1bp resolution. So what do you, and please define this quantitatively, mean by the data reaching 5kb resolution?

SMALL THINGS

Fig 1j, specify color scale, is it linear or log, what's the dynamic range?

Fig 2b, not clear what is above and below the diagonal from the figure, please label

Line 230, yes the original stripe reports were architectural, but many papers since then have studied CRE stripes including some of the ones cited here, no need to overstate things.

Line 374, what's a chromatin hairpin?

Reviewer #3:

Remarks to the Author:

In this revised and strongly reworked manuscript, the authors focus on the development of the scMicro-C technology and use it for the analysis of "stripy" genes. Overall, I find the coherence and focus of the manuscript much improved. As mentioned in my previous evaluation, the results from the scMicro-C assay are impressive and the most novel aspect. The increased attention, combined with the addition of data from more cells, make the novelty better emerge. In its current state, I find the manuscript much better suited for publication in Nature Genetics.

Minor issues:

- Line 41: the authors should precise that the 1 million CREs are highly cell type specific, with a much smaller subset active in individual cells.
- Line 48: the observations in citations [15,16] were preceded by citation [12]. It should be added.

- Line 56: this observation was made in two preceding research studies: Calderon et al, eLife 2022 and Kane et al, Nat Struct Mol Biol 2022. They should be added.
- Line 57: citation [12] determined the scanning effect of loop extrusion with single cell precision. A study by Chang et al, Nature Communications 2023 used a Nano-C assay to further characterize intra-TAD loop scanning in the presence and absence of functional Cohesin. Both should be added to citations [26-28].
- Line 109: samples should read individual nuclei?
- Section 114-120: for a better understanding of these results, it will be useful if the authors can mention in the text or figure how many cells were included in each category.
- Line 130-131: I don't understand this sentence. How does the 355 cells relate to the 800 cells?
- Section 137-148: In figure 1, the authors show that their improved protocol has a very strong impact on ligation for single-cell analysis. It's surprising to see then that the bulk experiment appears very similar to the original Micro-C procedure. Could the authors reflect why this difference does not appear when comparing the bulk results? Do the two experiments differ in features that are not discussed (e.g. the number of reads containing ligation events?). Is the effect of the improved protocol for single cells largely mitigated by the incorporation of biotin enrichment in the bulk protocol?
- Section 158-214 and Figure 2: although the results that are presented in this section are impressive, they do not expand much on the methodology and results that the authors previously reported using their Dip-C approach. In its current state it therefore remains mostly limited to a proof-of-principle, rather than showing how scMicro-C goes beyond current technology. Could the authors present their results more in the context of a comparison with their previous Dip-C result? Could the authors expand the functional interpretation of the result shown in Fig. 2D-F?
- Line 192: correlation should read anti-correlation
- Line 203: it may be good to indicate that these loop anchors are anchored by CTCF
- Line 212: the same nested structures were characterized by MC-4C as well. Citation [12] should be added.
- Line 221: instead of "downstream genomic region" use "gene body"?
- Section 236-256: the order of supplemental figure panels is inverted. Fig. S10a,b appear first, followed by Fig. S10c-e and I, Fig. S9b,c and finally Fig. S10g,h. The organization of Figs. S9 and S10 should be reordered.
- Lines 265 and 275: Figs. 2e and 2f should be 3e and 3f?
- Fig 1j: could the authors indicate the size of Chr5? Adding tick marks on the axes will make the matrices easier to read. This remark expands to all interaction matrices in the supplemental information as well.
- Fig. 3a: the title "stripy – non-stripy" is not intuitive. Could this be replaced by a more clear description (e.g. delta(stripy vs non-stripy))
- Fig. 3f: I'm not sure what this model is supposed to show. I don't understand what the relative position of floating bubbles is supposed to show, and how this links to the arrows. Are certain lines/connections are missing? Is this supposed to show the scanning mechanism? Based on the recent result from Barshad et al, Nature Genetics 2023, can the authors formally exclude that scanning is not (in part) initiated from the enhancers? The title contains a typo: medaited should read mediated.
- Fig. S2a: I'm curious to see that the patterns around CTCF motifs are very similar, whereas a considerable difference is observed for TSS upon treatment with 200U and 1000U. Can the authors reflect on this?
- Fig. S2c: what is the difference between the Pooled and scMicro-C tracks? Is the scMicro-C track the larger number of 800U-treated cells that are used in the remainder of the study? The authors should clarify this. Adding the number of cells that each track is based on will be useful.
- Fig. S6. Add the word "ensemble" to the title (i.e. Ensemble scMicro-C faithfully ...)

Reviewer #4:

Remarks to the Author:

The manuscript by Wu and colleagues has been very extensively revised. The authors still present the single-cell Micro-C method, but have removed most of their perturbation experiments from the manuscript. As such, the focus of the paper has changed substantially. The main messages of the paper are now: (1) development of single-cell Micro-C; (2) long genes often form stripes which allow promoters to interact with enhancers; (3) enhancer-promoter interactions are dependent on Cohesin; and (4) enhancers and promoters interact cooperatively in hubs.

Although my technical concerns with respect to the single-cell Micro-C method have been addressed, the value of this method is still not convincingly demonstrated, as points 2-3 are mostly supported by bulk analysis. In addition, both the observation of stripes at gene promoters and their dependence on Cohesin are not novel contributions. The cooperative enhancer-promoter interactions do rely (to some extent) on single-cell analysis, but it is important to mention that similar structures have been observed in multi-contact 3C analysis in two back-to-back Nature Genetics papers in 2018 that have not been acknowledged (<https://pubmed.ncbi.nlm.nih.gov/29988121/> <https://pubmed.ncbi.nlm.nih.gov/30374068/>).

As such, the new biological insights from this paper are very limited, but I do think that the development of the single-cell Micro-C approach could be of interest to the field.

Version 2:

Decision Letter:

30th Sep 2024

Dear Sunney,

Your Technical Report, entitled "High-resolution 3D genome structures via single-cell Micro-C uncover multi-enhancer hubs", has now been seen by 2 of the original referees. You will see from their comments below that while they find your work improved, an important point is raised by reviewer #2 regarding "resolution". We are interested in the possibility of publishing your study in Nature Genetics, but would like to consider your response to this concern in the form of a revised manuscript before we make a final decision on publication.

We therefore invite you to revise your manuscript taking into account all reviewer comments. Please highlight all changes in the manuscript text file. At this stage we will need you to upload a copy of the manuscript in MS Word .docx or similar editable format.

We are committed to providing a fair and constructive peer-review process. Do not hesitate to contact me if there are specific requests from the reviewer that you believe are technically impossible or unlikely to yield a meaningful outcome.

*1) Include a "Response to referees" document detailing, point-by-point, how you addressed each referee comment. If no action was taken to address a point, you must provide a compelling argument. This response may be sent back to the referee(s) along with the revised manuscript.

*2) If you have not done so already please begin to revise your manuscript so that it conforms to our Technical Report format instructions, available [here](http://www.nature.com/ng/authors/article_types/index.html). Refer also to any guidelines provided in this letter.

Link Redacted

We hope to receive your revised manuscript within eight weeks. If you cannot send it within this time, please let us know.

Sincerely,

Tiago

Tiago Faial, PhD
Chief Editor
Nature Genetics
<https://orcid.org/0000-0003-0864-1200>

Reviewers' Comments:

Reviewer #2 (Remarks to the Author):

Being a reviewer for this paper is a bit strange, because I think the paper is pretty good, scMicro-C is technically impressive, and I have been supportive of publication, and I expect scMicro-C to receive general interest if published. In my reviews, I have asked for minor text changes and a couple of extremely simple calculations (like define and calculate resolution, calculate unique cis-reads). Whenever I submit a paper, my dream reviewer is someone who is supportive and asks for minor text changes and minor calculations that take less than a day to do, without asking for extensive experiments. Yet, despite me asking twice, the authors have still not defined resolution. This is a problem because the authors use the term "resolution" 61 times and often use it quantitatively to claim their method reached e.g. 5-kb resolution. I genuinely do not understand why the authors do not want to calculate the resolution when it is easy to do and there are well-known definitions such as Rao 2014. But if they do not clearly define resolution, all their resolution claims are meaningless. So I would like to INSIST for the THIRD time that the authors calculate the resolution of their maps or remove the word "resolution" from their paper. More about this below. I really don't get the resistance, it is easy to do, and other than this the paper is mostly pretty great, can you just please define and calculate resolution using the Rao 2014 definition?

RESOLUTION:

In my first review, I asked the authors to quantitatively define resolution. They did not do it.

In my second review, I asked the authors to quantitatively define resolution. They still did not do it.

They give some vague responses that are not clear and they say that there are no clear definitions of resolution. I do not think this is true. The Rao 2014 Hi-C paper (PMID: 25497547), which claims kilobase resolution, is arguably one of the top 3 most famous papers in the 3D genome field. It has about 8,000 citations. In their main text, they define resolution QUANTITATIVELY as "We define the "matrix resolution" of a Hi-C map as the locus size used to construct a particular contact matrix and the "map resolution" as the smallest locus size such that 80% of loci have at least 1,000 contacts.". I think their language is a little unclear, so what they mean is that the resolution of the contact map is the bin size where >80% of row-columns has at least 1000 unique contacts. So, if you have 5-kb resolution, at least 80% of each 5-kb segment along the genome should have at least 1000 unique contacts. Please use this definition, perform this calculation, and report the outcome in main text and main figures.

Why do I ask the authors to define "resolution"? Because they use it 61 times in the manuscript and often use it quantitatively along the lines of "our method is awesome, it achieves 5kb resolution". These statements are meaningless if they are not clearly quantitatively defined. And honestly, I am a little tired of this. Asking people to define their terms is not unreasonable and I have already asked them twice and it takes only some minor quick calculations.

So, in my THIRD review, I am asking them for the THIRD time to define resolution. I would like to ask the authors to use the definition of Rao 2014 outlined above. This is a famous definition from a paper with 8000 citations that every person in the 3D genome field has read and knows. I would like to ask the authors to report the resolution of the maps only using this Rao 2014 calculation (PMID: 25497547).

Overall, I think their scMicro-C technique is impressive and I think this paper is generally suited as a technical report for Nature Genetics and expect it to receive wide interest. But if the authors refuse to define and quantitatively calculate resolution – when it is easy and fast to do – after asking them 3 times, I honestly think this paper should just be rejected and I really do not understand why the authors are so opposed to define resolution.

SMALL THINGS:

1. I asked the authors to show the percent of useful contacts defined as percent of reads that are Unique+CIS+>1kb. Not sure why, but the authors say the provided this in Fig. 1e, but according to the figure legend, Fig. 1e does not show this. Can they please just report Unique+CIS+>1kb?
2. Line 405: Typo "aw well as" should be "as well as"
3. The new polymer simulations seem interesting!

Reviewer #3 (Remarks to the Author):

The authors have sufficiently addressed my remaining concerns. The modeling studies (Fig. S14) are a nice and convincing addition.

Version 3:

Decision Letter:

Our ref: NG-TR63456R2

30th Jan 2025

Dear Sunney,

Thank you for submitting your revised manuscript entitled "High-resolution 3D genome structures via single-cell Micro-C uncover multi-enhancer hubs" (NG-TR63456R2). My colleagues and I find that the paper has improved in revision, and therefore we'll be happy in principle to publish it in Nature Genetics, pending minor revisions to comply with our editorial and formatting guidelines.

Thank you again for your interest in Nature Genetics. Please do not hesitate to contact me if you have any questions.

Congratulations!

Sincerely,

Tiago

Tiago Faial, PhD
Chief Editor
Nature Genetics
<https://orcid.org/0000-0003-0864-1200>

Reviewer #1:

Remarks to the Author:

In this manuscript, Wu and colleagues present single-cell Micro-C data to investigate high resolution 3D genome structures and how they form. The scMicro-C data themselves are of high quality (please see my comments on usage below), and the analysis is in most parts technically thorough. The key question addressed here, as formulated by the authors is: "we set out to investigate whether cohesin-mediated loop extrusion facilitates transcription" and this is also pointed out in the concluding sentence of their abstract. However, I would argue that topoisomerases (with all the caveats discussed below) and RNAPs seem to have an equal, if not a more important, role here. Despite its technological advance, which not at all small, the manuscript does suffer in some aspects, and I provide below my thoughts on these that I hope the authors will find useful.

Thank you for your insightful comments and thorough evaluation of our manuscript. We greatly appreciate the feedback you have provided, particularly regarding the quality of the single-cell Micro-C data and the technical robustness of the analysis.

In response to your points concerning the scientific narrative, we have carefully considered your perspective on the roles of topoisomerases and RNAPs in addition to cohesin-mediated loop extrusion in facilitating transcription. Following a comprehensive reanalysis of the data, we have significantly revised our manuscript to emphasize the single-cell Micro-C analysis.

We have refocused our attention on exploring the regulatory function of the 'promoter stripe' in our scientific narrative. To address concerns about the lack of solid experimental support for certain aspects of the transcription elongation loop model, we have reframed the terminology.

Specifically, we now refer to the original transcription elongation loops as 'promoter stripes' and the genes associated with them as 'stripy genes'. This renaming aims to avoid any misinterpretation that these dynamic loops are solely formed through RNAPII-mediated transcription elongation. Our revised findings highlight that 'stripy genes' typically have multiple enhancers distributed along the 'promoter stripe', linking the promoter to these enhancers through cohesin-mediated loop extrusion.

Moreover, our single-cell Micro-C analysis has revealed dynamic looping events between promoters and multiple enhancers of 'stripy genes'. We have also identified a novel spatial structure, termed the 'multi-enhancer hub', wherein multiple enhancers cooperatively interact to form cluster that associates with the promoter in individual cells.

For a more detailed understanding of these revisions, we invite you to review our updated manuscript and figures. We sincerely value your feedback and look forward to any additional insights you may offer. Your contributions are instrumental in enhancing the quality and significance of our research.

KEY REMARKS:

- As the authors themselves state, the "noted that early investigations uncovered related gene-associated structures, including gene-body-associated domains and gene domains, which manifest as squares on chromatin contact maps at actively transcribed genes. However, no distinct "line" structure was discernible at the TSS, probably due to the limited resolution provided by Hi-C." This is indeed true, and although the higher resolution does help makes these structures crisper, I am still a bit skeptical as to whether high-resolution TELs are now attributed to different mechanistic origins than those inferred previously (please also see my comments below). Moreover, although the bulk of "stripes" are not TELs, and when averaged out "stripes" do not collectively behave as TELs, in multiple instances in this work (see the examples in Extended Data Fig. 6, where the two clearly mix in multiple loci) and in previous work by others having used Micro-C, TELs were clearly a subset of stripes. And I feel that this needs to be pointed out in the main text.

We are grateful to the Reviewers for their thoughtful feedback. We acknowledge the valid point you've raised regarding the overlap between promoter stripes and architectural stripes.

In response to your suggestion, we have included this clarification in the revised manuscript, specifically on Page 8, lines 234–235. We emphasize that while there is indeed a subset of promoter stripes overlapping with architectural stripes, it is important to note that the overall strength of architectural stripes surpasses that of promoter stripes, as illustrated in Extended Data Figures 4f and 9b. Furthermore, the regulatory function of architectural stripes is not clear. Regarding the promoter stripe, our proposed function is to facilitate interactions between the promoter and multiple downstream enhancers, as evidenced by focal interactions at enhancer sites.

- The discussion of the data on TELs following RNAPII depletion is very confusing to me. The authors show an (expected in my opinion) dependency of TELs on RNAPII, which, must be noted, appears less strong than that of Rad21 depletion. This is left without any discussion, as is the fact that RNAPII depletion leads to a reduction of cohesin occupancy (shown recently by Zhang et al., Sci Adv 2021; Zhang et al., Nat Genet 2023). It would be good to see how this relationship between the cohesin effects, the RNAPII effects (and later also the TOPO effects) are reconciled by the authors mechanistically, and how their relative contributions or hierarchies are dissected. Right now, this remains unaddressed.

We sincerely appreciate the Reviewers for this insightful comment. We totally agree that there is a need for additional experimental data to precisely unravel the intricate interplay between cohesin, RNAPII, and TOPO in the formation and regulation of gene stripes.

Due to space limitations in this article, we have decided not to incorporate the RNAPII and TOPO results. This decision was made with two purposes in mind: to emphasize that the involvement of RNAPII and TOPO in transcriptional loops relies on bulk perturbation experiments rather than scMicro-C, and to align with the focus of this technical report, as advised by most reviewers, which emphasizes scMicro-C analysis over mechanistic studies of TELs.

Regarding the weaker effect of RNAPII depletion on the promoter stripe, we agree with your observations and findings (Zhang et al., Sci Adv 2021; Zhang et al., Nat Genet 2023). Our data also

indicate that RNAPII depletion reduces the binding of cohesin at the promoter region. We acknowledge the need for longer depletion periods to observe a stronger effect, as you have suggested.

We maintain our belief that the promoter stripe is closely linked to transcription. Our findings are supported by RNAPII ChIA-Drop results, which also observed RNAPII-mediated gene stripes (Zheng et al., Nature 2019). Additionally, recent preprint research further supports the presence of concerted transcriptional loops mediated by cohesin and RNAPII (Kim et al., bioRxiv 2024, <https://doi.org/10.1101/2024.03.25.586715>). We intend to pursue further research to explore the mechanistic function of transcriptional loops by integrating RNAPII, TOPO, and transcriptional bursting results into another paper.

Once again, the invaluable insights from your lab regarding the role of RNAPII in 3D genome organization deeply resonate with us.

- Then, the authors move to R-loops, to build an assumption that is not supported by any data (that I can find at least). They do not map R-loops to the TSSs of TELs in their lines, they do not perform experiments with RNase H overexpression to assess potential changes in TEL structure and emergence, nor do they correlate high RNAPII occupancy with R-loops (which is a general trend found elsewhere). Also, I do not see how this at all relates with the reduced cohesin occupancy caused by RNAPII depletion, how this is relevant to TELs themselves, etc. I feel that this part of the manuscript is largely irrelevant or that it needs significantly more work to merit inclusion (mostly as a cartoon) and a place in a main figure.

We greatly appreciate the Reviewers for this suggestion. We acknowledge the valid point raised by the Reviewer regarding the lack of direct evidence supporting the speculation about the association between R-loops and the cohesin barrier at the promoter region. Our initial speculation was primarily based on previous observations indicating the enrichment of R-loops in the active promoter region (Sanz et al., Cell 2016; Dumelie & Jaffrey, eLife 2017). Additionally, a recent single-molecule study by Zhang et al. (Molecular Cell 2023) demonstrated that R-loops can function as cohesin extrusion barrier *in vitro*.

Upon carefully considering the Reviewer's feedback, we have made significant revisions to the manuscript by removing the R-loop-related results entirely.

- I think it is important to note here that TELs seems to be equally affected by FLV and RNAPII depletion, which is on the one hand expected (one would intuitively think that TELs rely on elongating RNAPs), but on the other hand surprising, as complete removal of RNAPII should fully abolish TELs. Indeed, chromatin-bound PolII cannot be so efficiently degraded in these degree systems, and the authors don't seem to degrade more than 70% or so of Ser2/5-PolII (which in our hands is not enough to see very strong 3D-genome changes and comparable to a prolonged TRP treatment). This is due to the 4-hour time point used, and it would be nice to employ a longer time point and get depletion to >85% to provide a more precise picture to readers.

We appreciate the valuable suggestion provided by the Reviewer. As mentioned earlier, we have already removed the RNAPII-related findings from the revised manuscript.

It is indeed acknowledged that the depletion of chromatin-bound RNAPII, as indicated by Ser2/5 phosphorylation, may not have been sufficiently extensive within the 4-hour timeframe of auxin treatment. Employing a longer degron treatment duration, such as the 14-hour timeframe suggested in your study, could potentially yield a more comprehensive understanding of RNAPII depletion effects.

Regarding the impact of RNAPII on TELs, we hypothesize that both promoter-bound RNAPII, which acts as a barrier for cohesin, and elongating RNAPII, which interacts with extruding cohesin, may have dual roles, as suggested by computational simulations conducted by the Mirny lab (Banigan et al., PNAS 2023). However, further experimental evidence is required to validate these hypotheses and provide a clearer understanding of their implications.

- The next part on topoisomerase and TELs, is potentially very interesting, but there are a number of concerns associated with it. First, although the premise of the hypothesis is really interesting, the means by which the authors have chosen to interfere with TOP1 and TOP2 activity can prove very problematic. Both TPT (a drug developed on the basis of camptothecin) and Etoposide are both strong inducers of DNA damage, they do not really alleviate TOPO activity but rather lock complexes in place (see Gothe et al., Mol Cell 2019 for an exemplar of this) and certainly cause significant DNA damage (this is why they are used as anticancer drugs) even at short time frames and low doses. The authors have not controlled for this, but looking at the cell cycle analysis, there is a pronounced S-phase signal accumulation signaling cell cycle arrest (as would be expected by these drugs and by damage). To my surprise though, the numbers in the inset of the graphs in Extended Data Fig. 10a do not at all reflect the curves in the plot -- S-phase percentages seem to only increase mildly, and G1 ones not to really drop, but the structure of the data curves clearly shows something else. Moreover, the figure legend (as most figure legends in the manuscript actually) offer very little information to understand the plot. For example, the the coloured contours seems to be fitted to the squiggly line that represents the raw data. Is this true? Because the fit is not good at all. What is quantified here? Simple comparison of the overarching squiggly line disagrees with the numbers presented. I am also pretty sure that the genes unregulated by these drugs would show clear signs of cell cycle arrest and DNA damage response induction, but they do not seem to be analysed in this direction. Therefore, the proper way of addressing the question the authors are posing here is to use either siRNA-mediated depletion or, even better, TOP1/TOP2 degron lines. These pharmacological inhibitors are not fit for purpose.

We sincerely appreciate the constructive feedback provided by the Reviewer. In response to their insightful comments, we have decided to exclude the results pertaining to the inhibition of topoisomerases from the revised manuscript.

We acknowledge the concerns raised regarding the use of pharmacological inhibitors to modulate TOP1 and TOP2 activity and their potential to induce DNA damage. However, it's important to note that topoisomerase inhibition specifically affects gene stripes but not architectural stripes.

As for the cell cycle concerns, the treatment duration and cell cycle characteristics of GM12878 are different from existing literature predominantly based on cancer cell lines, which typically involve longer treatments (24 or 48 hours) leading to significant cell cycle arrest. In our study, we treated GM12878 cells with Etoposide and Topotecan for 8 hours, which may explain the discrepancies in our FACS data compared to previous studies. We apologize for any confusion caused by the discrepancy between our FACS data fitting and the actual cell cycle composition, and we are committed to further investigating and refining this aspect of our analysis.

Regarding the suggestion to employ siRNA-mediated knockdown or TOP1/TOP2 degron lines for more targeted investigations, we appreciate the recommendation. However, due to the complexity associated with establishing degron cell lines for simultaneous depletion of both TOP2A and TOP2B, and considering the redundancy of these enzymes in mammalian cells, we believe that exploring siRNA-mediated knockdown in mESC represents a more feasible and targeted approach for future studies to validate the impact of TOP2 on TELs. We are committed to incorporating these considerations into our future research endeavors.

- Second, even with the caveats of these inhibitors aside, the authors report effects on TELs following TOPO inhibition for 8 h. They report that "TEL signals were significantly weakened upon TOP2 inhibition, while [...] slightly strengthened upon TOP1 inhibition" and they then correlate that with "cohesin binding at gene body [being] significantly weakened upon TOP2 inhibition but not TOP1". Given the interdependency of RNAPII elongation and supercoiling resolution (e.g., see work by the Baranello lab), and the fact that TELs rely on elongating RNAPs, was this not to be expected? To advance our mechanistic understanding, one would need to map actual supercoils/topoisomerases and were these accumulate upon drug inhibition to really assess the phenomenon. Moreover, the drop in cohesin levels and "retention of cohesin at gene body" claimed is not that strong in the mean plot of Fig. 2g (and the word "significantly" implies some sort of statistical evaluation that I cannot locate in the text), but also does not explain the strengthening of upon TOP1 interference if cohesin remains unchanged. But, what happens to RNAPII in either case is not investigated and therefore the cartoon in panel 2h remains purely speculative.

We appreciate the valuable feedback provided by the Reviewers. In response to your insightful comments, we have decided to exclude the model from the updated manuscript.

Regarding the effects of topoisomerase inhibition on cohesin dynamics, it is plausible that these effects are mediated through RNAPII, as topoisomerases are closely associated with transcribing RNAPII. Recent preprints by Davidson et al. (bioRxiv 2024) and Janissen et al. (bioRxiv 2024) found that cohesin introduces negative supercoiling into DNA loops during loop extrusion. Therefore, it is conceivable that the effects of topoisomerase inhibition could involve perturbations in supercoiling accumulation, thereby interfering with cohesin extrusion. However, further experimental data are necessary to validate this speculation conclusively.

We acknowledge the reviewer's observation regarding the metagene plot showing changes in cohesin binding at gene bodies. While we did observe reduced cohesin binding at specific genes,

we recognize the need for more quantitative measurements, such as spike-in calibration, to validate these changes more comprehensively.

We thank the Reviewers for the suggestion of checking RNAPII distribution followed by topoisomerase inhibition. Previous TOP1 inhibition on mouse cortical neuron found topotecan reduces RNAPII density on gene body of long genes (King et al., Nature 2013). The effect of TOP2 inhibition on RNAPII was not investigated in the previous study. We intend to explore RNAPII distribution changes upon topoisomerase perturbation in future work.

In consideration of clarity and scientific rigor, we have decided to remove the TEL regulation cartoon model from the revised manuscript.

- Now, concerning the single-cell conformation analyses, I remain a bit ambiguous. On the one hand, it is clear to this reviewer that scMicro-C surpasses Dip-C and is probably the best tool to date. At the same time though, there are three issues that need to be addressed in my opinion. First, it is the resolution issue. The central claim of "improving the spatial resolution of single-cell 3D genome mapping to 5 kb" (already highlighted in the manuscript's summary), can be misleading. How is resolution defined in such a sparse dataset? How is effective resolution defined? I assume one could bin Dip-C data to 5-kbp resolution and would still see sparse signal. How can this signal be interpreted on its own. It seems to me that the authors (a) rely on the bulk signal to identify looping interactions and TELs and then map these back onto single-cell structures, usually on the basis of single in 1 or 2 bins at best (see examples in Extended Data Fig. 4), and (b) rely on the distance modelling to infer conformations. None of these two are per se wrong, of course, but they do substantially moderate the claim of bona fide single cell-resolution observations. My reading of this work is that it relies on high-quality bulk data observations, and that the single-cell component is rather a derivation. Especially, the (otherwise very nice) structures deduced from a TEL cannot be claimed to represent the gradual elongation of the loop, as these snapshots can only be temporally ordered artificially and cannot be related with the respective positions of RNAP and/or cohesin in each cell. Therefore, I would urge the authors to tone-down these claims, and to also remove or rephrase the single-cell bit of their title accordingly.

We sincerely appreciate the Reviewers for bringing up concerns regarding the definition of resolution in scMicro-C and the analysis of single-cell 3D genome structures. We apologize for any confusion that may have arisen due to unclear descriptions.

To address your concerns, we will provide detailed explanations for each point raised. Firstly, concerning resolution in single-cell Micro-C or Dip-C, it is crucial to understand that the definition differs from that of bulk Micro-C or Hi-C. In our methodology, resolution is determined by the reconstructed 3D genome structure, where, for instance, a resolution of 20 kb signifies that each particle of the 3D genome structure represents a 20 kb genomic bin. In the case of 20 kb resolution, there are approximately 30,000 particles per cell. Achieving a 5 kb resolution means that each particle represents a 5 kb genomic bin, resulting in approximately 120,000 particles per cell. This concept aligns with DNA FISH-based microscopy methods, where resolution is determined by the labeling unit.

We define the resolution of single-cell 3D genome structures based on modeling uncertainty. Specifically, if a cell demonstrates low uncertainty at a specific resolution, we claim that the single cell has achieved that resolution. For each resolution, we generate five independent structures for each single cell and then align these structures to calculate the root-mean-square deviation (r.m.s.d) for each particle. If the median r.m.s.d value across all particles is less than 2, the cell is considered to have low uncertainty at that resolution. Please see Extended Data Fig. 6 for a representative single-cell with low uncertainty. In Dip-C, cells can achieve a 20 kb resolution reconstruction with low uncertainty, while in scMicro-C, cells can achieve a 5 kb resolution reconstruction with low uncertainty. We hope this explanation clarifies the resolution issue in scMicro-C.

It's important to note that 3D genome modeling is based on contact maps, and we have demonstrated a high correlation between the mean 3D distance from modeled single-cell structures and bulk Micro-C contact frequency (Fig. 2c and Extended Data Fig. 6c).

Regarding the detection of chromatin loops or promoter stripe patterns, we did rely on bulk Micro-C or ensemble scMicro-C profiles. For single-cell analysis, we use either single-cell contact maps or modeled 3D genome structures. We define a looping/contact event if there is a detected contact between two genomic bins or if the 3D distance between genomic bins is less than a certain value (in this study, set at 3.5 particle radii, approximately 240 nm). However, we acknowledge that looping events detected in single cells may not directly translate into interactions mediated by RNAPII or cohesin, as simultaneous protein binding information is lacking. Additionally, these interactions may capture random co-localization of genomic loci. In the revised manuscript, we performed single-cell 3D genome structure analysis focusing on interactions between multiple enhancers and the promoter of 'promoter stripe', considering them as functional E-P interactions, rather than interpreting them as a reflection of the cohesin-mediated loop extrusion process.

Furthermore, single-cell structure analysis reveals structural variability and dynamics not observed in bulk Micro-C analysis. In the revised manuscript, we uncover dynamic interactions between promoters and multiple downstream enhancers of stripy genes among individual cells. Additionally, we find that these enhancers cooperatively interact to form spatial clusters associated with the promoter. These observations highlight the unique insights gained from single-cell analysis that cannot be derived from bulk Micro-C analysis alone.

- The changes in bursting inference is a welcome addition to the paper, but (a) simple modelling of scRNA-seq data does not suffice to substantiate the claim and orthogonal real-time data is needed, and (b) the dependence of bursting on cohesin has been thoroughly demonstrated before (e.g., Luppino et al. Nat Genet 2020, Robles-Rebollo et al., Nat Commun 2022) and does not seem to be confined to TELs, even in this work. So, I would suggest to turn this part of the manuscript into a Supplemental Figure.

We thank the Reviewers for providing the constructive feedback. In response to their suggestion, we have removed the transcriptional bursting data from the updated manuscript.

Regarding the need for orthogonal validation methods such as single-molecule RNA FISH or real-time imaging, we acknowledge the importance of these approaches in corroborating our findings. However, we would like to emphasize that inferring transcriptional bursting kinetics from scRNA-seq data is a well-established method in the literature, pioneered by researchers such as Rickard Sandberg and John C. Marioni. While we acknowledge the higher sensitivity of RNA FISH, it is worth noting that comparisons between scRNA-seq and single-molecule RNA FISH have shown comparable results for certain genes (Larsson et al., Nature 2019). Real-time imaging, while powerful, is limited by the necessity to construct specific cell lines for tagging endogenous genes with MS2/PP7, thus providing a more targeted rather than comprehensive view of the transcriptome.

Regarding the relationship between cohesin and transcriptional bursting, we are aware of the studies by Luppino et al. and Robles-Rebollo et al. that demonstrate the impact of cohesin on transcriptional bursting of TAD boundary-proximal genes and inducible genes, respectively. In the original manuscript, we aimed to attribute changes in bursting following cohesin depletion to increased RNAPII pausing and to associate cohesin's effects with a specific gene stripe structure. However, we recognize the importance of validation through orthogonal methods. Therefore, in the revised manuscript, we have removed the transcriptional bursting findings as suggested.

SECONDARY REMARKS:

- I see that the authors explicitly avoid "overdigestion" with MNase and therefore have picture of mono-, di-, ori-nucleosomes, etc. The "classical" Micro-C protocols urge users to indeed digest down to mono-nucleosomes. How do you think the high-order nucleosome stretches affect your data? Can you decipher and look at them separately? What can we learn from it?

We sincerely appreciate the insightful comments provided by the Reviewers. Our Micro-C protocol intentionally avoids "overdigestion" with MNase, deviating from bulk Micro-C protocol. This decision is based on our observation that overdigestion leads to significant DNA loss and low ligation efficiency, particularly problematic for single-cell applications where short ligation products are incompatible with transposon-based whole-genome amplification. It's noteworthy that the avoidance of overdigestion is a principle embraced by other Micro-C derivatives, including MCC (Hua et al., Nature 2021) and tiled-MCC (Aljahani et al., Nature Communications 2022).

One of the key strengths of Micro-C lies in its utilization of MNase, which fragments chromatin without requiring specific recognition sequences, unlike the restriction enzymes (REs) used in Hi-C. Hi-C encounters limitations due to the limited number of cutting sites generated by REs, leading to limited and predetermined ligation possibilities between RE-digested fragments and resulting in blurry matrices at higher resolutions. In contrast, Micro-C's use of MNase allows for ligation at any chromatin site, generating sufficient ligation complexity at higher resolutions. It's important to note that the resolution of Micro-C is not solely determined by the fragment size after MNase digestion; rather, given the generation of second-generation sequencing paired-end reads of 150 bp, the detected chromatin contacts consistently involve pairs of nucleosome-sized fragments. This principle is also endorsed by MCC and tiled-MCC.

In our methods section, we meticulously outline the generation of two bulk Micro-C replicates. Replicate 1 uses all ligated fragments (ranging from 300 to 3000 bp), subsequently sonicated to approximately 350 bp, while replicate 2 only selects di-nucleosome fragments for library preparation. Notably, our analysis (Extended Data Fig. 4) reveals no significant discrepancies between the two replicates.

Therefore, the rationale behind titrating MNase to prevent overdigestion serves two main purposes: ensuring compatibility with single-cell Micro-C experiments by minimizing DNA loss and preserving adequate ligation product lengths for efficient Tn5 transposase-based whole-genome amplification.

- Ext. Data Fig. 1c: showing 500-kbp-resolution maps from whole chr6 does not allow assessment of the true scMicro-C coverage, which, although impressive compared to the single-cell 3D assays, is not in itself devoid of very high sparsity (see Fig. 3a). Therefore, please replace with zoom-ins of the level of information claimed by the manuscript's summary (i.e. 5 kbp) from different chromosomes. The same should be done at different instances in the main Figs.

We thank the Reviewers for this suggestion. To address this concern, we have plotted scMicro-C contact map zoom-ins of 5 kb resolution at a 3 Mb genomic region.

To assess the coverage of scMicro-C at 5 kb resolution, we quantified the non-zero bins for chromosome 2, focusing solely on intra-chromosomal contacts. The median percentage of non-zero bins for chromosome 2 was found to be 76%.

- Ext. Data Fig. 1f,g should include data from the "standard" and from the authors' modified bulk Micro-C for comparison.

We sincerely appreciate the valuable suggestion from the Reviewers. In response, we have revised Extended Data Fig. 5 to incorporate both "standard" and "modified" bulk Micro-C data into the violin plot for enhanced comparison (for consistency, we only take the ≥ 1 kb contacts for both single-cell and bulk into consideration). However, we have chosen not to include bulk Micro-C data in the scatter plot depicting reads versus contacts, as this integration would not provide meaningful insights.

- Ext. Data Fig. 1e: the authors claim an average of 1M contacts per cell, but the histogram shows a red line at the 0.8M bin. Which is true?

We thank the Reviewers for bringing this to our attention. We apologize for any confusion caused. In the original dataset, the median number of contacts among the 355 cells was recorded at 0.926 million. In the updated dataset, the median number of contacts among the 750 cells stands at 0.835 million.

- The GRO-seq axis labels in Fig. 1 seem to have moved out of place.

We thank the Reviewers for this notice. We have removed GRO-seq signal from the revised manuscript.

- The statement that "Lastly, in the absence of CTCF and cohesin unloading factor WAPL, cohesin accumulates at the 3' end of some active genes, implying that cohesin tends to unload near TTSs" is probably not very accurate, since this simply reflects where cohesin will end up given the forces exerted on it and around it on the finer (e.g., transcription, replication, diffusion). I would rephrase or the authors can test this by combining WAPL depletion with RNAPII depletion to assess each effect's weight on this effect.

We greatly appreciate the Reviewers for this suggestion. We totally agree that the accumulation of cohesin at the transcription termination sites (TTSs) following double knockout of CTCF and WAPL is possibly regulated by elongating RNAPII. In response, we have decided to remove this specific description from the revised manuscript.

- RNAPII is referred to as a "barrier to cohesin" in many places in the manuscript by citing the Bannigan et al. work, whereas direct evidence to this was recently provided by Zhang et al., Nat Genet 2023. I would probably cite both papers accordingly.

We sincerely apologize for this oversight. Indeed, the study by Zhang et al. (Nat Genet 2023) provided compelling evidence regarding RNAPII's role as a barrier to cohesin, particularly

demonstrated through the rewiring of CTCF loops upon RNAPII depletion. In the revised manuscript, we have removed the RNAPII related results. Moving forward, we will ensure to appropriately cite both Bannigan et al. and Zhang et al. in any future work that integrates RNAPII and TOPO results.

- I am sorry if I missed this, but are all cells to which scMicro-C was performed synchronised and in the G1-phase? If not, how can cell cycle effects be ruled out?

We sincerely appreciate the Reviewers for bringing up this point. We apologize for any confusion caused. The cells subjected to scMicro-C were FACS selected, ensuring that only G1-phase cells were sorted for amplification. To address this, we have included this clarification in the revised methods section of the manuscript.

- The statement "Conversely, an analysis of a similarly sized genomic region without TELs or stripe structures (chr2: 114.25-115.15 Mb) reveals a decrease in the number of structures with a larger separation of genomic distance (Extended Data Fig. 13g,i)" is very unclear to me and it would be helpful if the authors explained what this region is, why it was selected, and how this conclusion was reached. However, I would have expected that the control in this case would be an equisized and equally well transcribed but TEL-free region of the genome.

We thank the Reviewers for providing this valuable suggestion. We apologize for any confusion caused. In the revised manuscript, we have addressed this concern by selecting six non-stripe control regions for comparison, as depicted in Extended Data Fig. 10. These control regions were chosen from areas of the genome where no specific chromatin structure (such as loops, stripes, or boundaries) was identified, using as background. The inclusion of these control regions aims to elucidate how the presence of "promoter stripe" structures enhances interactions between the promoter and multiple downstream enhancers.

- I feel that the text would benefit from more detailed explanations in most parts, especially in the single-cell section, and that figure legends suffer rather than benefit from their brevity.

We sincerely appreciate the Reviewer's feedback and acknowledge the suggestion for more detailed explanations, particularly in the single-cell section of the manuscript. We also recognize the concern regarding the brevity of figure legends. In response to these valuable points, we have extensively revised both the manuscript and figure legends to ensure a comprehensive understanding for readers. Thank you for highlighting these areas for improvement, and we hope that the revised manuscript will better meet the needs of our readers.

A. Papantonis

Reviewer #2:

Remarks to the Author:

This paper reports the development of a single-cell version of Micro-C and transcriptional elongation loops (TELS). scMicro-C seems impressive from a technical perspective. Getting 1 million unique contacts per cell is impressive. By re-analyzing Micro-C from Hsieh 2022 they show that TELS are cohesin dependent and by performing RNA Pol II depletion micro-C they also show that TELS are RNA POL II dependent. Furthermore, experiments +/- TOP1 and TOP2 inhibition show the involvement of topoisomerases.

Although the authors only gain modest value from the scMicro-C data (in the sense that most of their conclusions could have been made with bulk Micro-C), overall I think the combined contributions of a highly efficient scMicro-C protocol and interesting insights into TELS together make for a nice contribution to the literature, that I think will be of significant interest to the broad readership of Nature Genetics if the authors can address the major and minor concerns listed below.

We thank the Reviewer for considering our manuscript “impressive.” We also appreciated the Reviewer’s recognition of our scientific findings. We would like to express our gratitude to the Reviewer for their kind words.

In this revision, we have refrained from using the phrase ‘transcription elongation loops (TELS)’ to characterize the line structures we observed. Instead, we now refer to these structures as ‘promoter stripes.’ More specifically, we refer to the line structures that appear at promoters and align with genes as ‘promoter stripes’ and genes with such structures as ‘stripy genes.’ The term TELS may lead readers to misunderstand that this structure is directly formed by the elongating RNA Pol II. Another reason to use the term ‘promoter stripes’ is to acknowledge prior work.

To align with the scope of Technical Report, we focus on technical aspects of single-cell Micro-C and have removed all data related to transcription perturbation, topoisomerase inhibition and transcription bursting from the manuscript, as suggested by most Reviewers. In this revision, we have conducted additional validations of our experimental procedures and expanded the dataset from 340 cells to more than 800. We have meticulously conducted analysis of the single-cell 3D genome structures of stripy genes, with a specific focus on the intricate enhancer-promoter interactions related to this structure. We observed the formation of multi-enhancer hub using single-cell data, wherein multiple enhancers cooperatively interact with the same promoter to form a hub in individual cells.

We have made numerous substantial changes to the text and figures and look forward to hearing your valuable thoughts and insights.

MAJOR COMMENTS

BURSTING: I know there are papers out there claiming to quantify transcriptional bursting from single-cell RNA-Seq but I believe those papers are wrong for the following reasons:

- 1) Cells contain multiple allele (especially after replication) – thus the statistics will need to reflect the number of gene copies which the model does not
- 2) The authors fail to distinguish intrinsic vs. extrinsic noise
- 3) scRNA-Seq has low detection efficiency, if you cannot measure single RNAs reliably, you cannot quantify the distribution
- 4) transcriptional bursting is a dynamic property about the lifetime of bursts – you can measure dynamic properties using snapshot methods.

I think the authors should remove all bursting claims. If they want to make claims about transcriptional bursting, they should demonstrate their claims using live cell imaging of nascent transcription such as MS2 and PP7 imaging demonstrated in e.g. Wan...Larson Cell 2021.

We thank the Reviewer for this very helpful suggestion. We completely agree with the Reviewer that there are lots of technical issues associated with transcriptional bursting estimation based on scRNA-seq data. Single-molecule FISH and live cell imaging of nascent transcription are considered the gold standards for measuring transcriptional bursting kinetics; however, we currently lack the necessary experimental equipment to conduct these experiments. Due to the above reasons, we have removed all data and claims related to transcription bursting from the manuscript.

We find that promoter stripes enrich active enhancers and are associated with multi-enhancer hubs. Hence, we have added the following sentence in the Discussion section to discuss their potential function:

“Given the dynamic nature of the multi-enhancer hub assembly in individual cells, it may potentially modulate the transcriptional bursting of stripy genes.”

LOW-RESOLUTION MICRO-C: It appears from Fig. 1b that their Micro-C has lower resolution than Hi-C using 4-cutters because the ligation products seem to be around 1kb in size. The authors should comment on this limitation in the main text. The authors should also provide an scMicro-C protocol with their paper.

We sincerely appreciate the reviewer's insightful comment and suggestion. Firstly, we'd like to clarify that our bulk Micro-C dataset, comprising 4.4 billion contacts, indeed demonstrates a higher resolution compared to bulk Hi-C, which encompasses 4.9 billion contacts. This resolution difference is evident in our high-resolution contact maps, where Micro-C effectively reveals clearer and more distinct chromatin features such as chromatin loops and stripes, as compared to bulk Hi-C.

To address the reviewer's suggestion, we have published a comprehensive step-by-step scMicro-C protocol in protocols.io ([dx.doi.org/10.17504/protocols.io.kqdg39wbzg25/v1](https://doi.org/10.17504/protocols.io.kqdg39wbzg25/v1)), now we incorporated this link in the revised methods section.

TEL: around lines 334-340 the authors claim that TELs are novel. This is not fair – this concept has been known and discussed for decades. This paper is still a nice contribution, but it is not fair to prior work to describe things the way they do here. They should cite earlier studies on TELs as well.

We thank the Reviewers for this suggestion. We apologize for any oversight regarding prior literature on the concept of transcription elongation loops (TELs). In response to this feedback, we have made necessary revisions in the manuscript. Specifically, we have renamed ‘transcription elongation loops’ to ‘promoter stripes’ to better align with existing terminology and to avoid any potential misinterpretation. Additionally, we have included references to earlier studies that have discussed similar gene structures, albeit with a limited number of genes referenced (Page 7, line 221–225).

PROVE: at many points in the abstract and text that author use the words “proof” and “prove” – given the caveats associated with experiments, I think this is too much and not reasonable. I would like to ask the authors to use words like show or demonstrate instead.

We sincerely thank the reviewer for providing such valuable feedback. We acknowledge and agree with the observation regarding the use of the word “prove,” and we understand the importance of using more cautious language. In response to this suggestion, we have revised the manuscript accordingly and replaced instances of “prove” with more appropriate terms such as “show” or “demonstrate.”

MINOR COMMENTS

The authors use the word Resolution several time – can they precisely define what they mean by this term?

We sincerely appreciate the valuable feedback provided by the reviewer. We apologize for any confusion arising from the lack of precise definition regarding the term “resolution” in our manuscript.

In our study, we employ the term “resolution” in two distinct contexts. Firstly, in the analysis of contact maps, resolution refers to the bin size of the contact matrix. For example, a resolution of 5 kb indicates that each pixel in the contact map represents a genomic bin of 5 kb, with the value indicating the normalized total contacts within this bin. Secondly, in single-cell 3D genome modeling, resolution signifies the genomic bin size represented by each particle in the 3D genome structure. For instance, at a resolution of 20 kb, each particle represents a 20 kb genomic bin, resulting in approximately 30,000 particles per cell. Achieving a 5 kb resolution means that each particle represents a 5 kb genomic bin, leading to approximately 120,000 particles per cell. This notion aligns with DNA FISH-based microscopy methods, where resolution is determined by the labeling unit.

In our methodology, the resolution in 3D genome structure modeling is determined by the uncertainty of the reconstructed structures. For each cell, we generate 5 independent structures for each resolution, and then align these structures to calculate the root-mean-square deviation (r.m.s.d). Only cells with low uncertainty (in this study, defined as a median r.m.s.d ≤ 2) are considered for analysis.

We have incorporated a clear definition of resolution in both contexts in the revised manuscript for clarity (please refer to Page 5, line 150, and Page 6, lines 167–169).

Line 121 refers to an old NIPBL-ChIP-Seq study, but the recent paper from Banigan showed that many NIPBL ChIP-Seq papers suffered from bad antibodies. Please double-check this reference.

We thank the Reviewers for bringing up this valuable point. We have carefully considered your feedback and decided to remove the specific text from the revised manuscript.

Regarding to the NIPBL ChIP-seq results, we think more experimental data is needed to reconcile these inconsistencies. In the paper by Banigan et al. from the Mirny lab, they proposed that cohesin loads uniformly at all genomic loci based on their NIPBL ChIP-seq findings, which showed limited overlap with active transcription start sites (TSSs). This finding has caused confusion because the antibody they used is the same as the one mentioned in the literature (Zuin et al., PLOS Genetics 2014), which claims that over 1/5 of NIPBL ChIP peaks are located at active TSSs. We are perplexed as to why the same antibody generates conflicting results. Therefore, we believe that further confirmation of the NIPBL ChIP-seq results is necessary.

It is important to note that whether cohesin tends to load at active DNA elements remain subjects of ongoing debate and warrant further experimental validation. Several observations, such as the formation of chromatin jets (Guo et al., Molecular Cell 2022) and chromatin fountains (Galitsyna et al., bioRxiv 2023 (from Mirny’s lab); Isiaka et al., bioRxiv 2023; Kim et al., bioRxiv 2023), suggest cohesin specifically loads at active enhancer sites. Additional evidence, such as the effects

of enhancer insertion and deletion on cohesin occupancy at neighboring CTCF-binding sites (Rinzema et al., Nat. Struct. Mol. Biol. 2022; Vos et al., Molecular Cell 2021), further support the notion of active regulatory elements recruitment of cohesin. The bias distribution of NIPBL and enhancers towards the anchor side also contributes to this discussion (Vian et al., Cell 2018). Taken together, the binding of NIPBL at active promoters and the loading of cohesin at active DNA elements are subjects of extensive debate and require further experimental validation.

In light of the lack of consensus regarding cohesin loading patterns, we have opted not to discuss cohesin loading in the updated manuscript.

The authors refer to dynamic loops in lines 92 and 246 and other places, but they have not measured dynamics. Therefore they should cite recent papers that measured cohesin loop dynamics such as Gabriele 2022 and Mach 2022.

We sincerely appreciate the insightful suggestion provided by the Reviewer. We acknowledge that our use of single-cell 3D genome structures only provides a snapshot of looping events at the time of fixation, rather than direct measurement of dynamics. In response to this feedback, we have included references to recent papers by Gabriele (2022) and Mach (2022), which measure cohesin loop dynamics through live-imaging techniques, in our revised manuscript.

The authors mentioned that they did not have sufficient resolution to detect TELs at short genes. A recent very high resolution Micro-C study by Goel et al. 2023 includes cohesin depletion and transcriptional inhibition – can the authors use these data to test if TELs exist for short genes? This would help establish the generality of these TEL structures.

We thank the Reviewer for this very helpful suggestion. We have downloaded the dataset from Goel et al. 2023 and attempted to identify the promoter stripe structure at 500 bp resolution. In the five captured genomic regions (specifically, chr3, chr5, chr6, chr8 and chr18), a total of 75 gene are expressed, and 108 genes are not expressed. Initially, we used the Stripenn method to calculate the stripe structure, but encountered a bug that precluded us from obtaining output. We think that Stripenn does not support captured data. Subsequently, we visually inspected these 181 genes and did not find any stripe structures. The screenshot below shows a genomic region that was captured. Notably, among these genes, three lengthy genes—Rbks (73.2 kb), Slc2a3 (73.8 kb), and Cacna1a (301.6 kb)—were identified as stripy genes at 5kb resolution based on Micro-C data from Hsieh et al. 2020. However, the two stripes (Slc2a3, Cacna1a) were a little discontinuous/weak at 500 bp resolution. This discrepancy may be due to the sparseness of data at 500 bp resolution or differences caused by different resolutions. In sum, we did not observe any stripe structure for short genes based on data from Goel et al. 2023. In the future, genome-wide ultra-high-resolution data may be more helpful in answering this question.

5kb resolution

500bp resolution

Reviewer #3:

Remarks to the Author:

Review of “Extruding transcription elongation loops observed in high-resolution single-cell 3D genomes”.

Transcription is a fundamental biological process, which involves a profound remodeling of chromatin/DNA structure. In this Technical Report, the authors use a new scMicro-C method to characterize Transcription Elongation Loops (TELs) in highly expressed long genes. These observations are combined with the (re)analysis of large amounts of bulk-cell studies (Micro-C, Hi-C, RNA-seq) and single-cell RNA-seq upon the perturbation of various factors involved in transcription (RNA-PolII, TOP2, Cohesin). Based on these studies, the authors conclude that TELs are the result of the interplay between the transcription, loop extrusion and topoisomerase machineries, and that TELs may be important to increase transcriptional bursting by paused RNA PolII release.

Using scMicro-C, the authors describe an intriguing pattern of chromatin interactions at transcribed long genes, which provide a promising insight into the link between chromatin/ DNA structure and transcription. A very large part of the study (Figs. 2 & 4) is dedicated to results from other types of experiments (bulk-cell and single-cell RNA-seq), with outcomes that in many cases remain correlative at best (and for some results even highly speculative).

Besides the fact that this part of the study deviates from the goals of a Technical Report, I am not convinced that it is appropriate to interpret the TEL kinetics, obtained from scMicro-C studies in normal cells, in the context of the correlative mechanistic insights obtained from the bulk-cell and single-cell RNA-seq studies. In my opinion, to make these results a better fit for a Technical Report in Nature Genetics, the authors should reduce their attention to the bulk-cell and single-cell RNA-seq data, and instead expand their analysis of the scMicro-C results, focusing on the different contact patterns at various genic and non-genic regions in the genome.

For ease of discussion, I have grouped my remarks for each figure together. Some general comments are provided at the bottom.

We really appreciated the Reviewer’s many insightful questions and suggestions, which have undoubtedly enhanced the quality and clarity of our manuscript. We therefore thank the Reviewer for their thorough review and thoughtful comments. Before addressing specific points, we would like to point out the significant changes made in this revision.

First, we have abstained from using the phrase ‘transcription elongation loops (TELs)’ to characterize the line structures we observed. Instead, we now refer to these structures as ‘promoter stripes’ and genes with such structures as stripy genes. We make this adjustment to acknowledge prior work and to avoid potential misunderstandings that this structure is directly formed by the elongating RNA Pol II.

Second, to make this manuscript fit for a Technical Report, we have removed all bulk data associated with transcription inhibition and topoisomerase inhibition, as well as scRNA-seq data related to transcriptional bursting, as suggested by the Reviewer.

Third, we have conducted additional validations of our experimental procedures and expanded the dataset from 340 cells to more than 800. We have meticulously conducted analysis of the single-cell 3D genome structures of stripy genes, with a specific focus on the intricate enhancer-promoter interactions related to this structure. We observed the formation of multi-enhancer hub using single-cell data, wherein multiple enhancers cooperatively interact with the same promoter to form a hub in individual cells.

We have implemented numerous significant revisions to the text and figures and are eager to receive your valuable feedback and insights.

Figure 1.

R1.1: The development of scMicro-C is an impressive achievement, with a number of important improvements over their previously developed Dip-C method. The way that these improvements (increased resolution, increased contacts/cell) are justified is different from how the advantages of Micro-C are generally presented. As shown in Hsieh et al., Mol Cell 2020 and Krietenstein et al. Mol Cell 2020, the main advantage of Micro-C is not so much the increased resolution (commercial Hi-C kits using 4 restriction enzymes should theoretically generate much higher resolution data), but rather that the cuts are more regularly spaced. This generates data with a much-improved signal over noise, allowing reliable measurement of chromatin organization at shorter ranges: compare Fig. S2b to similar plots in Hsieh et al. and Krietenstein et al.

We greatly appreciate the Reviewer's recognition of our technical advancements.

To provide clarification, while Micro-C indeed offers enhanced resolution compared to restriction enzyme-based Hi-C methods, it's essential to note that this improvement stems from the use of MNase, which fragments chromatin without necessitating specific recognition sequences, unlike restriction enzymes. Consequently, the limitations encountered by Hi-C, such as the restricted number of cutting sites generated by restriction enzymes leading to predetermined ligation possibilities and blurry matrices at higher resolutions, are overcome by Micro-C's utilization of MNase. This versatility allows for ligation at any chromatin site, facilitating sufficient ligation complexity at higher resolutions (as illustrated below). It is indeed true that both Micro-C and scMicro-C demonstrate significantly improved signal-to-noise ratios for chromatin loops and stripe detection (refer to Extended Data Figure 4e,f for bulk Micro-C and Extended Data Figure 5a,b for scMicro-C).

In response to your valuable suggestion, we have incorporated a zoomed-in short-range contact probability versus genomic distance curve ($P(s)$) in the revised figure.

- The authors should modify the sections from line 43 and line 71 to better reflect this advantage of Micro-C (see Fig. S1g).

We thank the Reviewer for this very helpful suggestion. We have modified this section in this revision.

“Compared with the original procedure, we consistently observed that SDS treatment greatly improves ligation efficiency, regardless of chromatin digestion efficiency (Fig. 1b and Extended Data Fig. 1a).”

“Systematic analysis demonstrated that with increasing MNase concentration, the single-cell profiles exhibited enhanced cost-effectiveness, evidenced by a higher percentage of reads containing contacts (referred as contact ratio) and a greater number of unique contacts at the same sequencing depths (Fig. 1e-f and Extended Data Fig. 2c).”

“And higher MNase digestion levels led to a slightly higher proportion of inter-chromosomal contacts (Extended Data Fig. 2b).”

- The authors compare their modified bulk Micro-C protocol from GM12878 cells to matching bulk Hi-C data (from line 64). Confirming previous comparisons, their modified Micro-C indeed reports more loops and stripes as compared to Hi-C. A better validation that the modifications do not compromise data quality would be to include data from conventional Micro-C as well. Although such data does not appear available for GM12878 cells, the inclusion of data from cell types that were sequenced at similar depths should be informative. Data from human cells is available in Krietenstein et al., or from the 4Dnucleome portal.

We genuinely appreciate the Reviewer's insightful suggestion. In response, we would like to clarify that our bulk Micro-C data comprises two replicates:

- Replicate 1 was generated using the modified bulk Micro-C protocol, where all ligated fragments were utilized for downstream library preparation via sonication.

- Replicate 2 was generated using the original bulk Micro-C protocol, where only ligated di-nucleosome sized fragments were selected for sequencing.

This additional detail has been included in the revised methods section. Notably, the comparison between replicate 1 and replicate 2 reveals no significant difference in chromatin structure (refer to Extended Data Figure 4).

- Supplemental figures 1 and 2 are discussed out of order: Fig. S2 is discussed before Fig. S1e-g. I have not found any reference to Fig. S1c,d.

We apologize for any confusion caused. In our revised manuscript and figures, we have rectified the order of citation to ensure that supplemental figures are referenced in the correct sequence.

R1.2: Next, the authors discuss the visibility of TELs that cover long and active genes in bulk Micro-C data.

- Line 89: the definition of “long” is only provided in line 109. The authors should mention here what they consider long. For a better understanding, it will also help if the authors mention how many “long” genes exist, including the fraction that is active in GM12878 cells.

We appreciate the Reviewer's suggestion. In response to your query, in the human genome, there are 9757 genes that exceed 50 kb in length, with 4077 of these genes being expressed in GM12878 cells. Furthermore, in the revised manuscript, we have identified 819 genes exhibiting the ‘promoter stripe’ characteristic. This information has been incorporated into the revised manuscript (page 8, line 241).

- An in-depth characterization of TELs and their link to long genes has not been reported. Yet, other studies beyond those mentioned from line 100 have observed similar loops: the first observation was made using 3C in yeast by the Proudfoot lab: Tan-Wong et al., Science 2012. Moreover, gene loops were highlighted by Hsieh et al., Mol Cell 2020 in their Micro-C results as well (see Fig. 4a).

We thank the Reviewer for bringing this to our attention. In response, we have made revisions to our manuscript. Instead of focusing on gene loops between the transcription start site and transcription termination site, we have shifted our emphasis to the ‘promoter stripe’ structure, which connects the promoter with multiple enhancers distributed along the stripe.

Specifically, we have included citations referencing previous reports on stripe structures associated with promoters, such as ‘promoter-associated stripe’ (Cheng, N. et al. eLife 2022), ‘promoter-anchored stripes’ (Chen, L.-F. et al. Molecular Cell 2023), and ‘gene stripes’ (Hsieh et al., Mol Cell 2020) in the revised manuscript (Page 7, lines 221–2226). Thank you for your valuable feedback.

- Line 97 and 112: I don’t think that the references to Fig. S1f and Table S1 are correct.

We apologize for the oversight regarding the references to figures and supplementary tables. In the revised manuscript, we have taken care to ensure the accuracy of all references. Thank you for bringing this to our attention.

Figure 2.

R2.1: After introducing scMicro-C and TELs in GM12878 cells, the manuscript switches gears to address the potential mechanisms of TEL formation. First the authors focus on the reanalysis of Rad21 and WAPL degra Micro-C data from mouse ES cells.

- Two recent studies in Nature Genetics have addressed the link between transcription, 3D genome organization and loop extrusion using Micro-C based studies (Zhang et al., 2023 and Barshad et al., 2023). Some of their insights are highly relevant for the interpretation of the results in Fig. 2 (e.g. from line 115 and line 147). The authors should critically revisit this part and, where necessary, incorporate findings from these two studies.

We sincerely appreciate the Reviewer's diligent examination of the literature and valuable feedback. We acknowledge the relevance of the studies mentioned in Nature Genetics provide valuable insight into the relationship between transcription, 3D genome organization and loop extrusion, notably emphasizing the roles of RNAPII and Mediator in enhancer-promoter interactions.

In consideration of the manuscript's length constraints, we have opted not to delve into the mechanistic regulation of the 'promoter stripe' and have consequently removed the sections related to RNAPII and transcription inhibition. Our focus remains on exploring the dependency of the 'promoter stripe' on cohesin-mediated loop extrusion.

Moving forward, we are committed to further investigating the mechanistic regulation of the 'promoter stripe.' We plan to integrate findings from RNAPII depletion and topoisomerase perturbation results to elucidate the intricate interplay between transcription, supercoiling, and loop extrusion. Moreover, we will ensure to cite the two referenced papers in our forthcoming work.

- I find the paragraph from line 132 difficult to follow. How does this increased signal at the TTS provide evidence about Cohesin unloading time? What is the barrier that is mentioned in line 136? Why does the increased signal upon WAPL depletion suggest nonetheless that this process is not WAPL dependent (line 144)? Instead, could the accumulation, and possibly loading of Cohesin at promoters carrying paused PolII play a role in this process (see Barshad et al., Nat Genet 2023)?

We apologize for any confusion caused by unclear writing in the previous version of the manuscript. In the revised version, we have omitted the discussion concerning the interaction pattern between the broad region encompassing the transcription termination site (TTS) and the upstream regions of the transcription start site (TSS).

We acknowledge that further conclusive data is necessary to fully understand the mechanistic regulation of 'promoter stripes', particularly concerning the identity of cohesin barriers at the

promoter and the regulation of cohesin unloading at the TTS. Therefore, we have removed the discussion about the mechanistic regulation of ‘promoter stripes’, and instead focus solely on its dependence on cohesin-mediated loop extrusion.

It’s possible that the cohesin barrier at the promoter is related to transcription-related proteins, as evidenced by reduced cohesin binding at the E-P anchors upon RNAPII depletion (Zhang et al., Nature Genetics 2023) or Mediator depletion (Barshad et al., Nat Genet 2023). Additionally, our own data indicates a significant reduction in cohesin binding at the promoter regions upon RNAPII depletion (original Fig. 2d).

- Line 127: it should be mentioned that this result refers to Fig. 2a. Are these long highly transcribed genes, or all highly transcribed genes?

We apologize for the oversight in not specifying the reference to Fig. 2a. In the original manuscript, we categorized genes into three groups based on their expression levels, focusing solely on genes longer than 50 kb. However, in the revised manuscript, we conducted a *de novo* calling of the ‘promoter stripe’ structure from the contact maps, revealing that this pattern is specifically associated with ‘stripy genes.’

- Line 129: signals are reduced (to 1.13), but not completely eliminated. This should be rephrased.

We thank the Reviewer for this suggestion. We have changed the word from ‘completely eliminated’ to ‘significantly reduced.’

- Line 130: these correlative results show that TEL formation requires active loop extrusion, but they do not directly confirm that TELs are formed by loop extrusion.

We thank the Reviewer for this comment. We completely agree with the Reviewer that these results do not directly confirm this. We have toned down this statement.

- For all Micro-C pile-ups (similar to Fig. 1e), it should be indicated that the tick marks represent TSS to TTS. The mention of 1 kb is not useful, because the span that is covered differs from gene to gene.

We thank the Reviewer for this suggestion. In response, we have included labels for ‘TSS’ and ‘TTS’ in the revised pile-up figures. Regarding the resolution, we acknowledge that although the gene length is rescaled in the pile-up results to maintain consistency, it’s essential to specify the resolution of the contact matrix used.

R2.2: Next, the authors generate Micro-C data in RBP1-degron cells, further complemented by reanalysis of Micro-C data upon transcriptional inhibition.

- Line 165: this result is very similar to the result reported in Zhang et al., Nat Genet 2023. Does this analysis include all active promoters, or only those of long genes?

We thank the Reviewer for this comment. First of all, we have removed the RNAPII depletion Micro-C results from the updated manuscript.

We appreciate the reference to the RNAPII depletion Micro-C paper from Akis group, which explores the role of RNAPII in mediating enhancer-promoter loops. While we did refer to their data for the analysis of topologically associating domains (TELs), we found that the pile-up signal from their data is weaker and noisier compared to our own. Thus, we decided to generate our own RNAPII depletion Micro-C data. Nevertheless, the trend observed in their data is consistent with ours, indicating that RNAPII depletion weakens TELs.

- The section from line 167 about R-loops as barriers for loop extrusion is extremely speculative, with very little proof beyond a weak correlation between gene expression and R-loop signal (Fig. S9d,e). As it contributes little to the study, this should be removed. From line 172, it is discussed that RNAPII depletion has a less prominent impact than transcription inhibition. How is this determined? Visual comparison between Fig. 2c and S9b? If so, the authors should include a direct comparison here. Of note, the authors should keep in mind that the Auxin system is leaky, and that PolII levels in untreated RBP1-AID cells may already be considerably reduced. From line 173: the authors should abstain from making claims about degradation efficiency in different cell lines and different studies/laboratories, unless they have access to the material themselves.

We sincerely appreciated the Reviewer for this insightful suggestion. In response, we have removed the speculative discussion regarding R-loops from the revised manuscript.

Regarding the comparison between RNAPII depletion and transcription inhibition, the conclusion drawn about the less prominent effect of RNAPII depletion is based on the observed reduction in

values compared to the control. Direct comparison between two different datasets is challenging as the values are calculated based on observed versus expected values.

The claim regarding RNAPII depletion efficiency is inferred from the levels of chromatin-bound RNAPII. The concern about insufficient degradation efficiency was also raised by Reviewer 1 (Akis), who observed that less than 80% depletion did not lead to significant changes in 3D genome structure. Consequently, prolonged auxin treatment (such as 14 hours in their manuscript) may be necessary to achieve more thorough RNAPII degradation and observe a stronger effect on TELs.

R2.3: In this last part, the authors generate Micro-C data upon inhibition of Topoisomerase 1 and 2. Here they find that particularly inhibition of TOP2 reduces TELs.

- In line 218, it's mentioned that TELs are slightly strengthened upon TOP1 inhibition, yet no difference can be observed in Fig. 2f. In contrast, such a difference is observed in Fig. S11e. Is this because the DMSO signal in Fig. 2f is the average (or combination) of the result from the two rather different DMSO experiments in Fig. S11e and S11f? If yes, the authors should consider if it's correct to merge these control experiments. Conversely, this also shows that TEL signal in Micro-C experiments may be prone to experimental variation, with the Micro-C signal for downregulated genes upon TOP2 inhibition being mostly indiscernible from DMSO treatment in the TOP1 inhibition experiment. The authors should discuss this.

We appreciate the insightful comment provided by the Reviewer. In response, we have made significant revisions to the manuscript, focusing it into a Technical Report by removing all data related to topoisomerase inhibition.

We apologize for any confusion arising from the unclear description of the data. It's important to clarify that we generated three Micro-C datasets for topoisomerase inhibition: DMSO, TOP1 Inhibition (TPT), and TOP2 Inhibition (ETO), all within a single batch. Contrary to the assumption, we did not generate separate DMSO Micro-C datasets for TPT and ETO experiments.

The discrepancy between Fig. 2f and Extended Data Fig. 11e-f arises from the utilization of different gene sets, which were determined by the specific genes affected by topoisomerase inhibition (TOP1 or TOP2). Specifically, Fig. 2f use genes affected both by TOP1 and TOP2 inhibition, Extended Data Fig. 11e use genes affected by TOP1 inhibition and Data Fig. 11f use genes affected by TOP2 inhibition.

- Line 225: did the authors use a statistical test to determine the significance of this difference? If not, they should abstain from making this claim. Considering the rather weak reduction of Cohesin in the gene body upon TOP2 inhibition, I'm not convinced that this is sufficient to explain the rather drastic reduction of TELs at downregulated genes. The claim of cooperativity between cohesin, RNAPII and topoisomerase (line 229) remains therefore based in weak correlations. I consider it at least as likely that the effect of topoisomerase on cohesin is indirect, mediated solely by its effect on RNAPII activity at downregulated genes.

We thank the Reviewer for this comment. We did not use any statistical test. We have removed this claim. We agree with the Reviewer that the effect of topoisomerase on cohesin may be indirect. However, the cooperativity between cohesin, RNAPII and topoisomerase remains unclear and warrant further investigations. Since cohesin directly interacts with topoisomerases and introduces negative supercoiling when extruding DNA (Uusküla-Reimand et al., Genome Biology 2016; Davidson et al., bioRxiv 2024; Janissen et al., bioRxiv 2024), it remains possible that cohesin is directly affected by topoisomerase inhibition.

Figure 3.

In this section, the authors return to their scMicro-C data to characterize TEL kinetics in WT cells. This is potentially a very exciting application of the scMicro-C technology, but for now this has not been used up to its full potential.

- The sorted single-cell contact maps are not easy to grasp and lack annotation. I assume that these plots only indicate contacts that involve the 5kb bin that covers the TSS? This should be explicitly mentioned. Moreover, the position of the TSS and TTS should be indicated (particularly, no indication is provided in Fig. 3a). Instead, why do the authors not use a similar visualization as done for ChIA-Drop complexes (Zheng et al., Nature 2019) or Nano-C data (Chang et al., Nat Commun 2023)?

We sincerely apologize for any confusion caused by the lack of clarity in our manuscript. Indeed, the plots presented in the manuscript specifically indicate contacts involving the 5kb bin covering the promoter. In response to your suggestion, we have now included explicit mention of this in the revised figure legend, along with the positioning of the TSS and TTS.

Regarding the visualization of sorted single-cell contact profiles, we appreciate your suggestion to consider approaches similar to those used in ChIA-Drop or Nano-C data visualization. However, it's important to note that scMicro-C data differs from ChIA-Drop or Nano-C data in terms of the nature of the interactions detected. While ChIA-Drop and Nano-C data often involve individual interacting complexes or long concatenated reads, our scMicro-C data primarily detects pairwise interactions. If we sort the scMicro-C data by reads, then we will lose the single-cell information.

- The results are mostly discussed relative to the analysis of the MSI2 gene, where a nearly linear drop in the single-cell contact maps is observed. This drop is quite different from the EPHB1 gene, where a more gradual distance-dependent reduction is observed (Fig. S13a). Why did the authors decide to not include a single-cell contact map for the Piezo2 gene? The drop in contacts for the EPHB1 gene shares similarities with both the control locus (Fig. S13g) and resembles single-allele Nano-C data for non-genic regions as well (Chang et al., Nat Commun 2023). This raises the question if the organization at the MSI2 gene is truly as representative as it is presented, or if more diverse configurations may exist (linked to transcription level?). The characterization of TELs would be much improved if the authors would systematically analyze their scMicro-C data, for instance by providing pile-up plots of single-cell contact maps for different transcription levels, and for other non-genic regions (architectural stripes, "empty" regions). For an ideal understanding,

these WT data on TEL dynamics would be compared to scMicro-C data from cells where the transcription machinery or loop extrusion is perturbed.

We thank the Reviewer for your thorough and insightful comments. In response to your feedback, we have refined the analysis of single-cell 3D genome structure, particularly focusing on the dynamic loops between the promoter and multiple downstream enhancers of 'stripy genes', as highlighted in the revised manuscript.

Though we have substantiated that 'promoter stripe' is dependent on cohesin-mediated loop extrusion, we must note that our reconstructed 3D genome structures lack simultaneous cohesin binding information as argued by other reviewers. Therefore, we have refrained from making claims about the direct reflection of loop extrusion dynamics in the observed dynamic structures. Instead, we have redirected our attention towards elucidating the functional enhancer-promoter interactions.

We have clarified the distinction between sorted scMicro-C contact profiles of 'promoter stripes' and non-stripe control regions. Specifically, 'promoter stripes' exhibit increased interaction frequency between the promoter and downstream genomic loci, as evidenced by a higher proportion of single cells capturing these interactions. Additionally, the structure of 'promoter stripes' promotes multi-loci interactions, as indicated by the abundance of sporadic interactions in the sorted scMicro-C profiles (Fig. 4b and Extended Data Fig. 11b versus Extended Data Fig. 10). We have chosen not to highlight the decay pattern in the sorted scMicro-C contact profiles, as stochastic interactions not mediated by cohesin are also captured.

In the revised manuscript, we have expanded our analysis to incorporate multiple genes, namely EBF1, IRF2, and PIEZO2, providing a broader context for our findings. Our investigation demonstrates that enhancer-promoter interactions are dynamic and heterogeneous across various genomic loci. Furthermore, we have observed that multiple enhancers tend to form "multi-enhancer hubs" within individual cells, further supporting the dynamic nature of these interactions. These conclusions are further reinforced by our pile-up analysis of all 'stripy genes.'

We appreciate your suggestion to include pile-up results of architectural stripe and 'promoter stripe' using scMicro-C data, which we have incorporated into the revised manuscript.

Regarding your suggestion to conduct perturbed experiments, we regret that we are unable to pursue these efforts due to constraints related to experimental cost and time limitations.

- The average distance matrixes resemble those that are used to visualize ORCA data (optical reconstruction of chromatin architecture; Mateo et al., Nature 2019). The authors should better explain, both in the Results and the Material & Methods sections, how these distances are determined and that this is inferred by modeling from the scMicro-C data.

We thank the Reviewer for this very helpful suggestion. In this revision, we have explained the method by adding the following paragraph to the Material & Methods section:

“For the genomic region of interest, we extracted the spatial coordinates from the 5 replicate 3D genome structures of each cell with “dip-c reg3”. In order to ensure the reconstruction consistency of this specific region, we aligned the 5 replicate structures obtained and calculated root-mean-square r.m.s.d. for each allele in every cell with low structural uncertainty with “dip-c align”. We retained only those structures with root-mean-square r.m.s.d. ≤ 1.5 . Finally, we selected one replicate structure and calculated pairwise distance matrix using the “scipy.spatial.distance.pdist” function with “dip-c rg -d”. Distance matrices derived from different alleles and cells were then concatenated and averaged.”

We have added the following sentences to the Results section:

- “Based on scMicro-C contact map, we applied our previously developed Dip-C algorithm to reconstruct the 3D genome structure of individual GM12878 cells (see methods).”
- “To assess the fidelity of the reconstructed single-cell kilobase 3D genome structures in preserving genome architecture, **we calculated the pairwise distance matrix from the reconstructed 3D genome structures** and compared it with the bulk Micro-C contact matrix.”

- Line 241: Fig. 3c is discussed prior to Fig. 3b.

We apologize for the oversight. In the revised manuscript, we ensured the figures are cited in order.

Figure 4.

In this last part, the authors continue their efforts to characterize TELs, using various scRNA-seq data to link their presence to transcriptional bursting and paused PolIII.

To streamline our response and ensure clarity, we would like to make a concise statement. In line with the recommendations provided in the technical report and the feedback from the majority of reviewers, we have decided to prioritize scMicro-C analysis. As a result, we have chosen to remove the results pertaining to transcriptional bursting from the manuscript. These findings were primarily based on bulk assays and scRNA-seq data rather than the scMicro-C analysis.

- Line 263: the authors focus their analysis on long TOP2-downregulated genes, which display strong TEL signal (Fig. 4a). A very similar enrichment of TEL signal is observed in long TOP1-downregulated genes (Fig. S11a, S15b), yet the authors have previously discussed that these TELs are not affected upon TOP1 inhibition (Fig. 2e and S11e). The evidence that TOP2 exerts its function through TEL regulation will be much stronger if the authors can confirm that there are differences between the TOP2 and TOP1 downregulated genes.

We thank the Reviewer for this comment. In this revision, we have removed all data related to topoisomerase inhibition, transcription bursting and paused RNA Pol II from the manuscript.

We would like to point out that in addition to this study, another study has also showed that TOP1 and TOP2 inhibition have similar effects on gene transcription in mouse cultured neurons (King et al., Nature 2013). We believe that the differences between the TOP1 and TOP2 downregulated

genes would be marginal. However, TOP1 and TOP2 differ in their mechanisms: TOP2 mainly operates at the promoter, whereas TOP1 primarily functions within gene body and ahead of RNA Pol II (Pommire et al., Nature Reviews Molecular Cell Biology 2022). Hence, we think it's still a plausible explanation that TOP2 exerts its function through the 3D genome structure, even though TOP1 inhibition has little impact.

- Line 265: the phrase “were significantly enriched in two categories: housekeeping genes and cell-type-specific genes” is confusing, because these two categories essentially encompass all genes. Instead, it appears as if the enriched GO categories are mostly non-related. Based on this result, I'm not convinced that the conclusion can be drawn that “genes with TEL structure are important for cell identity and function.” Instead, I'm wondering if these genes share the characteristic that they are long, but are otherwise not functionally enriched. To address this question, the authors should compare their results to similar GO analyses with all long genes (expressed or not) and with downregulated long genes upon TOP1 inhibition.

We express our gratitude to the Reviewer for this insightful suggestion. We have performed the GO analyses for EOT- or TPT-downregulated genes and all long genes (> 50kb, whether expressed or not), as suggested by the Reviewer. Specifically, in the GM12878 cell line, among the top 30 enriched terms, 5 terms were shared between TPT-downregulated genes and all long genes, while only 1 term was shared between ETO-downregulated genes and all long genes. In the mESC cell line, 8 terms were shared between TPT-downregulated genes and all long genes, whereas 9 terms were shared between ETO-downregulated genes and all long genes. These results demonstrate that there indeed some common shared terms with all expressed long genes, and we agree that we should tune down the functional enrichment claim about the ETO- or TPT- downregulated genes.

ETO downregulated genes

TPT downregulated genes

All long genes (> 50kb, 9582)

Top 30 enriched GO terms (GM12878)

Top 30 enriched GO terms (mESC)

- Line 275: burst kinetics are not directly measured, but rather indirectly inferred. The authors should tone down this statement.

We thank the Reviewer for this suggestion. We have removed all statements related to burst kinetics.

- The results in the section from line 282 raise the question how these differences relate to genes without TELs but that are nonetheless downregulated by TOP2 inhibition. In a similar vein, how are burst kinetics affected for genes with TELs upon TOP1 inhibition?

We thank the Reviewer for raising this question. We observed that those short genes downregulated by TOP2 inhibition did not exhibit increased burst frequencies, compared to short genes that are unchanged by TOP2 inhibition. Furthermore, those short genes downregulated by TOP1 inhibition displayed even lower burst frequencies. Given that longer genes are more likely to have this structure, we think burst kinetics may be associated with chromatin structure.

In this revision, we have removed all data related to topoisomerase inhibition and abstained from making any claim about transcription bursting, as suggested by the Reviewers.

- Section from line 300: I have problems interpreting these results. The authors claim that upon Rad21 degradation, particularly long genes are enriched that are downregulated and display a reduction in burst frequency. Yet, according to Fig. S17h, around 40% of downregulated genes, independent on length, are associated with a reduced burst frequency (blue bars at left bottom). Rad21 depletion therefore may have a stronger effect on the activity of long genes that are sensitive to TOP2 inhibition, but this appears independent from burst frequency.

We thank the Reviewer for raising these questions. First of all, we would like to clarify that we have removed the burst frequency results from the revised manuscript.

Indeed, about 40% of downregulated genes, independent on length, are associated with a reduced burst frequency after cohesin depletion (blue bars at left bottom). However, for genes with reduced bursting frequency, longer genes are more sensitive to TOP2 inhibition (blue bars at top right). We think one potential explanation is that some downregulated genes are insensitive to cohesin depletion, while longer genes that are sensitive to cohesin depletion are also sensitive to TOP2 inhibition.

- Section from line 309: the conclusion that cohesin facilitates the release of RNAPII pausing, thereby ensuring high frequency bursting is based on a single correlative analysis and appears detached from the rest of the study. The authors should either remove it or analyse this aspect in more detail, for instance by reanalysis of Micro-C data upon perturbation of RNAPII pausing (Barshad et al., Nat Genet 2023).

We sincerely appreciate the Reviewer's valuable suggestion. In light of this feedback, we have decided to remove the conclusion regarding cohesin's role in facilitating the release of RNAPII pausing from the manuscript.

General comments:

- The manuscript makes many switches between results from human GM12878 cells and mouse ES cells, including a number of back-and-forths between figures and supplemental figures. To facilitate

the interpretation, the authors should clearly mention for each figure/figure panel if it concerns human GM12878 cells or mouse ES cells.

We thank the Reviewer for this suggestion. We have added all necessary indications to avoid any confusion.

- Many sentences contain grammatical inconsistencies. The manuscript could benefit from improved proof-reading.

We sincerely appreciate the Reviewer's constructive feedback regarding the grammatical inconsistencies in our manuscript. We have extensively revised the manuscript and ensured that the manuscript undergoes thorough proofreading to address these issues effectively.

Reviewer #4:

Remarks to the Author:

In this manuscript, Wu and colleagues present a single-cell Micro-C (scMicro-C) approach. In addition, they use bulk Micro-C data to analyze patterns of genome folding at long, active genes. They find that such genes are characterized by “stripes” that span from the TSS to the TTS. They call these structures “Transcription Elongation Loops” (TELs). Using a combination of available and newly generated bulk Micro-C data, the authors show that these TELs are reduced upon depletion of Cohesin and RNA polymerase II and after treatment with inhibitors of transcription and topoisomerases. In addition, the authors use their scMicro-C data to show intermediate folding stages of TELs. Finally, the authors examine the relationship between TELs and transcription, including bursting and pausing.

This manuscript contains some innovative and interesting aspects. However, the rationale for and interpretation of the experiments is not always clear. In addition, some of the conclusions are overstated and not fully supported by the data.

We sincerely appreciate the Reviewers for their thorough reviewing of our manuscript. Your insightful comments and suggestions have been immensely valuable in refining our work.

Before addressing the specific concerns raised, we'd like to highlight some key updates made in response to previous feedback. In our revised manuscript, we have extensively reorganized both the text and figures. Notably, we have dedicated more attention to the analysis of single-cell Micro-C data. In response to the suggestions provided by most reviewers, we performed more thorough validation and benchmark of our scMicro-C technique, increasing the cell count to over 800 and meticulously validating aspects such as SDS treatment and MNase titration effects on scMicro-C contact profiles.

In terms of scientific illustration, we have made substantial changes. We have removed much of the previous bulk analysis related to transcription elongation loops, including results and models concerning RNA polymerase II and topoisomerase perturbations, as well as transcriptional bursting. To address concerns regarding experimental support for certain aspects of the transcription elongation loop model, we have reframed our terminology. Specifically, we now refer to the original transcription elongation loops as ‘promoter stripes’ and the associated genes as ‘stripy genes.’ We have refocused our narrative on exploring the regulatory role of ‘promoter stripe.’ Our revised findings highlight that ‘stripy genes’ typically feature multiple enhancers distributed along the ‘promoter stripe,’ with cohesin-mediated loop extrusion facilitating their interaction.

Furthermore, our single-cell Micro-C analysis has unveiled dynamic looping events between promoters and multiple enhancers of ‘stripy genes.’ We have also identified a novel spatial structure termed the ‘multi-enhancer hub,’ wherein multiple enhancers cooperatively interact to form clusters associating with the promoter in individual cells.

For a more comprehensive understanding of these revisions, we kindly invite you to review our updated manuscript and figures. We genuinely appreciate your feedback and eagerly await any additional insights you may offer. Your contributions are crucial in enhancing both the quality and significance of our research.

Major comments:

1. The development of a scMicro-C protocol is of potential interest to the field. However, since the resolution is 5 kb, it is unclear whether the MNase digestion is really beneficial, as this could in principle be achieved by DpnII digestion as well, which would allow for much more efficient ligation (of cohesive ends). It would be helpful if the authors could clarify the benefits of a scMicro-C procedure over the existing scHi-C procedures. It is a bit unfortunate that the authors start the paper by introducing their scMicro-C procedure, but discover and characterize the TELs based on bulk Micro-C data. The analysis of the intermediate TEL stages in Fig. 3 does rely on single-cell data, but it seems that this could have been achieved with existing scHi-C approaches as well. The unique benefit of the presented sc-Micro-C approach is therefore not clear. The authors mention higher cost-effectiveness and signal-to-noise ratio of scMicro-C compared to scHi-C. This indeed seems to be the case and may be useful for the field, although these improvements do not seem striking to me. If this is the only benefit of scMicro-C, it would be good if the authors could clarify this.

We thank the Reviewers for this valuable suggestion. We appreciate the opportunity to address your comments and clarify several key points regarding our scMicro-C protocol and its advantages over existing scHi-C procedures.

Firstly, we want to clarify that the 5 kb resolution mentioned in our manuscript pertains specifically to the reconstruction of single-cell 3D genome structures rather than contact maps. Our updated ensemble scMicro-C profiles, comprising 750 million unique contacts, enable the detection of chromatin interaction patterns at resolutions as fine as 2 kb or even 1 kb (refer to Fig. 1j and Extended Data Fig. 5c). Although sparse at these resolutions, it's important to note that the resolution of scMicro-C is not limited to 5 kb. Theoretically, at comparable number of contacts, there is no distinction of resolution between bulk Micro-C and scMicro-C.

In comparison with our previous state-of-the-art scHi-C method (Dip-C), scMicro-C offers distinct advantages:

- Higher Signal-to-Noise Ratio: scMicro-C demonstrates superior signal-to-noise ratio compared to scHi-C and bulk Hi-C, particularly evident in chromatin loop and stripe detection (see Fig. a-e).
- Retention of Nucleosome Occupancy Information: scMicro-C effectively retains nucleosome occupancy information (Fig. f-g), which can be utilized to infer transcription factor (TF) activity heterogeneity across individual cells. Although not elaborated in this manuscript, this

analysis allows us to deduce TF activity based on nucleosome occupancy around specific TF binding motifs.

- **Cost-Effectiveness:** scMicro-C exhibits higher cost-effectiveness as evidenced by a significantly greater percentage of reads containing contacts, translating into a larger number of unique contacts at equivalent sequencing depths (refer to Fig. h-i).

Furthermore, while we initially detected the 'promoter stripe' pattern using bulk Micro-C or ensemble scMicro-C data, the high-resolution single-cell 3D genome structures generated by scMicro-C have enabled detailed investigations into the folding dynamics and heterogeneity of 'promoter stripe' configurations among individual cells. Notably, scMicro-C has facilitated the identification of spatial clusters formed by multiple enhancers within individual structures, a level of detail that cannot be discerned through bulk Micro-C studies alone.

We acknowledge that certain deeply sequenced Dip-C (scHi-C) cells can achieve 5 kb resolution in 3D genome structure reconstruction. However, it's important to highlight that their analysis did not explore 3D genome structure at this resolution due to limitations in cell numbers (n = 17).

2. The development of the scMicro-C procedure would benefit from more detailed description and benchmarking. In particular, the authors claim that the use of SDS significantly improves the ligation efficiency. However, they do not present sufficient data that allow for a direct comparison between Micro-C data generated with and without SDS. For the profiles shown in Fig. 1b, the digestion with and without SDS looks different, even though SDS is added after the digestion step. The ligation in the conditions with SDS indeed looks much better compared to the conditions without SDS, but since the digestion profile without SDS looks over-digested compared to the digestion with SDS, this could have biased these results. In addition, the rationale for adding SDS and the potential mechanism by which the addition of a detergent would improve the ligation efficiency is not clear. It would be helpful if the authors could clarify this. Furthermore, it would be great if the authors could show a detailed direct comparison of Micro-C data with and without the addition of SDS to show that the proportion of useful ligation junctions is actually improved by SDS. The profiles shown in Fig. 1b are not that helpful, since it is not possible to discern if the longer ligation product may represent fragments that have not been digested in the first place. Finally, it seems that the authors do not biotinylate the ligation junctions. Since the authors present potentially important improvements to the current Micro-C procedure, it would be very helpful if more details could be provided. (I have understood from colleagues that they have tested addition of SDS after reading the pre-print of this paper but could not reproduce the results presented in Fig. 1b).

We sincerely appreciate the valuable comments provided by the Reviewers. In response to your suggestion, we conducted additional validation experiments to thoroughly investigate the effects of SDS treatment in our scMicro-C procedure.

To address concerns about digestion efficiency, we performed a new experiment where nuclei were subjected to MNase digestion, and the resulting digested nuclei were split into two tubes. One tube followed the original Micro-C protocol, while the other underwent SDS treatment. The results clearly demonstrated that SDS treatment significantly improved ligation efficiency compared to the original protocol (digested: 537 bp; original protocol ligation: 720 bp; SDS treatment: 1628 bp) (Fig. 1b). Importantly, this improvement was consistent across different MNase digestion efficiencies (see below).

Moreover, as you suggested, we isolated single cells with and without SDS treatment for sequencing to directly assess the impact of improved ligation efficiency on chromatin contact detection. The results showed that SDS treatment detected 8.1-fold more chromatin contacts than the original Micro-C protocol, even at comparable sequencing depth (Fig. 1c). Further analysis revealed that

SDS treatment significantly increased long-range intra-chromosomal contacts (Extended Data Fig. 1b-d, 44.9% versus 62.5%). Additionally, the proportion of reads containing contacts was much higher after SDS treatment (2.9% versus 11.3% at the tested MNase concentration of 600U).

Regarding the omission of biotin enrichment for single-cell sequencing, we made this decision for two reasons. First, biotin incorporation during the end repair step reduced ligation efficiency. Second, biotin pull-down for single cells was found to be inefficient and resulted in the loss of chromatin contacts. This omission of biotin enrichment was also widely adopted in other single-cell Hi-C methods (Flyamer, I. M. et al., Nature 2018; Tan, L. et al., Science 2018; Zhou, T. et al., bioRxiv 2023).

The addition of SDS treatment was inspired by the Hi-C procedure (Belton, JM. Et al., Methods 2012), which uses SDS to deplete non-crosslinked proteins and open the chromatin for restriction enzyme (RE) digestion. Our hypothesis is that SDS treatment enhances the accessibility of the end repair enzymes to chromatin, as their molecular sizes (T4PNK: 132 kDa, Klenow fragment: 68 kDa) are larger than MNase (16.9 kDa) and REs (MboI: 60 kDa; 68 kDa), making it challenging for them to access the chromatin efficiently. The resulting improvement in end repair efficiency ultimately leads to enhanced ligation efficiency. We have included this assumption in the revised manuscript, but further experiments are needed to validate the detailed mechanism.

In our hands, the SDS treatment has been highly robust. If your colleagues encounter difficulties reproducing the SDS improvement, please feel free to reach out to us. We would be more than willing to assist with troubleshooting.

3. I find the conclusions in this paper generally a bit overstated. In the abstract, the authors state that: “We proved that TELs formation results from the joint interactions between cohesin-mediated loop extrusion, RNA polymerase II (RNAPII) and topoisomerases.” I do not think this is actually shown. The authors show that the TELs are weakened in absence of Cohesin. However, the reduction in TELs following RNAPII depletion are not convincing. The authors explain this by incomplete depletion, which indeed seems to be the case. However, for that reason, they cannot draw this conclusion at the moment and would need additional data to support it (i.e. longer depletion times or a better degron line). Furthermore, the topoisomerase inhibitors only seem to remove the TELs in genes that are downregulated after treatment. This indicates that this effect could (likely) be secondary to reduced transcription, which is in line with the presented transcription inhibition experiments. Similarly, the effect of topoisomerase inhibitors on Cohesin distribution may be secondary to reduced transcription (which has been shown previously in the context of RNAPII and Mediator depletions, see other comment #9). It would be helpful if the authors could describe their results and discussion more precisely and take these limitations into consideration.

We thank the Reviewers for these suggestions. We totally agree that the proposed TEL model and the intricate interplay between cohesin, RNAPII and topoisomerases need further experimental data to reveal. Due to space limitations in this article, we have decided not to incorporate the RNAPII and topoisomerases results. This decision was made with two purposes in mind: to emphasize that

the involvement of RNAPII and TOPO in transcriptional loops relies on bulk perturbation experiments rather than scMicro-C, and to align with the focus of this technical report, as advised by most reviewers, which emphasizes scMicro-C analysis over mechanistic studies of TELs.

Indeed, the depletion of chromatin-bound RNAPII is not efficiently degraded with 4-hour auxin treatment, prolong auxin treatment is needed to completely deplete RNAPII and observe stronger effect on TELs. This is also suggested by the other reviewers and in their own results also show that less than 70% depletion of RNAPII is not enough to lead strong 3D genome changes (Zhang et al., Nature Genetics 2023).

The effect of topoisomerase inhibition on TELs is not limited to down-regulated genes, when we group the expressed gene according to expression level, we also observed the significantly weakened of TELs after TOP2 inhibition. The effect of topoisomerase inhibition on cohesin is possible by affecting transcription. Topoisomerase is response to resolve supercoils on chromatin. Recent two preprint papers found that cohesin introduce negative supercoiling into loops during extrusion (Davidson et al., bioRxiv 2024; Janissen et al., bioRxiv 2024). Thus, it is also possible that topoisomerase effect is through by perturbing supercoiling accumulation. This speculation needs further experimental data to explore.

4. In line with major comment #3, the strong statement about the role of Cohesin in RNAPII pausing is not at all supported by data. The authors refer to Extended Data Fig. 17f, which shows a Volcano plot with a brief legend that does not mention how the pausing phenotype can be appreciated. The statements regarding to pausing either need a lot of additional experimental data to support them or should be removed entirely from the manuscript.

We deeply appreciate the Reviewer's insightful feedback. In response to their suggestion, we have decided to remove the discussion concerning the impact of cohesin on RNAPII pausing from the revised manuscript.

We apologize for any confusion caused by the unclear description in the original manuscript. The speculation regarding the role of cohesin in regulating RNAPII pausing was based on the observation of a significant overlap between genes exhibiting increased pausing (as calculated from PRO-seq data) and genes showing a reduced burst frequency upon cohesin depletion. However, we acknowledge that further experimental data are necessary to elucidate the mechanism underlying the effect of cohesin depletion on reduced bursting frequency and its relationship with RNAPII pausing.

In the revised manuscript, we have chosen to focus on the regulatory function of the 'promoter stripe' in enhancer-promoter interactions. Our findings indicate that genes associated with the 'promoter stripe' typically possess multiple enhancers distributed along the stripe, leading to focal interactions at the enhancer sites. Therefore, we propose that the 'promoter stripe' facilitates interactions between the promoter and multiple enhancers. Although we observed cohesin depletion affecting the bursting frequency, we have opted not to include this result in the revised manuscript and figures.

Other comments:

1. It would be helpful if the authors could specify what proportion of genes contain TELs, so the readers can appreciate how prevalent/important these structures are.

We thank the Reviewers for raising this point. In the revised manuscript, we have conducted de novo identification of the ‘promoter stripe’ structure. However, it's important to note that we were only able to confidently detect the ‘promoter stripe’ on the bulk Micro-C at 5 kb and 10 kb resolutions for genes longer than 50 kb. For shorter genes, the stripe pattern couldn't be reliably called. Specifically, in the GM12878 dataset, we identified 819 instances of the ‘promoter stripe’ structure, while in the mESC dataset, we identified 853 instances.

2. I find it a bit confusing that the authors decided to name the stripes that they observe at active, long genes “Transcription Elongation Loops”, since these structures represent a stripe and not a loop.

We extend our gratitude to the Reviewers for this valuable suggestion. In response to their feedback, we have made a crucial revision in the manuscript. To prevent any misleading interpretations, we have decided to rename the observed structures from ‘transcription elongation loops’ to ‘promoter stripe.’ This renaming aims to avoid any misinterpretation that these dynamic loops are solely formed through RNAPII-mediated transcription elongation.

It is important to note that the chromatin stripe comprises a dynamic collection of chromatin loops with varying sizes between the two anchor points of the stripe.

3. Lines 58-60: “To achieve scMicro-C, we implemented three key improvements (Fig. 1a, Methods). Firstly, we optimized MNase titration to prevent over-digestion and reduce DNA loss (Extended Data Fig. 1a,b).” The authors present this as a new discovery, even though this has already been described in detail previously (<https://pubmed.ncbi.nlm.nih.gov/34108683/>).

We sincerely apologize for the oversight and the lack of acknowledgment of the relevant citation. Thank you for bringing this to our attention. In the revised manuscript, we have rectified this error by including the citation (see page 4, lines 106–108) to the previously described work on the over-digestion issue in MCC. Additionally, we have conducted a comprehensive analysis, systematically testing the impact of MNase digestion levels on chromatin structure and contact detection in scMicro-C, which is now included in the revised manuscript.

4. Extended Data Fig. 2E: The Hi-C specific loops are equally strong in the Hi-C and Micro-C panels. The same is true for the scMicro-C specific loops in the scMicro-C and Micro-C panels in Extended Data Fig. 3C. It therefore seems that the analysis/detection of these “specific” loops is not very robust.

We sincerely appreciate the Reviewers for highlighting this concern. Recognizing the need for refinement in aggregation plot, we have undertaken significant revisions to rectify this issue.

Specifically, we have modified the aggregation plot of chromatin loops by adjusting the flanking window size from 200 kb to 40 kb, which is a commonly used window size in the published literature. The revised figures now present updated pile-up results that clearly demonstrate the corresponding trends with improved clarity and robustness.

5. Lines 120-122: “Secondly, NIPBL, a cohesin loading factor, is enriched at promoters of active genes, which indicates that cohesin prefers to load at TSSs.” It has been shown that NIPBL is not (only) a loader, but travels with Cohesin as it is extruding (<https://pubmed.ncbi.nlm.nih.gov/31753851/>). Furthermore, NIPBL is not enriched at promoters (<https://pubmed.ncbi.nlm.nih.gov/36897969/>; this paper is cited in the manuscript but not in this context). This statement and the implications of it throughout the manuscript should be modified accordingly.

We thank the Reviewers for this comment. We have addressed it by removing the specific text from the revised manuscript.

In the paper by Banigan et al. from the Mirny lab, they proposed that cohesin loads uniformly at all genomic loci based on their NIPBL ChIP-seq findings, which showed limited overlap with active transcription start sites (TSSs). This finding has caused confusion because the antibody they used is the same as the one mentioned in the literature (Zuin et al., PLOS Genetics 2014), which claims that over 1/5 of NIPBL ChIP peaks are located at active TSSs. We are perplexed as to why the same antibody generates conflicting results. Therefore, we believe that further confirmation of the NIPBL ChIP-seq results is necessary.

It is important to note that whether cohesin tends to load at active DNA elements remain subjects of ongoing debate and warrant further experimental validation. Several observations, such as the formation of chromatin jets (Guo et al., Molecular Cell 2022) and chromatin fountains (Galitsyna et al., bioRxiv 2023 (from Mirny’s lab); Isiaka et al., bioRxiv 2023; Kim et al., bioRxiv 2023), suggest cohesin specifically loads at active enhancer sites. Additional evidence, such as the effects of enhancer insertion and deletion on cohesin occupancy at neighboring CTCF-binding sites (Rinzema et al., Nat. Struct. Mol. Biol. 2022; Vos et al., Molecular Cell 2021), further support the notion of active regulatory elements recruitment of cohesin. The bias distribution of NIPBL and enhancers towards the anchor side also contributes to this discussion (Vian et al., Cell 2018). Taken together, the binding of NIPBL at active promoters and the loading of cohesin at active DNA elements are subjects of extensive debate and require further experimental validation.

Therefore, in the revised manuscript, we focused on demonstrating the dependence of ‘promoter stripes’ on cohesin-mediated loop extrusion, without delving into the discussion on cohesin loading.

6. Lines 143-146: “The augmentation of interaction between the TTS and TSS upstream region suggests that cohesin stalling near TTS is independent on WAPL. These findings further confirm that both characteristics surrounding highly expressed genes are dependent on cohesin and regulated by its unloading factor WAPL.” I do not understand this statement: the interactions are independent of WAPL but also regulated by WAPL?

We apologize for the confusion caused by the statement. It was indeed a typo, and it should have stated "dependent" instead of "independent". However, we would like to clarify that in the revised manuscript, we have chosen not to include the results related to RNAPII. Therefore, we have removed the mentioned description accordingly. Thank you for bringing this to our attention.

7. The loading controls of the western blots in Fig. 2b are highly variable. It would be good if the authors could repeat this blot with proper loading controls.

We appreciate the Reviewers' suggestion. We understand the importance of adjusting the loading controls in the western blot results to ensure clarity regarding the depletion efficiency. However, as we have decided to exclude RNAPII-related results from the revised manuscript, the western blot pertaining to RNAPII depletion has been removed accordingly.

8. The authors do not refer to the cell cycle analysis presented in Extended Data Fig. 10 and it is not clear what the purpose of this analysis is.

We thank the Reviewers for this comment. The cell cycle analysis presented in Extended Data Fig. 10 aimed to demonstrate that topoisomerase inhibition does not substantially alter the cell cycle composition. This was important to clarify that the observed changes in TEL were not attributed to cell cycle effects. However, to enhance clarity and validity, we have decided to remove the results related to topoisomerase inhibition from the revised manuscript.

9. Lines 164-167: "To further investigate how RNAPII affects TELs, we assessed cohesin occupancy by performing Rad21 CUT&Tag after RNAPII degradation. Our analysis revealed a noteworthy reduction, to approximately half the previous level, in cohesin binding at active promoter regions (Fig. 2d)." The authors present this as a new finding, but this has been described before in the context of perturbations that affect transcription perturbation, both after RNAPII and Mediator depletion (<https://pubmed.ncbi.nlm.nih.gov/37012454/> & <https://pubmed.ncbi.nlm.nih.gov/37430065/>).

We apologize for overlooking these prior works. Indeed, the reduced cohesin binding at enhancer-promoter loop anchors and Mediator binding sites has been previously reported in the context of transcription perturbations, including RNAPII and Mediator depletion (Zhang et al. Nature Genetics 2023; Ramasamy et al. Nat. Struct. Mol. Biol. 2023). In the revised manuscript, we have excluded the results related to RNAPII depletion. In future work, we plan to integrate the bulk RNAPII and topoisomerase perturbation findings into another paper, focusing on a systematic mechanistic study of 'promoter stripes.' Rest assured, we will cite the mentioned papers accordingly.

10. Fig. 3 and associated Extended Data Figures: please show CTCF annotation.

We appreciate the Reviewers' suggestion. In response, we have included CTCF, RAD21, H3K27ac, and RNAPII ChIP-seq tracks above the relevant contact maps in Fig. 3 and associated Extended Data Figures. Please review the revised figures to verify these additions.

11. Lines 262-268: “Using RNA-seq data generated from GM12878 and mESC upon topoisomerase inhibition (Extended Data Fig. 15a-d), we found that TOP2 inhibition-downregulated genes, exhibiting strong TEL signal (Fig. 4a and Extended Data Fig. 11b), were significantly enriched in two categories: housekeeping genes and cell-type-specific genes (immune-related and development-related for GM12878 and mESC, respectively) (Fig. 4b, Extended Data Fig. 10h and 15e).” This statement does not seem very meaningful to me, as the expressed genes in a particular cell type can generally be divided in housekeeping and cell-type-specific genes. If both these categories are enriched, that indicates that there is no particular signature.

We thank the Reviewers' comment and feedback. In response, we have decided to remove the results pertaining to bulk RNA-seq followed by topoisomerase inhibition from the revised manuscript.

Regarding the function enrichment analysis, we acknowledge that our previous claim regarding the downregulated genes enriched in two categories—housekeeping genes and cell-type-specific genes—following TOP1 or TOP2 inhibition was inaccurate. Though these genes did exhibit significant enrichment in both housekeeping and cell-type-specific terms.

12. Lines 283-285: “We found that genes with TELs that are downregulated by TOP2 inhibition have higher burst frequencies than other genes (Fig. 4d, left and Extended Data Fig. 16g-i).” This is potentially interesting. However, I wonder if this could also simply reflect that these TELs are generally very highly expressed. Can the authors distinguish if this is due to the presence of TELs or reflects the expression levels? If not, this statement should be modified accordingly.

We greatly appreciate the Reviewers for this comment. We acknowledge the strong correlation between expression level and burst frequency, distinguishing between the presence of TELs and expression level proves challenging due to their substantial overlap. However, we have opted to exclude the transcriptional bursting results from the updated manuscript to address any potential ambiguity.

13. Lines: 297-299 “Notably, we also observed an increase in burst size among genes with reduced burst frequency (Fig. 4e right panel), which could potentially account for the modest impact on total gene expression after acute cohesin depletion.” Please clarify the colour scheme and legend of Fig. 4e to support this statement.

We appreciate the reviewer's suggestion and feedback. Again, we would like to clarify that we have removed the bursting-related results from the updated manuscript.

Regarding the clarification of the color scheme and legend of Fig. 4e, in the original figure, blue dots represent genes with significantly reduced burst frequency following 6-hour cohesin depletion, yellow dots represent genes with significantly increased burst frequency after 6-hour cohesin depletion, and green dots indicate genes without significant change. It's important to note that the

color scheme remains consistent across both the burst frequency and burst size plots. We hope this explanation addresses the concerns raised.

Reviewer #1:

Remarks to the Author:

First of all, I want to stress that the authors received and dealt with criticisms (from all reviewers) very seriously. At the same time, my appreciation of the quality of scMicro-C (esp. compared to Dip-C) still stands. However, this revised manuscript has essentially done away with all perturbations and mechanistic investigations they attempted in the original version. Instead, they now provide a more descriptive approach to the Micro-C stripes they observe linked to active genes. This limits the overall enthusiasm, as these observations are not entirely new and their characterization remains pretty much on the descriptive side. For example, multi-enhancer/-promoter hubs have been already proposed via other approaches (e.g., via Tri-C). Moreover, the manuscript now lacks any complementary approach to study these structures (the authors talk about the potential of imaging in their Discussion, but it might have been better if they added such work already), while, much like in the original manuscript, most of these observations rely on ensemble scMicro-C data rather than really single-cell-derived ones. As a result, the technical advance is appreciated, but the conceptual contribution is, in my opinion, now markedly more limited than what it was in the original publication (with all its caveats) and as such, I would see this manuscript as a better fit for a Methods-oriented journal.

We sincerely thank the Reviewers for the comprehensive comments, especially their appreciation of our scMicro-C method.

We acknowledged that we have removed most of the mechanistic investigation of the “promoter stripe,” which largely dependent on bulk analysis instead of derived from scMicro-C observation, as suggested by most of the reviewers in the first-round review to align with this technical report.

We apologize for the incomplete citation of previous multi-enhancer/promoter hubs reference, we have included both the Tri-C and MC-4C paper in our updated manuscript (Page 11, line 345-347). We totally acknowledge that the “multi-enhancer hub” has been proposed in previous literature, we have referred in our manuscript. However, previous studies are limited to a few genes (e.g., α/β -globin gene locus, ETS).

We would like to highlight that it's our first report that links the "promoter stripe" to enhancer-promoter interactions, we found that genes harboring "promoter stripe" tends to have multiple enhancers distributed along the stripe. And furthermore, we established the relationship between enhancer-promoter interactions and cohesin-mediated loop extrusion in this case. According to our updated analysis, we showed that enhancer-promoter interactions of "stripy genes" were more significantly affected upon cohesin deletion than non-stripy genes (see revised manuscript Page 8, line 281-285 and Extended Data Fig. 10I).

We expand the "regulatory hub" concept that involves the multi-way interactions between enhancers and promoter to most of the "stripy genes." Furthermore, though the characterization of "promoter stripe" is dependent on bulk Micro-C/ChIP-seq datasets, we would like to argue that the observation of dynamic enhancer-promoter loops and the multi-enhancer hub structure could only be readily captured by scMicro-C at single-cell and single-allele resolution instead of bulk analysis.

We greatly appreciated the Reviewers for their valuable suggestions to use complementary approach to validate our observations. We regret that we were unable to employ the multiplexed DNA FISH technique to confirm the multi-enhancer hub structure due to constraints of time and technical limitations. Drawing inspiration from your recent work, we utilized computational modeling to extend mechanistic exploration our experimental findings. Specifically, we employed polymer simulations to explore the mechanism underlying the interconnected multi-enhancer network at the EBF1 locus in stripy genes. The simulation results suggest that a combination of strong molecular affinities between promoters and enhancers, along with cohesin capture by enhancers, most accurately reflects our experimental observations. Notably, the single chromatin fiber simulations closely resemble our scMicro-C results, including the observed multi-enhancer hub structures. For a detailed description, please refer to the newly added section, "**Polymer simulations recapitulate multi-enhancer hub structure of stripy genes,**" and Extended Data Figs. 14 and 15.

For your convenient review, we have highlighted all changes in the revised manuscript.

Reviewer #2:

Remarks to the Author:

The authors have submitted a revised manuscript that is substantially different from the original manuscript. The revised manuscript now reads more like a methods paper. On the positive side, I think the revised manuscript makes much better use of the single-cell Micro-C data and I think some of the poorly substantiated claims related to transcriptional bursting has been removed. I also appreciate the submission of a protocol.

The authors have responded to my comments. Maybe because of a misunderstanding, they did not address my “define resolution” comment, so I have tried to explain what I meant more clearly.

My other main concern was that the original manuscript overstated novelty and undercited the prior literature. I think this has been improved, but I still think the authors undercite and overstate things and it would be nice if they could tone this down a bit.

Overall, in my view this is a very nice contribution and the number of single-cell contacts they achieve is impressive and I think this paper will be of wide interest to the field and receive significant attention and as such I fully support the eventual publication. In my view, leaving aside multi-omics, this paper represents the state-of-the-art of single-cell 3D genomics. I just wish the authors would tone down a bit with some of the overclaiming and present things in a more nuanced and balanced fashion.

We sincerely appreciate the Reviewers’ enthusiasm on our scMicro-C technique and their acknowledgment of its potential impact. We apologize for the oversight in citing relevant literature and have now incorporated additional references (Tri-C and MC-4C) to address this issue. Furthermore, we have revised the manuscript to present our findings in a more balanced and nuanced manner, in line with the Reviewers’ suggestions.

For your convenient review, we have highlighted all changes in the revised manuscript.

USEFUL READS

Line 116, Fig1e: why do they only get 10-25% of reads that contain contacts? The Micro-C protocol should give 100% contacts assuming you use biotinylated nucleotides and streptavidin pull-down.

Looking at the protocol (thanks to the authors by the way for providing the protocol, which is essential), it looks like they skip the biotin pull-down similar to the MCC/TMCC papers. This will result in a tiny fraction of reads being useful (see Fig. 1b in Goel 2023 for a comparison). If they skip this crucial step, the authors should include a brief discussion that compare to “regular Micro-C” (Hansen 2019, Hsieh 2020, Krietenstein 2020, Goel 2023) their protocol is substantially less efficient in

terms of sequencing. Their protocol may have other advantages, but this skipping biotin-pull down makes sequencing quite wasteful and the authors should acknowledge this in the text to keep the comparison to Micro-C balanced.

We appreciate the Reviewers for the insightful comment. Indeed, we chose to omit the biotin pulldown step in our scMicro-C protocol, similar to our earlier Dip-C method (Tan et al., 2018) and other single-cell Hi-C protocols (Flyamer et al., 2017). This decision was driven by concerns over significant DNA loss associated with biotin enrichment for single-cell input Hi-C product, which result in limited chromatin contacts detected in single cells, as demonstrated in Dip-C paper: 1 million (no biotin pulldown Dip-C) versus 0.1 million (biotin pull down scHi-C).

To ensure the high detection efficiency of chromatin contacts in single cells, we opted to omit the biotin pulldown step. We acknowledged that this approach results in a lower proportion of useful reads (15-25% containing contacts) compared to the bulk Micro-C. We have now included a brief discussion in the revised manuscript to address this limitation (see Page 4, Line 119-120), noting that the contact ratio is compromised compared to bulk Micro-C protocols employing biotin pulldown.

Along the lines of Goel 2023 Fig 1b, can the authors please report fraction of “useful reads”, (unique cis reads, >1kb distance), to help compare the protocols more easily (Extended Data Fig. 2 is a bit hard for me to decipher).

We apologize for this confusion, actually, we have the “useful reads” information in our figure that is Fig.1 e, that shows the proportion of reads containing valid contacts (> 1 kb distance plus inter-chromosomal contacts).

Since skipping the biotin-pull-down is a significant departure from the original bulk Micro-C method, I think more clearly explaining this early on would be helpful. E.g. in line 140-143 they compare the original and the new Micro-C protocols, but they do not explain what the key differences are.

We appreciate the Reviewers’ feedback and apologize for any confusion in our manuscript. To clarify, we want to clarify that the omission of biotin pulldown was applied solely in the scMicro-C protocol, not in the modified bulk Micro-C protocol.

The key distinctions between the original bulk Micro-C and our modified protocol are twofold:

1. The modified bulk Micro-C protocol employs SDS treatment to improve ligation efficiency, whereas the original bulk Micro-C protocol does not incorporate this step.
2. Additionally, in the modified bulk Micro-C approach, all ligated DNA fragments spanning all lengths are sonicated to approximately 350 bp for library preparation. Subsequently, biotin pull-down is performed to enrich for chromatin contacts. In contrast, the original bulk Micro-C method selectively uses ligated di-nucleosome DNA fragments for library preparation.

To improve clarity, we have revised the method section accordingly.

TITLE: If the paper is largely focused on “promoter stripes” why does the title say enhancer-promoter stripes? Stripes are typically originating from a single CRE, whereas an E-P interaction would result in a dot, not a stripe.

Thank you for your insightful comment. Our analysis revealed that “promoter stripes” serve to facilitate interactions between a promoter and multiple enhancers distributed along the stripe.

In consideration of your feedback, we will revise the title as “High-resolution 3D genome structures via single-cell Micro-C uncover multi-enhancer hubs on genes with promoter stripe”

Line 267-271, the authors analyze the data from Hsieh 2022, which found most E-P interactions to be robust to cohesin depletion, but it seems like the authors here report the opposite. Why is that? Is the strong effect of cohesin unique to stripy genes? If so, please state this more clearly so the reader can understand if the authors disagree with the general conclusions of Hsieh 2022, or if this is specific to the stripy genes.

We thank the Reviewers for this useful comment. In Hsieh 2022 paper, they found that approximately 22% of E-P loops were significantly weakened following cohesin depletion, accompanied by a global decrease in E-P interaction intensity of about

29.1%. Respond to your suggestion, we did a more detailed analysis, and found that the E-P interactions of stripy genes were more pronounced weakened upon cohesin depletion compared to non-stripy genes. Specifically, 21% of E-P interactions involving gene body enhancers in stripy genes were identified as E-P loops, compared to 10.5% in non-stripy genes. Among these E-P loops, 48.5% in stripy genes were significantly weakened, whereas only 35.5% of E-P loops in non-stripy genes exhibited such weakening (see following figure).

We have thoroughly expanded this paragraph to discuss the effect of cohesin depletion on E-P interactions (Page 9, line 280-285).

DEFINE RESOLUTION: In my first review, I asked the authors to define resolution. In line 150 they define resolution as the size of the bins used for plotting. I apologize if I was unclear, but I do not think this is a useful definition.

We can think of resolution in 2 ways: 1) in terms of data or 2) in terms of plotting. #2 is not so helpful, since any dataset can be plotted at any resolution. Even a Hi-C dataset with only 1 read across the genome can be plotted at 1bp resolution. Therefore, I would like to ask the authors to define “resolution” in terms of the data. Through the manuscript, they frequently discuss how their data reached this or that resolution. Therefore, if they want to discuss resolution in terms of their data, they should use a data-centric way of defining resolution.

E.g. line 365-366 says they reached 5kb resolution, but if resolution just refers to plotting, they could also plot their data at 1bp resolution and claim 1bp resolution. So what do you, and please define this quantitatively, mean by the data reaching 5kb resolution?

Thank you for your valuable feedback. We appreciate your clarification request regarding the definition of “resolution” in our manuscript. In our study, we have described resolution both in terms of plotting and data interpretation, which may have caused confusion.

Specifically, when referring to bulk Micro-C/Hi-C or ensemble scMicro-C contact maps, we pertain to plotting resolution, which defines the bin size used for visualization purposes. There is no consistent definition of whether a bulk contact matrix achieves a specific resolution. Our criterion for determining resolution in this context is whether the contact matrix can clearly reveal distinct features such as chromatin domains, stripes, or loops at that scale, rather than noise.

However, when discussing the resolution of reconstructed single-cell 3D genome structures, we adopt a data-centric perspective. Here, “resolution” quantitatively signifies the capability of our scMicro-C method to reconstruct 3D genome structures with precision. For instance, achieving a 5 kb resolution implies that the reconstructed structures exhibit low uncertainty (quantified by low root-mean-square-deviation) across 5 independent reconstructions.

We hope this explanation addresses your concerns satisfactorily. We have added this claimant in the updated manuscript.

SMALL THINGS

Fig 1j, specify color scale, is it linear or log, what’s the dynamic range?

We thank the Reviewers for the valuable suggestion. We have revised Fig. 1j to specify the color scale.

Fig 2b, not clear what is above and below the diagonal from the figure, please label

We apologize for the unclear labeling of Fig. 2b. Above the diagonal represents the pairwise mean 3D distance measured from reconstructed single-cell 3D genome structures at the indicated resolution, while below the diagonal indicates the bulk Micro-C contact frequency. We have modified the labeling to enhance clarity and understanding.

Line 230, yes the original stripe reports were architectural, but many papers since then have studied CRE stripes including some of the ones cited here, no need to overstate things.

We thank the Reviewers for this suggestion. This paragraph was retained based on another reviewer's recommendation to differentiate between "promoter stripe" and previously reported "architectural stripe."

Line 374, what's a chromatin hairpin?

We apologize for the confusion, the chromatin hairpin (Matthey-Doret, Cyril et al., Nature Communications, 2020) is a similar structure to chromatin jets/fountain, which is not widely studied. To avoid confusion, we have deleted the chromatin hairpins in the updated manuscript.

Reviewer #3:

Remarks to the Author:

In this revised and strongly reworked manuscript, the authors focus on the development of the scMicro-C technology and use it for the analysis of "stripy" genes. Overall, I find the coherence and focus of the manuscript much improved. As mentioned in my previous evaluation, the results from the scMicro-C assay are impressive and the most novel aspect. The increased attention, combined with the addition of data from more cells, make the novelty better emerge. In its current state, I find the manuscript much better suited for publication in Nature Genetics.

We sincerely appreciate the Reviewers' enthusiasm for our revised manuscript and their acknowledgment of its significant improvement. Your valuable comments and suggestions have definitely enhanced the quality of our manuscript.

For your convenient review, we have highlighted all changes in the revised manuscript.

Minor issues:

- Line 41: the authors should precise that the 1 million CREs are highly cell type specific, with a much smaller subset active in individual cells.

We sincerely appreciate the Reviewers for this suggestion. We have made necessary revision to ensure accurate description. Please refers to Page 2, line 41-42.

- Line 48: the observations in citations [15,16] were preceded by citation [12]. It should be added.

We appreciate the Reviewers' expertise in the literature. We have corrected the order of citations in the revised manuscript as per your suggestion.

- Line 56: this observation was made in two preceding research studies: Calderon et al, eLife 2022 and Kane et al, Nat Struct Mol Biol 2022. They should be added.

Again, we thank the Reviewers for your professional knowledge in the field. In the updated manuscript, we have added these two citations.

- Line 57: citation [12] determined the scanning effect of loop extrusion with single cell precision. A study by Chang et al, Nature Communications 2023 used a Nano-C assay to further characterize intra-TAD loop scanning in the presence and absence of functional Cohesin. Both should be added to citations [26-28].

We thank the Reviewers for this valuable suggestion. We have added above mentioned two literatures in the corresponding text.

- Line 109: samples should read individual nuclei?

Yes, we have changed the corresponding text.

- Section 114-120: for a better understanding of these results, it will be useful if the authors can mention in the text or figure how many cells were included in each category.

We thank the Reviewers for the important suggestion. We have added the cell number information in the corresponding analysis in the figure legend.

- Line 130-131: I don't understand this sentence. How does the 355 cells relate to the 800 cells?

We apologize for the confusion caused by our unclear description. Initially, we generated 70 cells for SDS treatment comparison and 384 cells for MNase titration experiments, respectively. Following the establishment of optimal experimental conditions, we generated an additional 355 single cells for downstream analysis. In total, our dataset includes more than 800 cells. We have now included specific cell number information in each figure legend and summarized it in a supplementary table for your reference.

- Section 137-148: In figure 1, the authors show that their improved protocol has a very strong impact on ligation for single-cell analysis. It's surprising to see then that the bulk experiment appears very similar to the original Micro-C procedure. Could the authors reflect why this difference does not appear when comparing the bulk results? Do the two experiments differ in features that are not discussed (e.g. the number of reads containing ligation events?). Is the effect of the improved protocol for single cells largely mitigated by the incorporation of biotin enrichment in the bulk protocol?

We appreciate the Reviewers' insight on this matter. The comparison between our modified bulk Micro-C and the original bulk Micro-C protocol serves to demonstrate that our modifications preserve the high-resolution chromatin contact map featured by Micro-C. Specifically, the modifications involve SDS treatment and the utilization of all ligated DNA across various fragment lengths for sequencing, as opposed to the original bulk Micro-C method which does not involve SDS treatment and only select di-nucleosome length ligated DNA for sequencing.

In our understanding, the improvements in our protocol should ideally enhance aspects such as starting cell number and library complexity for bulk samples. This enhancement should result in a greater number of ligation events (contacts) compared to the original Micro-C protocol.

- Section 158-214 and Figure 2: although the results that are presented in this section are impressive, they do not expand much on the methodology and results that the authors previously reported using their Dip-C approach. In its current state it therefore remains mostly limited to a proof-of-principle, rather than showing how scMicro-C goes beyond current technology. Could the authors present their results more in the context of a comparison with their previous Dip-C result? Could the authors expand the functional interpretation of the result shown in Fig. 2D-F?

We have demonstrated in the manuscript, Dip-C has the potential to reconstruct 5 kb resolution 3D genome structure when deep sequenced, due to the limited cell number, we were not able to perform systematic benchmark the structure-related analysis. We have included the 3D distance analysis of Dip-C cell and compared with scMicro-C, the results show the 3D distance between Dip-C and scMicro-C show high correlation (see below). We have incorporated these results in the revised manuscript (Page 7, Line 196-199).

We have expanded the functional interpretation of the co-occurrence of nested CTCF loops, which we speculate is the reflection of the synergistic stabilization effect of spatially clustered CTCFs (Page 7, line 220-222).

- Line 192: correlation should read anti-correlation

We have corrected this error in the corresponding text.

- Line 203: it may be good to indicate that these loop anchors are anchored by CTCF

We thank the Reviewers for this suggestion. We have included this information in the revised manuscript (Page 7, line 221).

- Line 212: the same nested structures were characterized by MC-4C as well. Citation [12] should be added.

We apologize for the omission of this citation. We have added this citation in the revised manuscript according to your suggestion (Page 7, line 219).

- Line 221: instead of “downstream genomic region” use “gene body”?

We appreciate the Reviewers suggestion regarding precise language. We have revised the text accordingly.

- Section 236-256: the order of supplemental figure panels is inversed. Fig. S10a,b appear first, followed by Fig. S10c-e and I, Fig. S9b,c and finally Fig. S10g,h. The organization of Figs. S9 and S10 should be reordered.

We appreciate the Reviewers’ helpful suggestion. In response, we have reorganized Extended Data Fig. 9 and Fig. 10 to ensure they are presented in the correct order as referenced, please see the updated extended data figures.

- Lines 265 and 275: Figs. 2e and 2f should be 3e and 3f?

We apologize for the oversight. This has been corrected in the revised manuscript.

- Fig 1j: could the authors indicate the size of Chr5? Adding tick marks on the axes will make the matrices easier to read. This remark expands to all interaction matrices in the supplemental information as well.

We sincerely thank the reviewers for this valuable suggestion. In response, we have updated all contact maps, including Fig. 1j and those in the supplemental information, to include tick marks on the axes. Please refer to the revised figures for improved readability.

- Fig. 3a: the title “stripy – non-stripy” is not intuitive. Could this be replaced by a more clear description (e.g. delta(stripy vs non-stripy))

We thank the Reviewers for this good suggestion. We have renamed the title of Fig. 3a according to your suggestion.

- Fig. 3f: I'm not sure what this model is supposed to show. I don't understand what the relative position of floating bubbles is supposed to show, and how this links to the arrows. Are certain lines/connections are missing? Is this supposed to show the scanning mechanism? Based on the recent result from Barshad et al, Nature Genetics 2023, can the authors formally exclude that scanning is not (in part) initiated from the enhancers? The title contains a typo: medaited should read mediated.

We apologize for the confusion caused by the unclear schematic in Fig. 3f. In response to your feedback, we have revised Fig. 3f to improve its clarity (see the updated figure below).

Fig. 3f is intended to illustrate the model in which cohesin-mediated loop extrusion facilitates the scanning of the promoter to multiple enhancers distributed along the stripe. Regarding the recent findings from Barshad et al., Nature Genetics 2023, we acknowledge the possibility that scanning could also initiate from enhancers, although our data primarily focus on promoter scanning. Actually, we have observed weak stripes originating from enhancers in some stripy genes (Extended Data Fig. 9a) and the enhancer-enhancer pile-up exhibiting weak stripe patterns (Extended Data Fig. 10k). This suggests that scanning could indeed be partially initiated from enhancers. We have added a discussion of this possibility in the revised manuscript (Page 10, line 290-293).

- Fig. S2a: I'm curious to see that the patterns around CTCF motifs are very similar, whereas a considerable difference is observed for TSS upon treatment with 200U and 1000U. Can the authors reflect on this?

We appreciate the Reviewers' insightful comment. We would like to clarify the nucleosome occupancy values presented in Extended Data Fig. 2a are normalized to the mean value for each cell. This normalization reflects relative nucleosome signals rather than absolute values.

The nucleosome patterns around transcription start sites of active genes are known to be sensitive to MNase digestion conditions. Therefore, the observed differences in nucleosome occupancy with varying MNase concentrations are expected. This observation is consistent with previous studies, such as Schwartz and Längst (*Nucleic Acids Research*, 2019, <https://doi.org/10.1093/nar/gky1203>, Fig. 5C). We hope this explanation addresses your query.

- Fig. S2c: what is the difference between the Pooled and scMicro-C tracks? Is the scMicro-C track the larger number of 800U-treated cells that are used in the remainder of the study? The authors should clarify this. Adding the number of cells that each track is based on will be useful.

We apologize for any confusion caused by the unclear labeling. The 'pooled' track represents signal from 340 cells (15 excluded due to poor quality), while the 'scMicro-C' track includes data from all single cells. Following your valuable suggestion, we have added this clarification regarding the number of cells used in each track to ensure clarity.

- Fig. S6. Add the word "ensemble" to the title (i.e. Ensemble scMicro-C faithfully ...)

We thank the Reviewers for this suggestion. We have modified the title of Extended Data Figure 6 accordingly.

Reviewer #4:

Remarks to the Author:

The manuscript by Wu and colleagues has been very extensively revised. The authors still present the single-cell Micro-C method, but have removed most of their perturbation experiments from the manuscript. As such, the focus of the paper has changed substantially. The main messages of the paper are now: (1) development of

single-cell Micro-C; (2) long genes often form stripes which allow promoters to interact with enhancers; (3) enhancer-promoter interactions are dependent on Cohesin; and (4) enhancers and promoters interact cooperatively in hubs.

Although my technical concerns with respect to the single-cell Micro-C method have been addressed, the value of this method is still not convincingly demonstrated, as points 2-3 are mostly supported by bulk analysis. In addition, both the observation of stripes at gene promoters and their dependence on Cohesin are not novel contributions. The cooperative enhancer-promoter interactions do rely (to some extent) on single-cell analysis, but it is important to mention that similar structures have been observed in multi-contact 3C analysis in two back-to-back Nature Genetics papers in 2018 that have not been acknowledged (<https://pubmed.ncbi.nlm.nih.gov/29988121/> / <https://pubmed.ncbi.nlm.nih.gov/30374068/>).

As such, the new biological insights from this paper are very limited, but I do think that the development of the single-cell Micro-C approach could be of interest to the field.

We sincerely appreciate the Reviewers for their comprehensive comment. First of all, we would like to apologize for the omission of these two critical papers. We have added the two literatures in our revised manuscript (Page 11, line 345-347).

We would like to argue that though similar “promoter stripe” structures have been reported in previous literature, which is claimed in our manuscript, they are not identical, the features are different and not been extensively characterized. Our focus is not on the stripe, instead, we focus on the regulatory function this structure—mediating multiple enhancer-promoter interactions. It’s our first time to report that the promoter stripe is associated with enhancer-promoter interactions, we found that genes with “promoter stripe” typically have multiple enhancers distributed along the stripe and promoter exhibits strong focal interactions at the enhancer loci, and for inter-connected enhancer network structures.

Though similar multi-enhancer hub structure has been reported in previous literature, which is also recognized in our manuscript. While previous study only focused on a specific gene (e.g., α/β -globin gene locus). We do not apply scMicro-C to prove the multi-enhancer hub structure in these known loci. Instead, we *de novo* reported the

same phenomenon in a large number of stripy genes, and these multi-enhancer hubs could only be observed using high-resolution single-cell 3D genome structures. Thus, we think it's unfair to comment that “the value of this method is still not convincingly demonstrated.”

Moreover, to gain more mechanistic understanding of the multi-enhancer hub structure observed in stripy genes, we newly added polymer simulations results. Specifically, we employed polymer simulations to explore the mechanism underlying the interconnected multi-enhancer network at the EBF1 locus in stripy genes. The simulation results suggest that a combination of strong molecular affinities between promoters and enhancers, along with cohesin capture by enhancers, most accurately reflects our experimental observations. Notably, the single chromatin fiber simulations closely resemble our scMicro-C results, including the observed multi-enhancer hub structures. For a detailed description, please refer to the newly added section, “**Polymer simulations recapitulate multi-enhancer hub structure of stripy genes,**” and Extended Data Figs. 14 and 15.

Reviewers' Comments:

Reviewer #2 (Remarks to the Author):

Being a reviewer for this paper is a bit strange, because I think the paper is pretty good, scMicro-C is technically impressive, and I have been supportive of publication, and I expect scMicro-C to receive general interest if published. In my reviews, I have asked for minor text changes and a couple of extremely simple calculations (like define and calculate resolution, calculate unique cis-reads). Whenever I submit a paper, my dream reviewer is someone who is supportive and asks for minor text changes and minor calculations that take less than a day to do, without asking for extensive experiments. Yet, despite me asking twice, the authors have still not defined resolution. This is a problem because the authors use the term “resolution” 61 times and often use it quantitatively to claim their method reached e.g. 5-kb resolution. I genuinely do not understand why the authors do not want to calculate the resolution when it is easy to do and there are well-known definitions such as Rao 2014. But if they do not clearly define resolution, all their resolution claims are meaningless. So I would like to INSIST for the THIRD time that the authors calculate the resolution of their maps or remove the word “resolution” from their paper. More about this below. I really don't get the resistance, it is easy to do, and other than this the paper is mostly pretty great, can you just please define and calculate resolution using the Rao 2014 definition?

We sincerely appreciate the Reviewer's valuable feedback. We apologize for any misunderstanding regarding the definition of resolution. We fully acknowledge the importance of quantitatively defining resolution in our claims. In response to your suggestion, we have calculated the resolution using the definition from Rao et al. (2014).

Additionally, we have reduced the frequency of the term "resolution" throughout the revised manuscript to enhance clarity.

Thank you once again for your constructive comments.

RESOLUTION:

In my first review, I asked the authors to quantitatively define resolution. They did not do it.

In my second review, I asked the authors to quantitatively define resolution. They still did not do it.

They give some vague responses that are not clear and they say that there are no clear definitions of resolution. I do not think this is true. The Rao 2014 Hi-C paper (PMID: 25497547), which claims kilobase resolution, is arguably one of the top 3 most famous papers in the 3D genome field. It has about 8,000 citations. In their main text, they

define resolution QUANTITATIVELY as “We define the “matrix resolution” of a Hi-C map as the locus size used to construct a particular contact matrix and the “map resolution” as the smallest locus size such that 80% of loci have at least 1,000 contacts.”. I think their language is a little unclear, so what they mean is that the resolution of the contact map is the bin size where >80% of row-columns has at least 1000 unique contacts. So, if you have 5-kb resolution, at least 80% of each 5-kb segment along the genome should have at least 1000 unique contacts. Please use this definition, perform this calculation, and report the outcome in main text and main figures.

Why do I ask the authors to define “resolution”? Because they use it 61 times in the manuscript and often use it quantitatively along the lines of “our method is awesome, it achieves 5kb resolution”. These statements are meaningless if they are not clearly quantitatively defined. And honestly, I am a little tired of this. Asking people to define their terms is not unreasonable and I have already asked them twice and it takes only some minor quick calculations.

So, in my THIRD review, I am asking them for the THIRD time to define resolution. I would like to ask the authors to use the definition of Rao 2014 outlined above. This is a famous definition from a paper with 8000 citations that every person in the 3D genome field has read and knows. I would like to ask the authors to report the resolution of the maps only using this Rao 2014 calculation (PMID: 25497547).

Overall, I think their scMicro-C technique is impressive and I think this paper is generally suited as a technical report for Nature Genetics and expect it to receive wide interest. But if the authors refuse to define and quantitatively calculate resolution – when it is easy and fast to do – after asking them 3 times, I honestly think this paper should just be rejected and I really do not understand why the authors are so opposed to define resolution.

Thank you for your insightful comments regarding the definition of resolution in our manuscript. We recognize that the term “resolution” was used in three contexts: bulk or ensemble Micro-C, reconstructed single-cell 3D genome structures, and single-cell contact maps. Below, we provide quantitative definitions for each scenario.

1. **Bulk or Ensemble Micro-C:** We have calculated the resolution as suggested using the Rao et al. (2014) definition. For our bulk Micro-C dataset, we found that at 1 kb resolution, over 80% of bins had contacts greater than 1202. For our ensemble scMicro-C, we achieved 5 kb resolution, with more than 80% of loci having contact counts exceeding 1303. A detailed plot is included below. And we have added corresponding description in the manuscript.

2. **Reconstructed Single-Cell 3D Genome Structures:** The conclusion of 5 kb resolution is based on the small deviation observed between independent reconstructions (Extended Data Figure 7a), as outlined in the methods section.
3. **Single-Cell Contact Maps:** We referenced the definition provided by Ulianov et al. (Nature Communications 2021, PMID: 33397980), which states that at a specific resolution, the percentage of non-zero bins is critical. For our scMicro-C data, we found that at 5 kb resolution, the median percentage of non-zero bins is 78%. A histogram illustrating this is provided below.

We hope these calculations adequately address your concerns.

Once again, we sincerely appreciate your support and constructive feedback, which have greatly enhanced the clarity of our manuscript.

SMALL THINGS:

1. I asked the authors to show the percent of useful contacts defined as percent of reads that are Unique+CIS+>1kb. Not sure why, but the authors say the provided this in Fig. 1e, but according to the figure legend, Fig. 1e does not show this. Can they please just report Unique+CIS+>1kb?

We apologize for the confusion. We misunderstood your question. In Fig. 1e, the contact was defined as cis > 1 kb plus trans contacts. If you only want uniq cis > 1 kb only, below is the plot showing the ratio of deduplicated unique cis >+1 kb in raw reads.

2. Line 405: Typo “aw well as” should be “as well as”

We have corrected this error in the revised text.

3. The new polymer simulations seem interesting!

We sincerely appreciate for this comment.